# CURATE: Automatic Curriculum Learning for Reinforcement Learning Agents through Competence-Based Curriculum Policy Search in Structured Task Spaces

## Abstract

Due to fundamental exploration challenges without informed priors or specialized algorithms, agents may be unable to consistently receive informative rewards, leading to inefficient or intractable learning. To address these challenges, we introduce CURATE, an automatic curriculum learning algorithm for reinforcement learning agents in structured task spaces of monotonic difficulty. Through "exploration by exploitation," CURATE dynamically scales the task difficulty to match the agent's current competence. By exploiting its current capabilities that were learned in easier tasks, the agent improves its exploration in more difficult tasks. Our key insight is that the learning improvement in tasks that are close to those used for training is inversely proportional to their difficulty, and an agent that chooses a nearby distribution of the easiest unsolved tasks at any given time can automatically induce an easiest-to-hardest curriculum in these task spaces. To achieve this, CURATE conducts policy search in the task space to learn the best task distribution for training. As the agent's mastery grows, the learned curriculum adapts in an approximately easiest-to-hardest and task-directed fashion, efficiently culminating in a performant agent. Our experiments across three diverse domains (MiniGrid, Procgen, BipedalWalker) demonstrate that CURATE learns effective curricula for sample efficiency and generalization, matching or exceeding prior curriculum methods with the potential for yielding broadly capable agents.

## 1 Introduction

The advent of reinforcement learning (RL) (Sutton & Barto, 2018; Kaelbling et al., 1996) with deep neural networks (LeCun et al., 2015; Goodfellow et al., 2016) has ushered in a promising era of impressive milestones in sequential decision making for deep RL (Mnih et al., 2013; 2015; Silver et al., 2018; Vinyals et al., 2019; OpenAI et al., 2019b;a; Li, 2018; 2023). Yet, without models, inductive biases, expert trajectories, or dense rewards, model-free deep RL algorithms are markedly sample inefficient due to fundamental challenges with exploration. Initially, the RL agent's actions are essentially random, requiring many interactions with the environment before the agent can learn useful behaviors that accrue rewards. However, agent learning can be structured through *curriculum learning* (Bengio et al., 2009), which specifies how training data should be sequenced in order to achieve two broad aims (Wang et al., 2022): to guide training (i.e., increase learning sample efficiency) and to denoise training (i.e., improve learning robustness and generalization through focus on high-confidence training data regimes). The advantages of improving sample efficiency, generalization, and exploration through a curriculum are generally recognized (Narvekar et al., 2020; Portelas et al., 2021; Parker-Holder et al., 2022b). However, achieving *automatic curriculum learning* in general — automatically learning the optimal curriculum for *any* domain and the transformative impact it would entail — remains an open problem, as "some curriculum strategies work better than others" (Bengio et al., 2009).

To make progress towards this problem, we introduce CURATE (**Cur**riculum **A**gent for **T**argeted **E**xploration),[1] an automatic curriculum learning algorithm designed for training a model-free, on-policy

---

[1]Upon acceptance, we will open-source CURATE, our codebase, and the Procgen Curriculum Suite.

reinforcement learning agent to solve a difficult target task distribution where rewards are sparse or uninformative. Our approach overcomes fundamental exploration challenges by conducting *exploration by exploitation*, as coined by Leibo et al. (2019).[2] Specifically, CURATE adapts the difficulty of the training tasks to the agent's current capabilities, or *competence*, through curriculum policy search, analogous to how children self-design curricula (Dahmani et al., 2025). Initially, the agent has not learned useful behaviors, so relatively easier tasks ensure that random exploration is (relatively more) viable such that rewards can be more consistently obtained. Then, as the agent's competence grows, more difficult training tasks are selected to match the current capabilities of the agent. In other words, the agent improves its ability to explore in more challenging tasks by exploiting its current capabilities that were gained from previous easier tasks. Our key insight is that tasks that are nearby those used for training will exhibit performance increases inversely proportional to their difficulty (App. F.1), and by choosing the easiest (i.e., highest return) unsolved tasks, CURATE can induce an easiest-to-hardest curriculum in structured task spaces. In this way, CURATE trains an RL agent through an approximately easiest-to-hardest progression, quickly training the agent to complete the target task distribution at the end of the curriculum. This progression can also yield greater performance more widely beyond the target task, broadening CURATE's applicability.

Our contributions are as follows. First, we introduce CURATE, an automatic curriculum learning algorithm that learns an approximately easiest-to-hardest sequencing of training tasks using competence-based curriculum policy search. Although CURATE is designed for solving a difficult target task distribution, we find that, in some domains, it can train a broadly capable agent that generalizes to other tasks. Second, our findings across three domains that are diverse in task parameterization, dimensionality, actions, and observations show that learned CURATE curricula train agents that match or outperform prior curriculum baselines. Lastly, we introduce the Procgen Curriculum Suite, which adds a structured task space to each Procgen game, enabling rigorous and systematic benchmarking of curriculum learning in this domain.

## 2 Related work

**Curriculum learning for reinforcement learning** As formalized by Bengio et al. (2009), curriculum learning concerns how to meaningfully organize data for training machine learning models, including those used for reinforcement learning. For comprehensive surveys in curriculum learning for RL, please refer to Narvekar et al. (2020), Portelas et al. (2021), and Parker-Holder et al. (2022b). Both Graves et al. (2017) and Matiisen et al. (2019) introduce automatic curriculum learning methods based on nonstationary multi-armed bandit algorithms based on measures of progress, e.g. learning progress (Oudeyer et al., 2007; Oudeyer & Kaplan, 2007). Wang et al. (2019; 2020) show that curricula can emerge from co-evolving environments and agents. Portelas et al. (2020) introduce ALP-GMM, a Gaussian mixture model in the parameter space of the environment. Klink et al. (2020) maintains a Gaussian training task distribution with a Kullback-Leibler divergence objective to induce task directedness. Huang et al. (2022) and Klink et al. (2024) explore curriculum learning as optimal transport, leading to the GRADIENT and CURROT algorithms, respectively. Li et al. (2023) propose that, under certain assumptions with one-dimensional curricula, solving tasks from easiest to hardest is optimal. Zhang et al. (2024) and Faldor et al. (2025) show that autocurricula can emerge when leveraging foundation model priors. Our algorithm, CURATE, is similar to Portelas et al. (2020) and Klink et al. (2020). CURATE also maintains a training task distribution similar to these methods, but a key distinction of CURATE is that its learning frontier is advanced through seeking out the easiest set of unsolved tasks in the direction of the target task distribution. Importantly, CURATE does not require an initial task distribution or predefined schedules and is suitable for relatively larger task spaces.

**Solving difficult tasks using curricula and exploration methods** The question of exploration, i.e., how to train RL agents to solve difficult tasks without dense rewards, has overlap between methods in curriculum learning and exploration. For curriculum learning, Florensa et al. (2017) introduce a curriculum learning algorithm that modifies the agent starting state, starting from a goal state and growing in reverse. This method was generalized in BaRC (Ivanovic et al., 2019) through integrating prior knowledge, e.g.,

---

[2]Leibo et al. (2019) describe "exploration by exploitation" as a form of exploration in agents that continuously adapt to exploit their abilities in non-stationary environments. In our work, our environments are not non-stationary, but the training distribution is made non-stationary by the learned curriculum policy.

physical priors. Tao et al. (2024) combine a reverse curriculum over expert demonstrations with forward curriculum learning. For exploration, approaches can be categorized based on their methodology, e.g., counts (Burda et al., 2019; Henaff et al., 2022), curiosity (Pathak et al., 2017; 2019; Raileanu & Rocktäschel, 2020), memory (Badia et al., 2020), or information theory (Seo et al., 2021). For more information, please refer to Amin et al. (2021) and Ladosz et al. (2022) for surveys. Both curriculum learning and exploration algorithms improve training efficiency by "densifying" the reward signal for the agent. CURATE achieves this by adapting the task difficulty such that the agent receives consistently available extrinsic rewards.

**Unsupervised Environment Design and Dual Curriculum Design**   First introduced by Dennis et al. (2020), the Unsupervised Environment Design (UED) paradigm provides a framework wherein parameters of an underspecified environment are varied by a teacher to produce distributions over environments for a student learner. Jiang et al. (2021a) unify the UED framework with prior work in replaying experiences with Prioritized Level Replay (PLR) (Jiang et al., 2021b) to form the Dual Curriculum Design (DCD) framework, while also introducing Robust PLR (also stylized as $PLR^{\perp}$) in which gradient updates only occur on replayed levels (i.e., tasks). Later, Parker-Holder et al. (2022a) introduce ACCEL, an evolutionary-based algorithm that randomly mutates levels starting from environments of minimal complexity. Other UED algorithms include Samvelyan et al. (2023), Mediratta et al. (2023), Beukman et al. (2024), and Rutherford et al. (2024). Although we do not consider CURATE to be a UED algorithm as it is concerned with curating a distribution of tasks (instead of the design of individual tasks), CURATE can be placed within the DCD framework by functioning as a teacher that offers levels that are at the leading edge of the student's competence.

## 3   Preliminaries

### 3.1   Underspecified POMDPs

The agent learns within an Underspecified Partially Observable Markov Decision Process (UPOMDP) framework (Dennis et al., 2020). The UPOMDP defines a distribution of Partially Observable Markov Decision Process (POMDP) tasks (Åström, 1965; Kaelbling et al., 1998) as determined by the selection of environment parameters. The UPOMDP is defined as follows:

$$\mathcal{M} = \langle \mathcal{A}, O, \Theta, \mathcal{S}^{\mathcal{M}}, \mathcal{T}^{\mathcal{M}}, \mathcal{I}^{\mathcal{M}}, \mathcal{R}^{\mathcal{M}}, \gamma \rangle \tag{1}$$

where $a \in \mathcal{A}$ is a set of actions, $o \in O$ is a set of observations, $\theta \in \Theta$ is a set of environment parameters, and $\gamma$ is a discount factor for future rewards. The remainder of the UPOMDP tuple is defined with respect to the chosen environment parameters $\theta$ and are thus superscripted by $\mathcal{M}$. Therefore, for the POMDP $\mathcal{M}_\theta$ specified by $\theta$, $s \in \mathcal{S}^{\mathcal{M}} : \mathcal{S} \times \Theta$ is a set of states from state space $\mathcal{S}$ that are not observable to the agent, $\mathcal{T}^{\mathcal{M}} : \mathcal{S} \times \mathcal{A} \times \Theta \rightarrow \mathcal{S}$ defines the transition function, $\mathcal{I}^{\mathcal{M}} : \mathcal{S} \times \Theta \rightarrow O$ is the observation (i.e., introspection) function, and $R \in \mathcal{R}^{\mathcal{M}} : \mathcal{S} \times \mathcal{A} \times \Theta \rightarrow \mathbb{R}$ is the reward function. A task is considered solved if its reward meets or exceeds a solved threshold $R_S$, which for our work, is assumed to be known prior to learning. Through reinforcement learning, the agent learns a policy $\pi(a|o)$ from maximizing the objective $J(\pi)$, the expected sum of discounted rewards over trajectories $\tau$ with maximum timesteps $T$:

$$J(\pi) = \mathbb{E}_{\tau \sim \pi} [\sum_{i=0}^{T} \gamma^i R_i] \tag{2}$$

In principle, the UPOMDP framework allows a temporally-varying trajectory of environment parameters, but in practice, we are concerned with environment parameters that only specify the construction of the task state space $\mathcal{S}$ via the underspecified state space $\mathcal{S}^{\mathcal{M}}$. This leads to the space of environment parameters $\Theta$ forming the *task space* of the UPOMDP. In this view, our use of UPOMDPs is similar to Contextual MDPs (Abbasi-Yadkori & Neu, 2014; Hallak et al., 2015; Modi et al., 2018).

### 3.2   Curriculum learning within DOUPOMDPs

For this work, we investigate a subclass of UPOMDPs where the environment parameters $\Theta$ and thus tasks are *difficulty ordered.* We introduce this UPOMDP subclass as the Difficulty-Ordered Underspecified Partially

Observable Markov Decision Process (DOUPOMDP). In the DOUPOMDP framework, the environment parameters $\Theta$ and thus tasks are ordered such that $\forall \theta_a, \theta_b \in \Theta, \ \theta_a \preceq \theta_b \implies \text{diff}(\mathcal{M}_{\theta_a}) \leq \text{diff}(\mathcal{M}_{\theta_b})$, where $\preceq$ is the standard partial order and diff is the difficulty of a task. In other words, tasks are arranged in monotonic difficulty order along each dimension of $\Theta$. However, the order of tasks by difficulty *across* dimensions is not obvious, and no claims are possible *a priori* in regards to $\text{diff}(\mathcal{M}_{\theta_a})$ and $\text{diff}(\mathcal{M}_{\theta_b})$ if $\theta_a \npreceq \theta_b$. Although we do not formally measure difficulty in this work, the difficulty of a task is proportional to the samples required to train a randomly initialized agent to obtain a reward of at least $R_S$ during policy evaluation. The easiest tasks occur at $\theta_e = \min(\Theta)$, and the hardest tasks occur at $\theta_t = \max(\Theta)$.

Specific to our work, $\theta_t$ are the environment parameters for our *target task distribution*, and the POMDP that specifies the target task distribution is $\mathcal{M}_{\theta_t}$. We also assume that the environment parameter space $\Theta$ is disentangled, i.e., each dimension controls a single factor of variation. Under the assumption that the environment parameters $\Theta$ are disentangled, curriculum learning can be conducted within the *axes of generalization* of the DOUPOMDP's *task space*, i.e., the space of environment parameters $\Theta$. The time-varying sequence of tasks that arises from a curriculum learning algorithm is called a *curriculum $\mathcal{C}$*.

## 4  Methodology

To accelerate exploration, CURATE automatically learns a curriculum $\mathcal{C}$ to train a control policy $\pi$. To do this, CURATE conducts policy search using sample-based evaluations to determine a curriculum policy that best shapes the training task distribution. Intuitively, the curriculum policy learns a local distribution of unsolved tasks with high returns that can be sampled for training. As the agent becomes more proficient and begins solving these tasks, this distribution shifts towards more difficult unsolved tasks in the direction of the target task distribution, approximating an easiest-to-hardest curriculum. The curriculum policy $\pi_c(\theta; \mu_\theta, \Sigma_\theta)$ is represented by a Gaussian distribution over environment parameters $\theta$ with mean $\mu_\theta$ and covariance $\Sigma_\theta$.

The training procedure for CURATE is summarized in Alg. 1; please see Alg. 2 (App. B.1) for a full description of the algorithm. Generally, CURATE follows the standard training procedure for training RL agents using parallel environments, except that the training tasks $\mathcal{M}_{\theta_c}$ are sampled from the curriculum policy $\pi_c$ via $\theta$. An off-curriculum sampling strategy can also be used, e.g., to improve generalization or mitigate forgetting. The agent conducts rollouts in $\mathcal{M}_{\theta_c}$ via $\pi$ to obtain a dataset of trajectories $\mathcal{D}$ with mean return $R_\mathcal{D}$. Following rollouts, the agent is updated via an on-policy reinforcement learning algorithm. Although any on-policy algorithm could be used in principle, we use Proximal Policy Optimization (PPO) (Schulman et al., 2017) due to its favorable performance and stability with both discrete and continuous domains. Following the update of the agent, curriculum updates may then be triggered if the training return $R_\mathcal{D}$ matches or exceeds the task solved threshold $R_S$. Note that for some domains, the task solved threshold used for learning may be different than the desired reward threshold for the target tasks (c.f., Sec. 5.2.1). Section 4.1 describes the curriculum update step, UPDATECURRICULUM. Note that an informed initialization is performed using DIRECTTOWARDSTARGET (Sec. 4.2), and CURATE hyperparameters can be initialized based on the properties of the task space $\Theta$ (Sec. 4.3).

### 4.1  Updating the curriculum using curriculum policy search

This section summarizes UPDATECURRICULUM, the curriculum update procedure that is fully described in Alg. 3 (App. B.2). UPDATECURRICULUM is a nonlinear, sample-based optimization within environmental parameter space $\Theta$ by using Relative Entropy Policy Search (REPS) (Peters et al., 2010) to learn the curriculum $\pi_c$ via the curriculum objective $J_c(\pi_c)$:

$$J_c(\pi_c) = \mathbb{E}_{\theta_j \sim \pi_c, \mathcal{M}_{\theta_j} \sim \theta_j, R_j \sim \mathcal{M}_{\theta_j}(\tau \sim \pi)}[\nu_j] \qquad (3)$$

$$\mathcal{R}_{comp} = \frac{R_j}{R_S} \mathbb{1}_{R_j < R_S \vee \theta_j = \theta_t} \qquad (5)$$

$$\nu_j = \begin{cases} \mathcal{R}_{comp} - \mathcal{L}_{diff}, & \text{initial update} \\ \mathcal{R}_{comp} - \mathcal{L}_{dist}, & \text{otherwise} \end{cases} \qquad (4)$$

$$\mathcal{L}_{diff} = \lambda_\theta ||\vec{v}_{\theta_j - \theta_e}||_2 \qquad (6)$$

$$\mathcal{L}_{dist} = \lambda_d ||\vec{v}_{\theta_j - \theta_i} \perp \vec{v}_{\theta_t - \theta_i}||_2 \qquad (7)$$

---

**Algorithm 1:** CURATE: CURRICULUM AGENT FOR TARGETED EXPLORATION (Simplified)

---

**Input:** task space $\Theta$, task solved threshold $R_S$, number of updates $N_u$, number of workers $N_v$
**Initialize:** control policy $\pi \leftarrow$ INITIALIZERANDOMPOLICY(), target task $\theta_t \leftarrow \max(\Theta)$, curriculum
  policy $\mu_{\theta_0}, \Sigma_{\theta_0} \leftarrow$ INITIALIZERANDOMCURRICULUMPOLICY($\Theta$) (Sec. 4.3),
  $\Delta\mu_\theta, \Sigma_{\theta_u}, \lambda_\theta, \lambda_d \leftarrow$ INITIALIZEHYPERPARAMETERS($\Theta$) (Sec. 4.3)

---

$\mu_\theta, \Sigma_\theta \leftarrow$ UPDATECURRICULUM($\pi, \mu_{\theta_0}, \Sigma_{\theta_0}, \lambda_\theta, use\_diff \leftarrow$ True) (Update via Eq. 3)
**for** $u = 1$ **to** $N_u$ **do**
    $\boldsymbol{\mathcal{M}_{\theta_c}} \leftarrow \emptyset$
    **for** $i = 1$ **to** $N_v$ **do**
        $\theta_i \sim \pi_c(\theta; \mu_\theta, \Sigma_\theta)$ (An off-curriculum policy can also be sampled here, e.g., Sec. 5.3-5.4)
        $\mathcal{M}_{\theta_i} \leftarrow$ TASKGENERATOR($\theta_i$)
        $\boldsymbol{\mathcal{M}_{\theta_c}} \stackrel{+}{\leftarrow} \mathcal{M}_{\theta_i}$
    **end**
    $\mathcal{D}, R_\mathcal{D} \leftarrow$ ROLLOUTAGENTONPARALLELTASKS($\pi, \boldsymbol{\mathcal{M}_{\theta_c}}$)
    $\pi \leftarrow$ UPDATEAGENT($\mathcal{D}$) (Update via Eq. 2)
    **if** $R_S \leq R_\mathcal{D}$ **then**
        $\mu_{\theta_u} \leftarrow$ DIRECTTOWARDSTARGET($\mu_\theta, \theta_t, \Delta\mu_\theta$) (Sec. 4.2)
        $\mu_\theta, \Sigma_\theta \leftarrow$ UPDATECURRICULUM($\pi, \mu_{\theta_u}, \Sigma_{\theta_u}, \lambda_d, use\_dist \leftarrow$ True) (Update via Eq. 3)
**end**

---

**Result:** control policy $\pi$

---

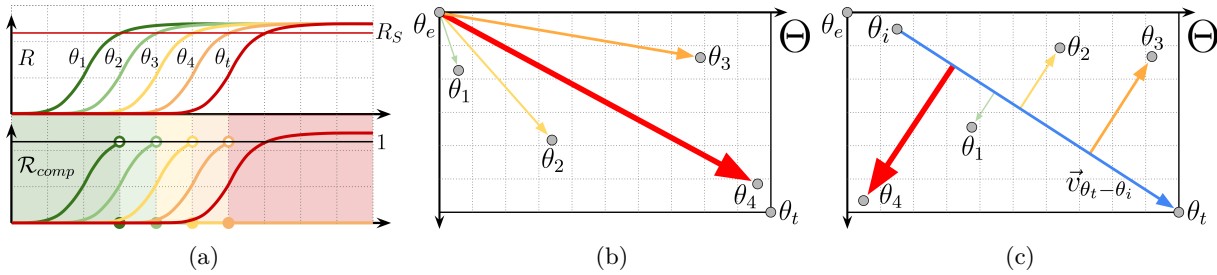

(a)                (b)                (c)

Figure 1: (a) Illustration of $\mathcal{R}_{comp}$. In this example, tasks are ordered in difficulty corresponding to $\theta_1 - \theta_t$. The curriculum reward is proportional to the return $R$, but if $R$ matches or exceeds $R_S$, the curriculum reward is reduced to zero. Exceptionally, the curriculum reward in the target task $\theta_t$ is not reduced to further boost performance in these tasks. (b) Illustration of $\mathcal{L}_{diff}$. This example illustrates that tasks are penalized proportional to their distance away from $\theta_e$. This loss term only occurs during the initial curriculum update, addressing the case where initial rewards are uninformative across the task space. (c) Illustration of $\mathcal{L}_{dist}$. For all curriculum updates except the initial update, tasks are penalized proportional to their distance from $\vec{v}_{\theta_j - \theta_i}$ (blue arrow). This term provides task directedness, encouraging progression towards the target tasks.

where $\mathcal{M}_{\theta_j}$ is a task distribution sampled from $\pi_c$ via parameters $\theta_j$, $R_j$ is the reward obtained by evaluating $\pi$ on $\mathcal{M}_{\theta_j}$, $\nu_j$ is the curriculum reward, $\mathcal{R}_{comp}$ is an objective that rewards competence in either the unsolved or target tasks, $\mathcal{L}_{diff}$ is a regularization loss that penalizes difficult tasks, and $\mathcal{L}_{dist}$ is a loss that induces tasks to remain close to $\vec{v}_{\theta_t - \theta_i}$, the vector that connects the initial task distribution $\theta_i$ learned in the initial update to the target task distribution $\theta_t$. Hyperparameters $\lambda_\theta$ and $\lambda_d$ are automatically determined based on heuristics (Sec. 4.3). The major components of $J_c$ (i.e., $\mathcal{R}_{comp}, \mathcal{L}_{diff}, \mathcal{L}_{diff}$) are illustrated in Fig. 1 and discussed further.

**Competence reward** The curriculum reward $\nu_j$ is primarily driven by the competence learning reward objective $\mathcal{R}_{comp}$ (Eq. 5, Fig. 1a). This objective rewards the curriculum policy $\pi_c$ to seek out the easiest (i.e., highest return) tasks that have not yet been solved by the agent. This prevents the curriculum policy from offering tasks the agent has already mastered, as those have high student reward $R_j$ but are at least

$R_S$, thus yielding zero curriculum reward. By matching tasks to the competence of the student, $\mathcal{R}_{comp}$ can be seen as a mathematical characterization of how children self-generate curricula (Dahmani et al., 2025). Note that the curriculum policy always receives full reward for target tasks, even if above $R_S$. This induces the training task distribution to remain close to $\mathcal{M}_{\theta_t}$ even if it is solved, boosting target task performance.

**Regularization loss**  During the initial curriculum update, a regularization loss $\mathcal{L}_{diff}$ (Eq. 6, Fig. 1b) is applied proportional to the distance of the sampled task parameters $\theta_j$ from the easiest task parameters $\theta_e$ via the vector $\vec{v}_{\theta_j - \theta_e}$. This loss addresses cases where samples consistently return zero reward (e.g., at the start of training, all tasks may be too difficult), leading to a bias towards easier tasks. Without this term, the curriculum would remain close to a uniform distribution if all sampled tasks return uninformative rewards. For all other curriculum updates beyond the initial update, this loss is not used, as it otherwise induces curriculum pressure away from the target task distribution and slows the agent's progression.

**Orthogonal distance loss**  For curriculum updates except the initial update, a loss $\mathcal{L}_{dist}$ (Eq. 7, Fig. 1c) is applied to improve the progression of the agent. A loss is applied that is proportional to the magnitude of $\vec{v}_{\theta_j - \theta_i} \perp \vec{v}_{\theta_t - \theta_i}$, the vector rejection of $\vec{v}_{\theta_j - \theta_i}$ from $\vec{v}_{\theta_t - \theta_i}$, where $\vec{v}_{\theta_j - \theta_i}$ is the vector from initial parameters $\theta_i$ to sampled parameters $\theta_j$ and $\vec{v}_{\theta_t - \theta_i}$ is the vector from initial parameters $\theta_i$ to target parameters $\theta_t$. This loss provides *task directedness*, preventing the agent from meandering through the task space by providing curriculum pressure to stay near the shortest distance between the starting point and the target tasks. Note that initial parameters $\theta_i$ are generally not the easiest task parameters $\theta_e$; $\theta_i$ is the value of the curriculum policy mean $\mu_\theta$ after the initial update and thus is the starting point of the learned curriculum. This loss is not calculated in the initial update, as $\theta_i$ is only determined at the end of the initial update.

## 4.2 Informed initialization of curriculum policy search

To improve the quality of curriculum policy search and further improve task directedness, an informed initialization is conducted to induce the curriculum policy search towards tasks with higher curriculum reward. Once a curriculum update is triggered (i.e., $R_S \le R_\mathcal{D}$), the mean used to initialize curriculum policy search becomes the current curriculum policy mean $\mu_\theta$ shifted towards the target tasks $\theta_t$ by $\Delta\mu_\theta$. This is done by the routine DIRECTTOWARDSTARGET, which rotates the vector $\Delta\mu_\theta$ via a Householder matrix (Householder, 1958) such that it points to $\theta_t$ from $\mu_\theta$.

## 4.3 Automatic determination of hyperparameters

To improve CURATE's utility as an automatic curriculum learning algorithm, the initial value of $(\mu_\theta, \Sigma_\theta)$ and selected hyperparameters (including $\lambda_\theta$ and $\lambda_d$) are automatically determined by heuristics according to the task space and given meta-hyperparameters. For more information, please refer to App. B.4.

## 5  Experimental results

In this section, we assess CURATE curricula and systematically evaluate CURATE against a variety of curriculum baselines across a diversity of domains. Specifically, we seek to answer the following five research questions. Q1: What are the characteristics of learned CURATE curricula? Q2: How does CURATE improve relative to prior curriculum methods with respect to target task performance? Q3: Can CURATE provide strong performance across all tasks, beyond just the target task? Q4: Can CURATE train broadly capable RL agents in large task spaces? Q5: What are the major important mechanisms for CURATE, and can we assess their importance through sensitivity and ablation experiments?

**Metrics**  We present our findings either in terms of learning curves or summary statistics that encapsulate major performance desiderata for performant curriculum learning. For Q2, our metrics include *time to threshold* (Narvekar et al., 2020) (in terms of training updates to reach a desired performance) and final target return. We also show final training return. For Q3, we assess time to threshold and average test performance over all tasks. For Q4, we primarily consider test performance over selected tasks of interest. Our metrics vary for Q5, depending on the components under investigation. Our analyses generally include calculations of

Figure 2: CURATE is evaluated on three experimental domains broadly grouped by task parameterization and actions: discrete (MiniGrid MultiRoom, Procgen Curriculum Suite) and continuous (BipedalWalker).

interquartile mean (IQM) with 95% confidence intervals (CI) using 50,000 bootstrap replications via Agarwal et al. (2021).

**Experimental domains**   Figure 2 provides an overview of the experimental domains. The domains have a rich diversity of task parameterization (discrete or continuous), task space dimensionality (1D to 8D), actions (discrete or continuous), and observations (state-based or image-based). Our experiments include *MiniGrid MultiRoom* (Chevalier-Boisvert et al., 2023), the 16 games of the *Procgen Curriculum Suite* (PCS) that is introduced in this work based on Cobbe et al. (2020), and *BipedalWalker* (Brockman et al., 2016), including the task space and terrain-specific challenges introduced in Parker-Holder et al. (2022a). More information for these domains can be found in App. C.

**Train and test procedure**   All methods use PPO (Schulman et al., 2017) with the Adam optimizer (Kingma & Ba, 2015) for training the control policy $\pi$. The optimizer runs continuously and is not reset during training. For experiments except BipedalWalker, the agent is evaluated on the target tasks $\mathcal{M}_{\theta_t}$ after every update of $\pi$. For BipedalWalker, the agent is evaluated every 100 $\pi$ updates, as we are not concerned with time to threshold in this domain. For CURATE, we include the teacher samples required for updating $\pi_c$ (except in the final environment evaluation for BipedalWalker, which is shown for the student only). These teacher samples include each round of curriculum policy search (when considering updates) and the steps the agent takes during an evaluation sample (when considering steps).

**Baselines**   We assess CURATE against a variety of curriculum baselines: non-learning baselines and learning-based baselines that learn implicit or structured curricula.

Our curriculum baselines without learning are *Domain Randomization* (DR) (Tobin et al., 2017), *Target* (NC), and, for MultiRoom only, *Hand Curriculum* (HC), DR represents a random curriculum. NC represents only training on the target tasks without a curriculum. HC is a hand-designed, structured curriculum that increments the task difficulty using an expert-chosen increment schedule when the agent reaches the task solved threshold. We only consider HC for MultiRoom, as the straightforward task space facilitates the hand-crafted creation of a performant curriculum, approximating what a "ground truth" curriculum would be. Because our other domains primarily consist of multidimensional task spaces, it is not clear what the best curriculum would be in these spaces without further analysis or specialized algorithms. Therefore, for PCS and BipedalWalker, we focus on DR and NC for our non-learning baselines.

Our UED and DCD baselines that learn implicit curricula are *Robust PLR* ($\text{PLR}^{\perp}$) (Jiang et al., 2021b) and *ACCEL* (Jiang et al., 2021a). $\text{PLR}^{\perp}$ focuses on student replay of levels, extending PLR (Jiang et al., 2021b) by only updating the agent on replayed levels. ACCEL (Jiang et al., 2021a) randomly mutates replayed levels and starts from the easiest set of tasks.

Lastly, for BipedalWalker only, we also evaluate *ALP-GMM* (Portelas et al., 2020), a learning-based curriculum algorithm based on absolute learning progress. We only evaluate ALP-GMM for Bipedalwalker as it is designed for environments with continuous task parameterizations.

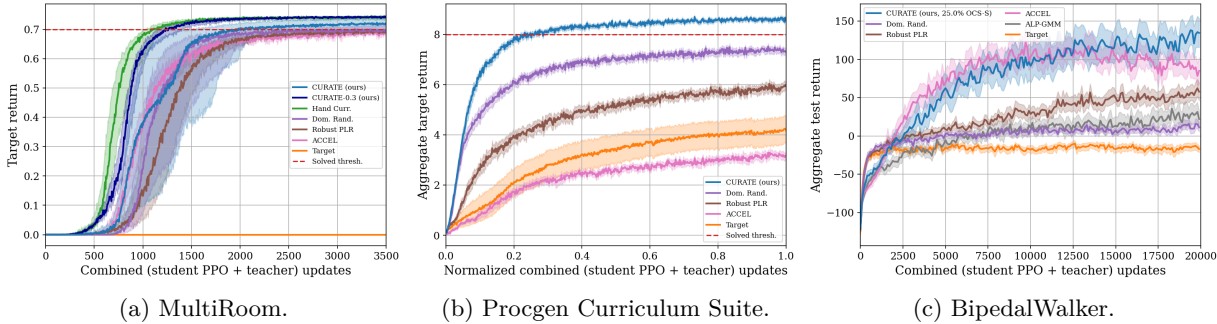

(a) MultiRoom.  (b) Procgen Curriculum Suite.  (c) BipedalWalker.

Figure 3: Major findings for CURATE's target task performance, shown as learning curves in terms of performance on the target tasks for (a) MultiRoom (10 trials); (b) PCS (aggregate over all 16 games, 6 trials per game); and (c) BipedalWalker (aggregate over 7 test environments for 10 trials). Curves are shown in terms of IQM with 95% confidence interval (Agarwal et al., 2021) and use interpolation. Curves also use subsampling every 5 updates for (a) MultiRoom and (b) PCS.

### 5.1 Q1: Visual analysis of CURATE curricula

What are the characteristics of learned CURATE curricula, and how do they contribute to CURATE's performance? To answer this question, we visualize the resulting CURATE curricula from experiments with MultiRoom and PCS. As shown in Fig. 4, we find that CURATE curricula 1) yield an approximately easiest-to-hardest progression without predetermined schedules or step sizes; 2) focus on a small, relevant distribution of tasks at a time; 3) remain task-directed in multidimensional task spaces; and 4) result in diverse starting task distributions and therefore resulting curricula. Together, these properties lead to curricula that efficiently progress the agent through the task space, so that it can focus training on the target task distribution. Furthermore, we find the properties of requiring neither informed initial starting distributions nor *a priori* schedules to distinguish CURATE from prior methods, leading to diverse, performant curricula that are specialized to the competence of the agent.

### 5.2 Q2: Improving target task performance using CURATE

Do CURATE curricula lead to beneficial performance on the target tasks compared to other approaches? For both MultiRoom and PCS, we find that CURATE outperforms naïve curricula baselines (DR, NC) and our implicit curricula baselines ($PLR^\perp$, ACCEL). We also find evidence to suggest that while prior methods optimize for training return, CURATE focuses on target task performance, and that greater training return does not necessarily lead to better performance on the target tasks.

We present our findings in Tab. 1 for MultiRoom and Tab. 2 for PCS, primarily for time-to-threshold for solving $\mathcal{M}_{\theta_t}$ and final $\mathcal{M}_{\theta_t}$ return.

#### 5.2.1 MultiRoom: Target task performance

For MultiRoom, Tab. 1 presents the main results for this domain in terms of summary statistics, and Fig. 3a shows the target performance learning curve. For more figures, please see App. F.

Generally, we find that CURATE offers the strongest performance in time to threshold in updates (TTU) to solve the target task for all approaches without expert knowledge (HC). Moreover, we found that using $R_S = 0.3$ for CURATE (CURATE-0.3) significantly boosts performance as compared to $R_S = 0.7$, nearly matching the performance of HC. We find that a lower threshold facilitates greater learning progress in the presence of a distribution shift: between tasks with 1 room and tasks with 2 rooms, the agent must now learn to open doors. This presents a "curriculum bottleneck" that leads to greater inefficiency with using a higher solved threshold of 0.7. For more analysis of the selection of $R_S$, please see App. E.1.

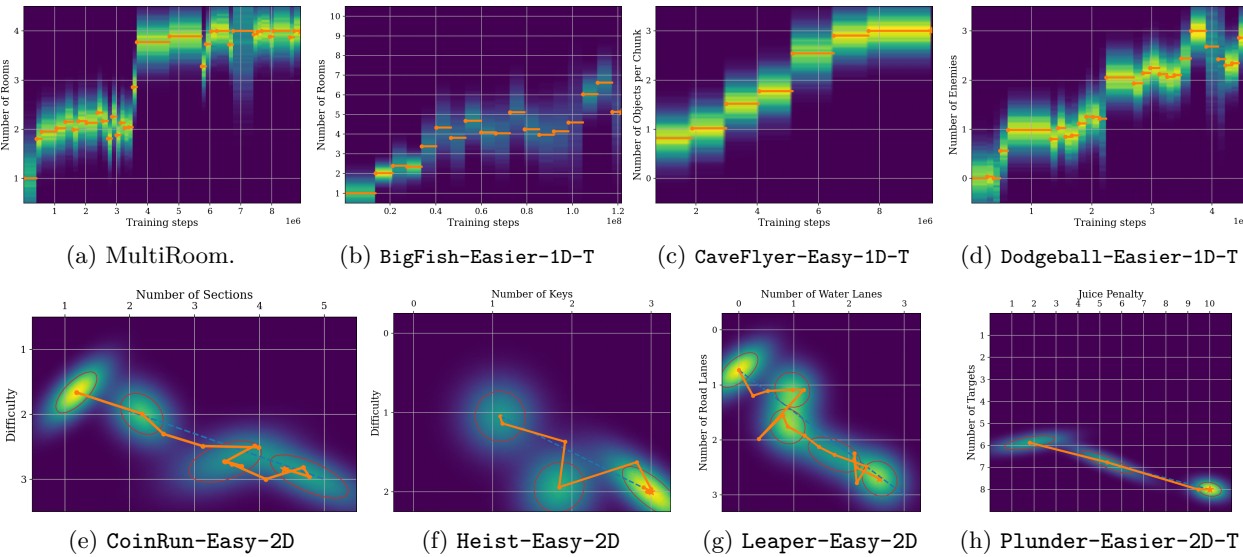

Figure 4: A selection of learned CURATE curricula for various 1D (top row) and 2D (bottom row) domains. Certain properties are evident, such as 1) an approximately easiest-to-hardest progression, which is achieved without *a priori* schedules; 2) maintaining relatively smaller training distributions at a given time, which focuses training to boost learning progress; 3) task directedness (for multidimensional task spaces), which keeps the curricula steered towards the target tasks; and 4) diversity in starting task distribution, which allows CURATE to bypass large sections of the task space that the agent can already solve. (We note that the starting task distributions shown here are after the initial CURATE curriculum update, which does not require an informed initialization.)

Table 1: Target task performance results for MultiRoom using 5,000 combined updates (student PPO and teacher, if used) for 10 trials. Summary statistics are shown in terms of success rate (if the target tasks $\mathcal{M}_{\theta_t}$ were solved), time to threshold in updates until $\mathcal{M}_{\theta_t}$ is solved or maximum updates reached (TTU), final training return (F. Train), and final target return (F. Target). Trials that do not solve the task still count towards summary statistics. TTU is $\times 10^3$. TTU, F. Train, and F. Target are shown in terms of IQM with 95% CI (Agarwal et al., 2021). The best approach for each metric is **bolded**, the second best approach is underlined, and an asterisk (*) denotes that the approach falls within the best approach's CI.

| Method | Success Rate | TTU ↓ | TTU CI | F. Train ↑ | F. Train CI | F. Target ↑ | F. Target CI |
|---|---|---|---|---|---|---|---|
| CURATE (ours) | **100.0%** (10) | 1.539 | (1.348, 1.979) | 0.742 | (0.721, 0.761) | 0.720 | (0.699, 0.741) |
| CURATE-0.3 (ours) | **100.0%** (10) | 1.195 | (1.113, 1.312) | 0.757 | (0.750, 0.766) | 0.741* | (0.739, 0.745) |
| Hand Curr. | **100.0%** (10) | **1.052** | (0.981, 1.156) | 0.745 | (0.737, 0.752) | **0.742** | (0.733, 0.745) |
| Domain Rand. | **100.0%** (10) | 1.814 | (1.664, 2.070) | **0.899** | (0.887, 0.915) | 0.700 | (0.696, 0.722) |
| Robust PLR | **100.0%** (10) | 2.177 | (1.997, 2.400) | 0.851 | (0.820, 0.873) | 0.697 | (0.691, 0.700) |
| ACCEL | **100.0%** (10) | 2.313 | (2.158, 2.509) | 0.849 | (0.804, 0.864) | 0.691 | (0.682, 0.697) |
| Target | 0.0% (0) | 5.000 | (5.000, 5.000) | 0.000 | (0.000, 0.000) | 0.000 | (0.000, 0.000) |

Although HC is more performant in terms of TTU, we find that when only considering student updates, CURATE-0.3 matches HC (IQM: 1.092, CI: (1.017, 1.196)), falling within the 95% confidence interval of HC. Here, we attribute the difference mostly to the fact that the curriculum is specified *a priori* for HC, whereas CURATE must learn the curriculum. Furthermore, we find that CURATE-0.3 offers nearly identical performance to HC in terms of final target return, In this light, we find CURATE's performance promising against the pseudo-ground truth HC in this domain.

Table 2: Aggregate target task performance results for the Procgen Curriculum Suite using at least $200 \times 10^6$ training steps for six trials over all 16 games. Maximum combined updates (student PPO and teacher, if used) are game-specific and normalized across games. Summary statistics are shown in terms of success rate (if the target tasks $\mathcal{M}_{\theta_t}$ were solved), time to threshold in normalized updates until $\mathcal{M}_{\theta_t}$ is solved or maximum updates reached (TTUN), final training return (F. Train), and final target return (F. Target). Trials that do not solve the task still count towards summary statistics. TTUN, F. Train, and F. Target are shown in terms of IQM with 95% CI (Agarwal et al., 2021). The best approach for each metric is **bolded**, the second best approach is underlined, and an asterisk ($^*$) denotes that the approach falls within the best approach's CI.

| Method | Success Rate | TTUN ↓ | TTUN CI | F. Train ↑ | F. Train CI | F. Target ↑ | F. Target CI |
|---|---|---|---|---|---|---|---|
| CURATE (ours) | **100.000%** (96) | **13.512%** | (12.772%, 14.408%) | 8.845 | (8.753, 8.928) | **8.634** | (8.540, 8.714) |
| Domain Rand. | 70.833% (68) | 47.654% | (44.014%, 51.949%) | **9.618** | (9.571, 9.657) | 7.297 | (7.135, 7.446) |
| Robust PLR | 34.375% (33) | 98.311% | (96.229%, 99.534%) | 5.206 | (4.550, 5.874) | 5.951 | (5.781, 6.117) |
| ACCEL | 11.458% (11) | 100.000% | (100.000%, 100.000%) | 6.103 | (5.356, 6.815) | 3.195 | (3.065, 3.325) |
| Target | 48.958% (47) | 78.177% | (71.548%, 84.785%) | 3.602 | (3.042, 4.165) | 4.176 | (3.613, 4.709) |

### 5.2.2 Procgen Curriculum Suite: Target task performance

The main aggregate results for PCS are shown as summary statistics in Tab. 2. The aggregate target performance learning curve is shown in Fig. 3b. Further results are available in App. G, including per-game target learning curves (Fig. 19) and summary statistics (Tab. 10 through Tab. 25).

The PCS games present a greater learning challenge than MultiRoom, both in terms of a greater challenge of image-based control, but larger and more diverse task spaces. In fact, some games significantly challenge the baseline methods, where they are unable to solve any of the six trials. In particular, `Chaser-Easier-2D-T` and `FruitBot-Easy-3D-T` are unsolvable for any trials for the naïve curriculum baselines of DR and NC. Additionally, `Plunder-Easier-2D-T` is unsolvable for DR and has a 50% success rate for NC.

Despite these challenges, we find that CURATE is the only method to offer broad performance across nearly all games in this domain. Across 16 games, CURATE exceeds all baselines in TTUN for 12 games, ranks second in TTUN for 3 games, and holds a strong performance lead in aggregate target performance. Notably, in 10 of these 12 games, CURATE's TTUN exceeds the 95% CI for all approaches. This includes the difficult `Chaser-Easier-2D-T`, `FruitBot-Easy-3D-T`, and `Plunder-Easier-2D-T` games, where CURATE's TTUN is 20.466%, 17.239%, and 33.788% of the second-best approach's TTUN, respectively.

Although the aggregate performance of CURATE outperforms all baselines, we also find evidence that DR or NC offer strong performance relative to CURATE in a small subset of games (4: `CaveFlyer-Easy-1D-T`, `Jumper-Easy-1D`, `Maze-Easy-1D`, `StarPilot-Easier-4D-T`). For example, in `CaveFlyer-Easy-1D-T`, we find that although CURATE performs well (IQM: 5.293%, CI: (4.769%, 5.851%)), NC is best (IQM: 4.425%, CI: (3.507%, 5.015%)), followed by DR (IQM: 5.129%, CI: (4.302%, 5.629%)). Generally, we hypothesize that in these games, the target tasks are relatively easier: rewards can be more consistently received, leading to stronger performance by DR or NC. The benefits of CURATE curricula to progress the agent from easy to hard tasks via transfer learning are less effective here, as naïve strategies are competitive if not slightly more performant. A notable exception is `StarPilot-Easier-4D-T`, where NC has outstanding performance (IQM: 5.302%, CI: (5.138%, 6.293%)) over CURATE (IQM: 31.768%, CI: (27.073%, 38.651%)). This game is unique insofar that for finish line times under a certain threshold, no enemies spawn, and thus the effect of the other environment parameters (related to enemies) do not affect the generation of the level. Thus, this task space has a challenging optimization landscape, and sampling tasks further in the task space may be beneficial. Put together, these findings suggest that the desiderata for an automatic curriculum learning algorithm also include understanding the tradeoff between using a curriculum to progress the agent through the task space versus training directly in the target task distribution, which may be preferable in some domains.

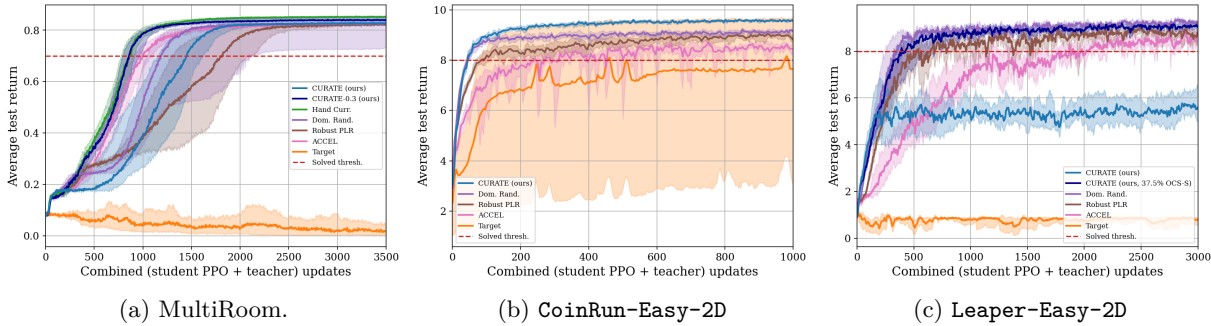

Figure 5: Major findings for CURATE's average task performance, shown as learning curves in terms of average task performance for (a) MultiRoom (10 trials); (b) PCS CoinRun (6 trials); and (c) PCS Leaper (6 trials). Curves are shown in terms of IQM with 95% confidence interval (Agarwal et al., 2021) and use interpolation and subsampling every 5 updates.

### 5.3 Q3: Improving average task performance using CURATE

Although CURATE is designed for training performant agents in the target tasks, we find that under certain conditions, CURATE curricula also offer competitive performance in average performance across all tasks. This property improves CURATE's utility as an automatic curriculum learning algorithm, as a curriculum may be desired to improve overall performance (not necessarily on the hardest tasks). Our findings are shown in average task learning curves (Fig. 5). Further results, including summary statistics and heatmaps, are available in App. D.

We demonstrate that CURATE yields strong average performance for MultiRoom (Fig. 5a, Tab. 6) and CoinRun (Fig. 5b, Tab. 7) normally, and, when paired with an off-curriculum sampling strategy, for Leaper (Fig. 5c, Tab. 8). Specifically for Leaper, we sample tasks such that 37.5% of the parallel environments uniformly sample $\theta \prec \mu_\theta$ (OCS-S), with the remainder sampling from $\pi_c$ as usual. Our result in this domain confirms CURATE's compatibility for off-curriculum sampling strategies. Importantly, we find that CURATE's average task performance in these domains outperforms PLR$^\perp$ and ACCEL, a notable result given that UED/DCD algorithms are primarily intended for task generalization.

### 5.4 Q4: Broadening CURATE's capabilities to larger task spaces

Our evidence suggests that CURATE is highly capable for automatic curriculum learning for difficult, target tasks, while also showing competitive performance for average task performance. We also find compelling evidence in BipedalWalker that CURATE curricula can train generally capable agents that succeed in large task distributions with continuous environment parameterizations. This domain also differs in that rewards are not sparse. Although BipedalWalker has dense rewards, initially, all returns are uninformative (around a return of -150), so a curriculum is still effective.

For BipedalWalker, we primarily assess the ability to generalize to seven test tasks of interest within a vast, 8-dimensional task space. These environments are `BipedalWalker`; `BipedalWalkerHardcore`; `BipedalWalkerMax`, which is our target environment $\mathcal{M}_{\theta_t}$ with maximum terrain difficulty; and the four terrain challenges introduced by Parker-Holder et al. (2022a): `BipedalWalkerStairs`, `BipedalWalkerPitGap`, `BipedalWalkerStump`, and `BipedalWalkerRoughness`. For these experiments, CURATE uses an 25% OCS-S off-curriculum policy to boost generalization and mitigate forgetting.

The test performance learning curve is shown in Fig. 3c (aggregate over all environments). Table 3 shows the aggregate results for a more comprehensive evaluation conducted across 10 trials and 100 episodes per environment per trial, conducted at 20,000 student updates (similar to the evaluation conducted in Parker-Holder et al. (2022a), except for seven environments and 20,000 rather than 30,000 student updates due to limitations in running experiments to the longer duration). More results, such as per-environment learning curves (Fig. 23) and summary statistics (Tab. 26), are available in App. H.

Table 3: Aggregate test performance for the BipedalWalker domain across seven environments of interest (F. Agg. Test). Tests were conducted at 20,000 student updates across 10 trials and 100 episodes per environment per trial. The environments consist of two classic environments (`BipedalWalker`, `BipedalWalkerHardcore`); `BipedalWalkerMax`, the target environment $\mathcal{M}_{\theta_t}$ with maximum difficulty; and the four terrain challenges introduced in Parker-Holder et al. (2022a) (`BipedalWalkerStairs`, `BipedalWalkerPitGap`, `BipedalWalkerStump`, `BipedalWalkerRoughness`). F. Agg. Test is shown in terms of IQM with 95% CI (Agarwal et al., 2021). 1st/2nd represents how many of the seven environments the method performed best/second best in terms of IQM. For 2nd, a parenthesis indicates how many were within the best method's CI. The best approach for each metric is **bolded**, the second best approach is underlined, and an asterisk (*) denotes that the approach falls within the best approach's CI.

| Method | 1st | 2nd | F. Agg. Test ↑ | F. Agg. Test CI |
|--------|-----|-----|----------------|-----------------|
| CURATE, 25% OCS-S (ours) | **4** | 2 (1) | **132.177** | (127.392, 136.960) |
| Domain Rand. | 0 | 1 | 10.340 | (9.493, 11.210) |
| Robust PLR | 0 | 0 | 42.818 | (40.588, 45.158) |
| ACCEL | 1 | **3** | 75.051 | (71.187, 78.942) |
| ALP-GMM | 1 | 1 (1) | 16.573 | (14.888, 18.310) |
| Target | 1 | 0 | -9.691 | (-10.554, -8.863) |

Overall, we find that at the end of training, CURATE outperforms all baselines, including ACCEL, holding a strong edge in terms of aggregate test performance. Although ACCEL performs better initially, CURATE continues to improve throughout training, overtaking ACCEL around 12,500 updates. Importantly, we find that CURATE is generally well-performing across most environments, performing best in 4 environments and second best in 2. The only environment in which CURATE performs poorly is `BipedalWalkerMax`. This is an extremely difficult domain that challenges all methods, where no method achieves over zero return.

### 5.5 Q5: Understanding CURATE components through sensitivity and ablation experiments

Following our evidence of CURATE's strong performance, we conduct sensitivity and ablation experiments across major components to better understand their importance. These experiments are presented in App. E, but we describe the key findings here.

App. E.1: **Selection of task solved threshold $R_S$ (MultiRoom).** We find that the relative ordering of methods are generally, but not always, consistent for different values of $R_S$. Lowering $R_S$ for intermediate tasks boosts CURATE's performance and avoids high-variance results at larger values of $R_S$. Lowering $R_S$ effectively resolves the "curriculum bottleneck" that otherwise exists, allowing faster progress through the curriculum. A value of $R_S = 0.3$ was found to lead to the best overall performance for CURATE.

App. E.2: **Selection of hyperparameters controlling the rigor of evaluations for the curriculum update (MultiRoom).** We perform sensitivity experiments for two CURATE hyperparameters: number of episodes per evaluation sample and minimum number of student PPO updates before a curriculum update can be run. A greater number of episodes per evaluation sample improves the informativeness of teacher samples at the tradeoff of greater teacher sample costs. Performing more frequent curriculum updates keeps the curriculum more synchronized to the competence of the agent, but this also increases teacher sample costs. We find that moderate values of both hyperparameters balance these competing mechanisms.

App. E.3: **Assessing the importance of $\mathcal{L}_{diff}$ (BipedalWalker).** We find that removing $\mathcal{L}_{diff}$ significantly reduces performance for most of training, which we hypothesize is due to the resulting initial training distribution being much broader, slowing down learning as compared to the narrower distribution otherwise.

App. E.4: **Analyzing the importance of $\mathcal{L}_{dist}$ and the informed initialization (`Leaper-Hard-2D`).** We find that both mechanisms for inducing task directedness — $\mathcal{L}_{dist}$ and the informed initialization through $\Delta\mu_\theta$ — are vital for CURATE's performance. Performance degrades when removing either component, yielding curricula that tend to meander throughout the task space and lack a mechanism for pushing the learning frontier forward. The effect of each component is likely task-dependent, but we find that in this domain, the informed initialization has a stronger effect.

# 6 Conclusion

We present CURATE, an automatic curriculum learning approach for training a model-free, on-policy reinforcement learning agent for structured task spaces with monotonic difficulty. CURATE navigates a curriculum through policy search in the task space to establish the best task distribution that matches the agent's current competence. In so doing, CURATE's "exploration by exploitation" approach addresses fundamental exploration challenges through curriculum learning. Moreover, CURATE is effective without requiring optimal initializations or predefined schedules as prior methods require. Our results show that CURATE either matches or outperforms prior curriculum methods at training agents that are not only highly performant for difficult tasks, but in certain domains, are broadly capable for a wide range of tasks. Through this work, we hope that CURATE and the Procgen Curriculum Suite spark greater interest towards realizing a general automatic curriculum learning algorithm and the profound impact it would entail for reinforcement learning.

## 6.1 Limitations

Although CURATE offers promising performance for automatically learning performant curricula, it is important to note that CURATE is designed for task spaces of monotonic difficulty along the axes of the environment parameters $\Theta$ (e.g., a DOUPOMDP). Therefore, CURATE requires the task space to be 1) defined, 2) structured in difficulty order along the task space axes, and 3) available to use for learning. We also assume that tasks within the task space are generally feasible and possible to learn in principle (though, in practice, they can be quite challenging).

## 6.2 Future work

Several avenues exist for future work that builds from the foundation set by CURATE through either relaxing assumptions or extending CURATE design choices. We outline two avenues below.

**Latent space learning.** We believe that learning a latent task space with the required DOUPOMDP properties would not only unlock the ability to use CURATE in a broader class of task spaces, but such efforts could also advance the curriculum learning field as a whole. These efforts towards "latent task space curriculum learning" would offer new avenues for curriculum learning. By transforming complex task spaces (e.g., containing infeasible tasks, or spaces without a predetermined difficulty ordering) into smooth latent spaces with favorable representational properties, curriculum learning algorithms can leverage properties of the space to make informed curriculum decisions (contrasting with methods such as $PLR^{\perp}$ or ACCEL that only make decisions on a per-task basis). For example, the property of a DOUPOMDP being difficulty-ordered allows the competence learning objective of CURATE to push the learning frontier in a direction towards harder tasks. In principle, this mechanism can work for latent task spaces if the latent representation has such a property. Towards this end, we believe the introduction of the Procgen Curriculum Suite and its predefined task spaces would form a useful basis of comparison and ground truth for future work in attempting to learn such latent task spaces with DOUPOMDP properties.

**Simultaneously training the student and teacher.** Our implementation of CURATE uses a separate curriculum update phase, which occurs at periodic intervals during training. The runtime cost of this phase can be reduced with asynchronous processing, with the degree of reduction depending on the parallelization of control policy $\pi$ evaluations that are used as teacher samples for learning the curriculum policy $\pi_c$. To further improve the runtime for CURATE, an extension could be explored that updates the curriculum policy $\pi_c$ not through a separate update phase, but simultaneously with the control policy $\pi$. Although this

would eliminate the separate curriculum update phase, this extension may be less stable to train and require additional exploration mechanisms.

**Broader Impact Statement**

CURATE, our approach for automatic curriculum learning, has broad applications for real-world reinforcement learning agents, such as robots, embodied agents, and agentic artificial intelligence systems. Although we do not study these applications in this work, we recognize that the broader impact and ethical consequences of our work are likely application-specific. Therefore, we recommend implementing ethical protocols and safety guardrails when deploying real-world agents trained using CURATE to address, mitigate, or resolve any negative societal impact that may arise in the application of our work.

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

## A  Overview of CURATE

Figure 6 illustrates the intuition behind CURATE.

## B  CURATE algorithms

Algorithm 2 (CURATE) and Sec. B.1 describe the training procedure used to train RL agents using CURATE in this work. Algorithm 3 (UPDATECURRICULUM) and Sec. B.2 describe the policy search procedure that CURATE uses to learn the curriculum policy $\pi_c$ during training.

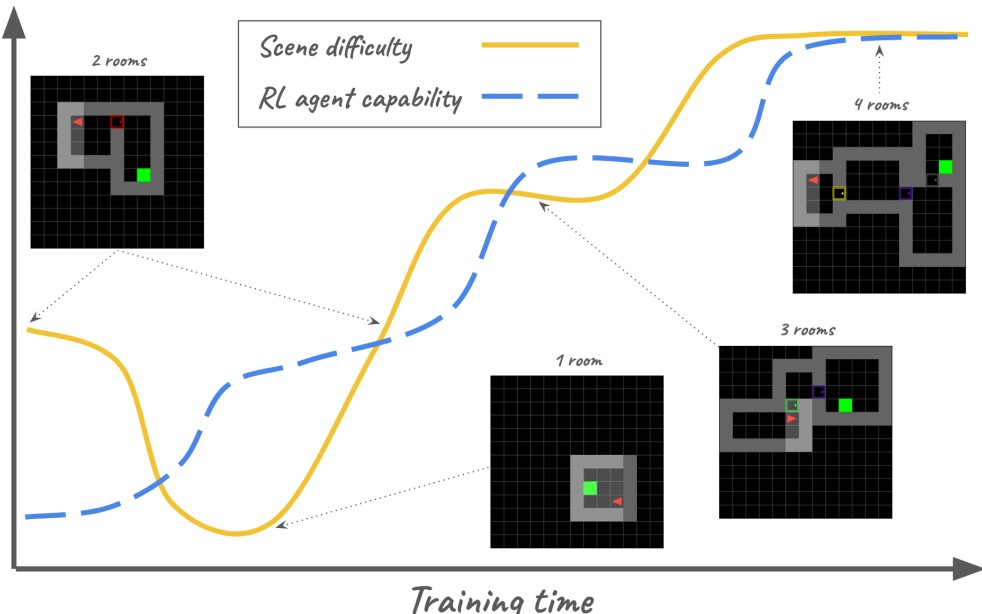

Figure 6: The CURATE algorithm automatically learns a curriculum for training an RL agent to complete a target task distribution that is initially too difficult for the agent. CURATE sequences the RL agent's training data by altering the difficulty of the training task distribution. The RL agent's current capability, or competence, is a measure of its performance in relatively more difficult tasks. In this visualization, the tasks offered by CURATE are initially too difficult, leading to a simplification of tasks. Once the RL agent begins solving these simple tasks, CURATE dynamically adjusts the training data accordingly to offer harder tasks. Finally, the agent solves the target task distribution at the end, indicating that training can conclude. Scenes are from the MiniGrid MultiRoom domain (Sec. 5).

---

**Algorithm 2:** CURATE: CURRICULUM AGENT FOR TARGETED EXPLORATION (Early Stopping)

---

**Input:** target task $\mathcal{M}_{\theta_t}$, task solved threshold $R_S$, maximum number of student updates $u_{max}$, number of parallelized workers $N_v$, target episodes per evaluation $N_{te}$, minimum updates between curriculum updates $N_{min}$, maximum updates between curriculum updates $N_{max}$

**Initialize:** training indicator $train \leftarrow$ True, target task solved indicator $converged \leftarrow$ False, number of student updates $u \leftarrow 0$, control policy $\pi \leftarrow$ INITIALIZERANDOMPOLICY(), target task $\theta_t \leftarrow \max(\Theta)$, curriculum policy $\mu_{\theta_0}, \Sigma_{\theta_0} \leftarrow$ INITIALIZERANDOMCURRICULUMPOLICY($\Theta$), $\Delta\mu_\theta, \Sigma_{\theta_u}, \lambda_\theta, \lambda_d \leftarrow$ INITIALIZEHYPERPARAMETERS($\Theta$), previous curriculum update $u_{prev} \leftarrow 0$

---

```
// Initial curriculum policy update
```
$\mu_\theta, \Sigma_\theta \leftarrow$ UPDATECURRICULUM($\pi, \mu_{\theta_0}, \Sigma_{\theta_0}, \lambda_\theta, use\_diff \leftarrow$ True)
$\theta_i \leftarrow \mu_\theta$
**while** $train$ **do**
    `// Sample tasks from the curriculum`
    $\mathcal{M}_{\boldsymbol{\theta_c}} \leftarrow \emptyset$
    **for** $i = 1$ **to** $N_v$ **do**
        $\theta_i \sim \pi_c(\theta; \mu_\theta, \Sigma_\theta)$
        $\mathcal{M}_{\theta_i} \leftarrow$ TASKGENERATOR($\theta_i$)
        $\mathcal{M}_{\boldsymbol{\theta_c}} \stackrel{+}{\leftarrow} \mathcal{M}_{\theta_i}$
    **end**
    `// Collect experience`
    $\mathcal{D}, R_\mathcal{D} \leftarrow$ ROLLOUTAGENTONPARALLELTASKS($\pi, \mathcal{M}_{\boldsymbol{\theta_c}}$)
    `// Update policy`
    $\pi \leftarrow$ UPDATEAGENT($\pi, \mathcal{D}$)
    $u \leftarrow u + 1$
    `// Update curriculum policy`
    **if** $R_S \leq R_\mathcal{D} \wedge N_{min} \leq (u - u_{prev})$ **then**
        $\mu_{\theta_u} \leftarrow$ DIRECTTOWARDSTARGET($\mu_\theta, \theta_t, \Delta\mu_\theta$)
        $\mu_\theta, \Sigma_\theta \leftarrow$ UPDATECURRICULUM($\pi, \mu_{\theta_u}, \Sigma_{\theta_u}, \theta_i, \lambda_d, use\_dist \leftarrow$ True)
        $u_{prev} \leftarrow u$
    **else if** $N_{max} \leq (u - u_{prev})$ **then**
        $\mu_\theta, \Sigma_\theta \leftarrow$ UPDATECURRICULUM($\pi, \mu_{\theta_u}, \Sigma_{\theta_u}, \theta_i, \lambda_d, use\_dist \leftarrow$ True)
        $u_{prev} \leftarrow u$
    `// Evaluate agent on target task`
    $R_t \leftarrow$ EVALUATEAGENT($\pi, \mathcal{M}_{\theta_t}, N_{te}$)
    `// Determine whether to continue training`
    **if** $R_S \leq R_t$ **then**
        $train \leftarrow$ False
        $converged \leftarrow$ True
    **if** $u_{max} \leq u$ **then**
        $train \leftarrow$ False
**end**

---

**Result:** control policy $\pi$, target task solved indicator $converged$, number of student updates $u$

---

## B.1 Training RL policies with CURATE

This section summarizes the full procedure for training the RL agent as described in Alg. 2, which expands upon Alg. 1 by showing additional terms and using early stopping. This training procedure is designed to close the RL training loop around the target task distribution $\mathcal{M}_{\theta_t}$, such that RL training ends with only the minimum number of updates needed to solve $\mathcal{M}_{\theta_t}$. First, the control policy $\pi$ is initialized randomly. Then, the curriculum policy $\pi_c$ is initialized by the Gaussian distribution that approximates a uniform distribution over the task space. Thereafter, the curriculum policy is updated prior to training with the (initially random) control policy $\pi$ via UPDATECURRICULUM. For each iteration in the training loop, tasks are sampled from the curriculum policy $\pi_c$ by first sampling environment parameters $\theta_i$, which are in turn transformed into tasks. An off-curriculum sampling strategy may also be used. Then, a trajectory dataset $\mathcal{D}$ is generated with mean training reward $R_{\mathcal{D}}$ from rollouts of $\pi$ in the sampled tasks. Next, $\pi$ is updated by an on-policy reinforcement learning algorithm (e.g., PPO (Schulman et al., 2017)) by performing gradient updates of policy parameters using the dataset $\mathcal{D}$. Following the policy update, the curriculum policy $\pi_c$ is updated via UPDATECURRICULUM if the training reward $R_{\mathcal{D}}$ meets or exceeds the task solved threshold $R_S$ and at least $N_{min}$ updates have occurred. This trigger indicates that the agent has mastered proficiency in its current training distribution and is ready for more challenging tasks. Curriculum learning can also be triggered if a maximum number of timesteps since the last curriculum update $N_{max}$ has been reached. This prevents training stagnation if the current tasks are too difficult for the agent. Finally, the agent is evaluated on the target task distribution $\mathcal{M}_{\theta_t}$ to obtain a task evaluation reward $R_t$. We typically conduct stochastic, rather than deterministic, policy evaluation. If the agent solves the target task ($R_S \leq R_t$), then training concludes successfully. Otherwise, training continues while the number of maximum training updates has not been reached.

## B.2 Updating the CURATE curriculum policy

UPDATECURRICULUM conducts evaluations to assess the current proficiency of the agent in a sample-efficient manner without exhaustive search of the task space. First, the initial parameter distribution for the curriculum policy $\pi_c$ is provided as $(\mu_\theta, \Sigma_\theta)$. Then, for each of $N_r$ rounds, the agent draws $N_s$ parameter samples from the curriculum policy parameter distribution $(\mu_\theta, \Sigma_\theta)$. For each parameter sample $\theta_j$, the corresponding task $\mathcal{M}_{\theta_j}$ is generated, and the agent is evaluated on this task to yield reward $R_j$. However, this reward is not used directly for the curriculum learning reward $\nu_j$. Instead, it is assessed whether it meets or exceeds the threshold $R_S$, i.e., the task is solved, and it is not the target task $\theta_t$. If so, the agent receives zero curriculum reward for this task, as the agent has mastered this task. Otherwise, the curriculum reward is first assessed as $R_j/R_S$. This reward signal $\mathcal{R}_{comp}$ induces the agent towards the easiest (i.e., highest return) tasks that have not yet been solved based on the agent's current competence. In this way, $\mathcal{R}_{comp}$ is a mathematical characterization of the objective used by children when self-generating curricula (Dahmani et al., 2025). Thereafter, a loss is calculated to update $\nu_j$: $\mathcal{L}_{diff}$ for the initial update and $\mathcal{L}_{dist}$ otherwise. Then, the parameter samples $\theta_j$ and curriculum reward $\nu_j$ are appended to buffers. These buffers are used by Relative Entropy Policy Search (REPS) (Peters et al., 2010) to yield an updated Gaussian distribution that maximizes the curriculum reward, subject to an information loss bound based on Kullback-Leibler divergence (Kullback & Leibler, 1951). Lastly, the continuous curriculum parameters $(\mu_\theta, \Sigma_\theta)$ are discretized to yield the updated curriculum policy $\pi_c$. This process repeats iteratively $N_r$ times before returning $\pi_c$ at the conclusion of the curriculum update. In calculating the number of teacher samples, each curriculum update is thus counted $N_r$ times.

## B.3 CURATE and the scale of the task space

In our implementation of CURATE, we assume that the dimensions of the task space are unnormalized. We found that this design choice was sufficient for demonstrating CURATE's performance for a diversity of task spaces across three domains (MiniGrid, Procgen, BipedalWalker). Furthermore, some terms of CURATE, such as the regularization loss $\mathcal{L}_{diff}$ and the orthogonal distance loss $\mathcal{L}_{dist}$, implicitly provide task space normalization through their automatic setting of hyperparameters based on properties of the task space (c.f., App. B.4). However, the choice of task space scaling may lead to differences in the curriculum update,

---

**Algorithm 3:** UPDATECURRICULUM: Curriculum update for CURATE

---

**Input:** curriculum policy $\pi_c$, initial curriculum policy mean $\mu_{\theta_0}$, initial curriculum policy covariance $\Sigma_{\theta_0}$, control policy $\pi$, task solved threshold $R_S$, easiest environment parameters $\theta_e$, target environment parameters $\theta_t$, regularization hyperparameter $\lambda_\theta$, orthogonal distance loss hyperparameter $\lambda_d$, number of rounds $N_r$, samples per round $N_s$, episodes per evaluation sample $N_{se}$, relative entropy bound $\epsilon$, minimum temperature $\eta$, enable regularization loss $use\_diff \leftarrow \texttt{False}$, enable orthogonal distance loss $use\_dist \leftarrow \texttt{False}$, initial environment parameters $\theta_i \leftarrow \texttt{None}$

**Initialize:** $\mu_\theta \leftarrow \mu_{\theta_0}$, $\Sigma_\theta \leftarrow \Sigma_{\theta_0}$

---

**for** $i = 1$ **to** $N_r$ **do**
    // Reset buffers
    $\boldsymbol{\theta_{eval}} \leftarrow \emptyset$
    $\boldsymbol{\nu_{eval}} \leftarrow \emptyset$
    **for** $j = 1$ **to** $N_s$ **do**
        // Sample task
        $\theta_j \sim \mathcal{N}(\mu_\theta, \Sigma_\theta)$
        $\mathcal{M}_{\theta_j} \leftarrow \text{TASKGENERATOR}(\theta_j)$
        // Evaluate agent on sampled task
        $R_j \leftarrow \text{EVALUATEAGENT}(\pi, \mathcal{M}_{\theta_j}, N_{se})$
        // Calculate curriculum reward based on competence
        **if** $R_j < R_S \vee \theta_j = \theta_t$ **then**
            $\nu_j \leftarrow R_j/R_S$
        **else**
            $\nu_j \leftarrow 0$
        // Calculate regularization loss, if applicable
        **if** $use\_diff$ **then**
            $\vec{v}_{\theta_j - \theta_e} \leftarrow \theta_j - \theta_e$
            $\nu_j \leftarrow \nu_j - \lambda_\theta ||\vec{v}_{\theta_j - \theta_e}||_2$
        // Calculate orthogonal distance loss, if applicable
        **if** $use\_dist$ **then**
            $\vec{v}_{\theta_j - \theta_i} \leftarrow \theta_j - \theta_i$
            $\vec{v}_{\theta_t - \theta_i} \leftarrow \theta_t - \theta_i$
            $\nu_j \leftarrow \nu_j - \lambda_d ||\vec{v}_{\theta_j - \theta_i} \perp \vec{v}_{\theta_t - \theta_i}||_2$
        // Append to buffers
        $\boldsymbol{\theta_{eval}} \xleftarrow{+} \theta_j$
        $\boldsymbol{\nu_{eval}} \xleftarrow{+} \nu_j$
    **end**
    // Run REPS and update curriculum policy
    $\mu_\theta, \Sigma_\theta \leftarrow \text{REPSUPDATE}(\boldsymbol{\theta_{eval}}, \boldsymbol{\nu_{eval}}, \epsilon, \eta)$
    $\pi_c \leftarrow \text{DISCRETIZEGAUSSIAN}(\mu_\theta, \Sigma_\theta)$
**end**

---

**Result:** updated curriculum policy $\pi_c$, updated curriculum policy mean $\mu_\theta$, updated curriculum policy covariance $\Sigma_\theta$

---

e.g., our version of REPS does not normalize the environment parameter values. Full normalization of the task space and environment parameter values may lead to improved performance in domains where some axes of the task space have a very large scale.

### B.4 Automatic determination of CURATE hyperparameters

To strengthen CURATE's utility as an automatic curriculum learning algorithm, six hyperparameters are automatically determined based on heuristics of the task space $\Theta$, easiest tasks $\theta_e = \min(\Theta)$, and hardest tasks $\theta_t = \max(\Theta)$. Some terms are controlled by meta-hyperparameters, which are used for the specific calculation of the hyperparameters based on $\Theta$. Because the best hyperparameters for any domain will likely be domain-specific, our goal for providing heuristically chosen hyperparameters is to provide an automatic, satisficing solution for most domains.

1. The method INITIALIZERANDOMCURRICULUMPOLICY takes as input $\Theta$ and returns $\mu_\theta \leftarrow (\theta_e + \theta_t)/2$ as the center of the task space and $\Sigma_\theta$ as the Gaussian distribution that best approximates a uniform distribution over $\Theta$ via an optimization procedure. The resulting initial $(\mu_{\theta_0}, \Sigma_{\theta_0})$ is then used for the initial CURATE curriculum update. In this way, the INITIALIZERANDOMCURRICULUMPOLICY method provides the initial CURATE curriculum with an approximation of a uniform distribution, accounting for the fact that the best initial training distribution could, in principle, be located anywhere within the task space.

2. The informed initialization $\Delta\mu_\theta$ hyperparameter is calculated as $\Delta\mu_\theta \leftarrow (\theta_t - \theta_e)/\Lambda_{\Delta\mu_\theta}$. This hyperparameter provides a slight bias towards progression after the training tasks are solved. We explore two choices for the meta-hyperparameter $\Lambda_{\Delta\mu_\theta}$, either 10 (coarse, MultiRoom and PCS) or 20 (fine, BipedalWalker). The value of the meta-hyperparameter $\Lambda_{\Delta\mu_\theta}$ is based on a heuristic that is intended to provide a small (e.g., 10% of the task space) initialization advancement to ensure a high-quality training distribution. Some domains, such as BipedalWalker, have very large task spaces, where a 10% advancement moves the training distribution too far in advance of the competence of the agent. Thus, a smaller advancement (e.g., 5%) was found to be advantageous.

3. The curriculum covariance for update $\Sigma_{\theta_u}$ hyperparameter controls how broad the initial distribution is at the beginning of the curriculum update (for all updates except the first). It is calculated as $\Sigma_{\theta_u} \leftarrow \mathrm{diag}(((\theta_t - \theta_e)/\Lambda_{\Sigma_{\theta_u}})^2)$. As before, we also explore two choices for the meta-hyperparameter $\Lambda_{\Sigma_{\theta_u}}$, either 5 (coarse, MultiRoom and PCS) or 10 (fine, BipedalWalker). The heuristic for meta-hyperparameter $\Lambda_{\Sigma_{\theta_u}}$ was chosen by considering the probability distribution around $\mu_{\theta_u}$ prior to the curriculum update. The highest curriculum rewards should usually (within $\pm 1$ standard deviation, i.e., $\sim 68\%$) fall no more than one-fifth of the task space range $(\theta_t - \theta_e)$ away from $\mu_{\theta_u}$. For some domains like BipedalWalker, the resulting distribution is still too broad, necessitating a narrower heuristic of one-tenth the task space range.

4. The regularization hyperparameter $\lambda_\theta$ controls how much to penalize tasks based on difficulty in the initial curriculum update. The calculation is $\lambda_\theta \leftarrow \Lambda_{\lambda_\theta}||\vec{v}_{\theta_t - \theta_e}||_2$, where $\vec{v}_{\theta_t - \theta_e} \leftarrow \theta_t - \theta_e$. This means that the penalty on the curriculum reward will be no worse than the meta-hyperparameter $\Lambda_{\lambda_\theta}$, which is either 0.1 (MultiRoom and PCS) or 10.0 (BipedalWalker). The meta-hyperparameter $\Lambda_{\lambda_\theta}$ is intended to control how much optimization pressure is applied to bias tasks towards the easiest tasks during the initial curriculum update. We found that a heuristic of 0.1 was satisficing, except for BipedalWalker, where a much larger value was needed due to its very large task space.

5. The orthogonal distance hyperparameter $\lambda_d$ controls how tightly the curriculum should follow a straight line. Let $\theta_d$ be the environment parameters that have the largest distance from $\vec{v}_{\theta_t - \theta_e}$. Then, $\lambda_d \leftarrow \Lambda_{\lambda_d}||\vec{v}_{\theta_d - \theta_e} \perp \vec{v}_{\theta_t - \theta_e}||_2$. The meta-hyperparameter $\Lambda_{\lambda_d}$ is either 0.5 (PCS) or 10.0 (BipedalWalker). As with the meta-hyperparameter $\Lambda_{\lambda_\theta}$, the meta-hyperparameter $\Lambda_{\lambda_d}$ controls how influential the orthogonal distance loss is applied. Testing suggested that a value of 0.5 was satisficing, except for BipedalWalker, which benefited from a value of 10.0 and the resulting strong optimization pressure towards the target tasks.

## C    Experimental details

**Implementation**    Our work is implemented within the Dual Curriculum Design (DCD) codebase (Jiang et al., 2021a).[1] We use the official implementations of PLR$^\perp$ and ACCEL as provided in this codebase, as well as the implementation of ALP-GMM.

**Task solved threshold**    The task solved threshold $R_S$ indicates when a task has been solved based on its reward. An agent that receives a reward of at least $R_S$ on a task is said to have solved a task. This threshold is used to determine when training is no longer needed in a few ways in this work:

1. Time to threshold metrics are calculated when the evaluation reward obtained on the target task distribution meets or exceeds $R_S$. The RL training procedure may also conclude if early stopping is used.

2. CURATE uses $R_S$ to calculate the rewards $\nu$ based on competence (Eq. 5) used for the curriculum policy, which favors learning the set of easiest tasks not yet solved.

3. When assessing CURATE's average task performance, $R_S$ is taken as the average task performance threshold for calculating time to threshold metrics.

4. $R_S$ is used by the Hand Curriculum baseline in MultiRoom to indicate when it is time to advance to the next set of tasks in the curriculum.

We assume that $R_S$ is provided as part of the task definition. In practice, we train an RL agent using a random curriculum (i.e., domain randomization) to obtain what the maximum achievable reward in the target task distribution is. Then, we set $R_S$ slightly below that value.

**Maximum number of training frames**    The maximum allowable frames $f_{max}$ provides an upper limit to how long the RL agent is trained. For MultiRoom, this is 100 million frames. For PCS, Robust PLR uses 450 million frames, ACCEL uses 300 million frames, and all other approaches use 200 million frames. Robust PLR and ACCEL require additional frames to reach the desired amount of student gradient updates. In BipedalWalker, the maximum frames is usually 1 billion, except for Robust PLR and ACCEL, which exceed 2 billion.

### C.1    MiniGrid MultiRoom Navigation

MiniGrid MultiRoom requires the RL agent to master grid-based navigation within the MiniGrid domain (Chevalier-Boisvert et al., 2023). In MultiRoom, the agent must navigate through a series of rooms that are sequentially connected with doors separating the rooms. The agent always starts in the first room, and the goal always exists in the last room. The environment is a reimplementation of `MultiRoom-Random-N4` (Jiang et al., 2021b). However, we use the typical MiniGrid observation space as described below.

**Observation space**    The agent receives two observations:

1. A field-of-view observation consisting of the states of the world within a 7 x 7 grid within the agent's line of sight. The agent cannot see through walls.

2. The direction the agent is facing, represented as an integer with one of four values that each represent a different direction.

**Action space**    The agent uses discrete actions with action space $|\mathcal{A}| = 7$. The actions include turning left, turning right, going forward, picking up an object, dropping an object, toggling the activation for an object, and doing nothing. As there are no objects for the agent to pick up in this domain, the actions for picking up and dropping an object have no effect. Toggling the activation in front of a door will either open it (if closed) or close it (if opened).

---

[1]`https://github.com/facebookresearch/dcd`

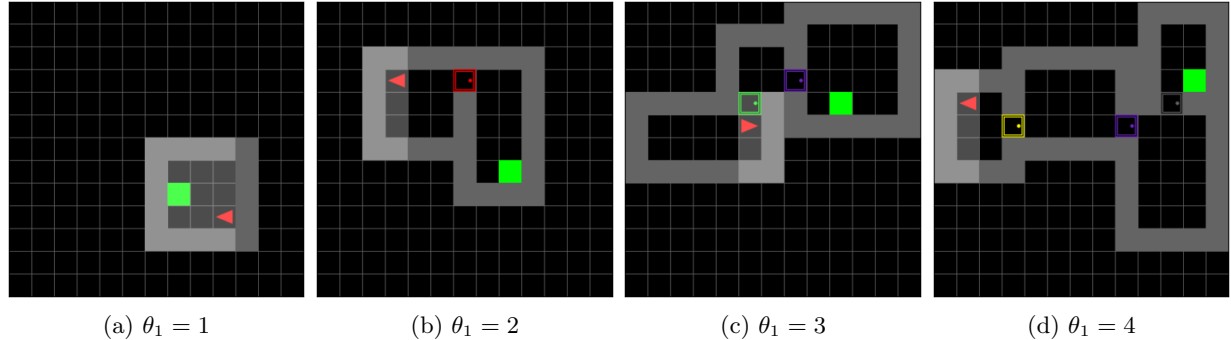

(a) $\theta_1 = 1$      (b) $\theta_1 = 2$      (c) $\theta_1 = 3$      (d) $\theta_1 = 4$

Figure 7: Variation in initial scenes for MultiRoom based on selection of environment parameters. Each figure represents an example task within the task distribution corresponding to the chosen environment parameters. For example, (a) represents a task with $\theta_1 = 1$ room.

**Reward**  The agent receives a time-discounted reward when solving the level by reaching the goal; zero reward is received otherwise. A task is considered solved if the agent receives at least 0.7 reward.

**Task space**  MultiRoom is a one-dimensional task space, where the curriculum axis $\theta_1 = [1, 4]$ controls the number of rooms in each task. Figure 7 shows example tasks from each parameter in this task space.

**Target task distribution**  For MultiRoom, the agent must solve a task distribution consisting of $\theta_t = 4$ rooms.

## C.2  Procgen Curriculum Suite

Procgen, as introduced by Cobbe et al. (2020), tasks RL agents to master different types of discrete control games. We build upon Procgen to introduce the Procgen Curriculum Suite (PCS) in this work. Please refer to App. I for a comprehensive description of PCS, e.g., including task spaces.

In our work, each game is adapted such that each level can be changed by specifying causal interventions in the environment parameters to change the initial level state. For example, the intervention $do(\theta_1 = 1, \theta_2 = 3)$ on level seed 0 in Leaper would yield the same level as without interventions, except with 1 road lane and 3 water lanes. Please refer to Fig. 8 for a visualized example. Note that for Leaper, intervention on these parameters may change other aspects of the initial state, such as the initial placement of cars and logs. Therefore, for Leaper, partial entanglement exists between $\Theta$ and other variables in the environment.

We implement the Procgen Curriculum Suite within the Procgen fork used by Jiang et al. (2021b)[2]

**Observation space**  The agent receives a 64 x 64 RGB image observation of the game.

**Action space**  The agent uses discrete actions with action space $|\mathcal{A}| = 15$. The actions generally correspond to eight directional actions, six special actions, and one action that does nothing. The actions are game-specific; please refer to Cobbe et al. (2020) for a complete description.

**Distribution mode**  We generally use the easy distribution mode of Procgen to avoid extra computational resources that would be required for the hard distribution mode. An exception is that we use the hard distribution mode for Leaper for a CURATE ablation. Unless specified to be the hard distribution mode, games are assumed to be the easy mode. Note that 7 games were too difficult for any consistently solve the target tasks, even in the easy distribution mode. Therefore, the task spaces for these games were simplified, leading to an "easier" distribution mode. Games in the easier distribution mode use the same mechanics as in the easy mode, except with smaller task spaces.

---

[2]https://github.com/minqi/procgen

Table 4: The 16 games used for the Procgen Curriculum Suite experiments.

| | | | |
|---|---|---|---|
| BigFish-Easier-1D-T | BossFight-Easier-3D-T | CaveFlyer-Easy-1D-T | Chaser-Easier-2D-T |
| Climber-Easier-2D-T | CoinRun-Easy-2D | Dodgeball-Easier-1D-T | FruitBot-Easy-3D-T |
| Heist-Easy-2D | Jumper-Easy-1D | Leaper-Easy-2D | Maze-Easy-1D |
| Miner-Easy-2D-T | Ninja-Easy-2D | Plunder-Easier-2D-T | StarPilot-Easier-4D-T |

**Games used for PCS Analysis**    Table 4 presents the 16 games used for the PCS experiments.

**Reward**    The rewards for Procgen games are game-specific and are explained in Cobbe et al. (2020), but generally, a reward of 10 is provided only upon solving a level, with no feedback provided otherwise. In order to realize this, 10 games were "sparsified" such that only a terminal reward was given when successfully completing a level, removing the intermediate rewards in the original game mechanics. These games are appended with -T in their game name. We found that otherwise, the intermediate reward tended to reduce or eliminate the need for a curriculum. For example, in Climber, the agent normally receives 1 reward for collecting coins that may spawn on platforms. However, this intermediate reward is a sufficiently informative proxy for climbing platforms, and thus, a curriculum becomes less beneficial. This mechanic was removed in terminal reward mode, so that the agent would only receive the level completion bonus when collecting all coins.

**Task space**    The task spaces for each PCS games are presented in Sec. I. The dimensionality of the task spaces vary from 1D to 4D.

**Target task distribution**    Generally, the target task distribution for each game contains the hardest levels that would be obtained in each game under the easy distribution mode (i.e., $\theta_t = \max(\Theta)$). In other words, no level is harder than what would have been possible to experience when randomly sampling levels from Procgen.

### C.3    BipedalWalker

In BipedalWalker, a bipedal walking robot must ambulate across challenging terrain without falling. This domain was first introduced by Brockman et al. (2016) and later expanded in Parker-Holder et al. (2022a). We use the experimental domain as given in Parker-Holder et al. (2022a), which features a broad task space and four different terrain challenges.

**Observation space**    The observation space $|\mathcal{O}| = 24$ consists of the state of the walker, including the walker hull angular position and velocity, the walker's translational velocity, angular position and velocities of the walker joints; whether the walker legs are in contact with the ground; and 10 LIDAR terrain readings in front of the walker.

**Action space**    The action space $|\mathcal{A}| = 4$ consists of torques to the motors controlling the walker joints.

**Reward**    Reward is given for forward progression. Small reward terms are also provided for shaping, e.g., more efficiently controlling the motors. The agent receives a penalty of -100 if it falls. An obtained reward of at least 300.0 generally indicates that the task was solved successfully.

**Task space**    The task space $|\Theta| = 8$ consists of parameters that control four factors that dictate the terrain difficulty. These parameters correspond to ground roughness, lower and upper bounds for pit gaps, lower and upper bounds for stump height, lower and upper bounds for stair height, and number of stairs.

### C.4    Experimentation hardware

For our MultiRoom experiments, our experiments are conducted on a computing cluster with an AMD EPYC 7413 (Zen 3) CPU (2.65 GHz, 128M cache L3) and an NVIDIA A100 SXM4 40GB GPU. For the Procgen

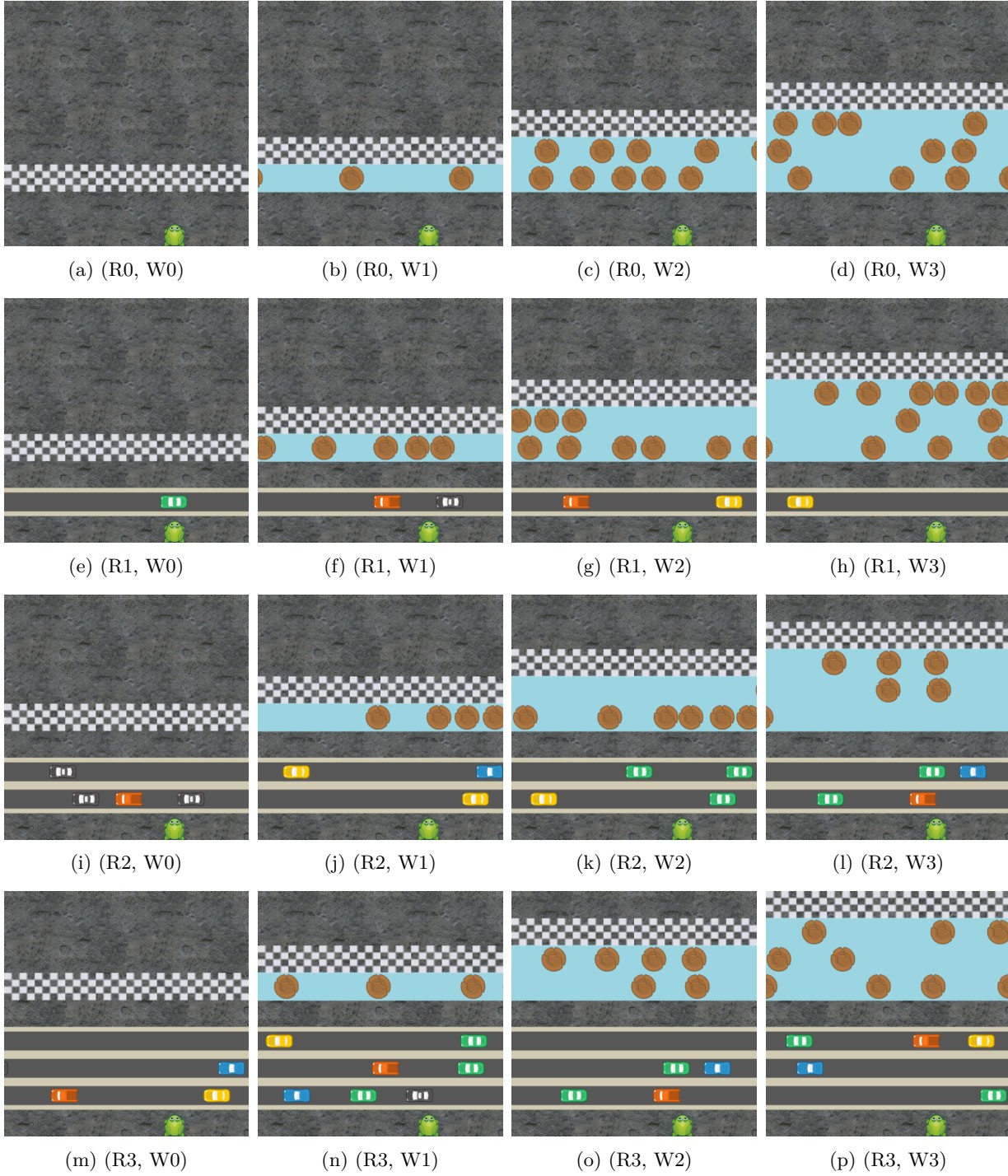

Figure 8: Example of variations in initial scenes for Leaper based on selection of environment parameters. Each figure represents an example task within the task distribution corresponding to the chosen environment parameters. For example, (h) represents a task with $\theta_1 = 1$ road lane and $\theta_2 = 3$ water lanes. All scenes are based on level seed 0.

Curriculum Suite, experiments were run on a different computing cluster with an Intel Xeon Gold 6448Y CPU (2.10 GHz, 60MB cache L3) and an NVIDIA H100 SXM5 80GB GPU. Our BipedalWalker experiments were run on the same cluster as our MultiRoom experiments, except training occurred on the CPU (no GPU was used).

### C.5 Hyperparameters

Table 5 presents the experimental hyperparameters. For MiniGrid MultiRoom, we use the hyperparameters from Jiang et al. (2021a) for their MiniGrid experiments. For Procgen, we generally use the same hyperparameters as the Procgen experiments in Jiang et al. (2021b) for the easy distribution. However, we use the episode length as defined by each game, and set the PPO rollout length to the nearest power of two. Then, we select minibatches per epoch such that each minibatch has 2048 samples, the same as in Jiang et al. (2021b). For BipedalWalker, we use the same hyperparameters as in Parker-Holder et al. (2022a).

For PLR$^\perp$, we generally use the same hyperparameters as Jiang et al. (2021a) for MultiRoom, Jiang et al. (2021b) for Procgen, and Parker-Holder et al. (2022a) for BipedalWalker. Our ACCEL hyperparameters come from Parker-Holder et al. (2022a) where possible for MultiRoom and Bipedalwaker, and they are similarly chosen for Procgen.

## D Average task performance results

This section presents the average task performance experiments for MultiRoom, CoinRun, and Leaper. We choose CoinRun and Leaper as representative games from PCS, as CoinRun is the most well-known Procgen game and Leaper is our most studied game. They both represent cases where CURATE offers strong average task performance without additional modifications (MultiRoom, CoinRun) and where off-curriculum strategies may be needed (Leaper). Overall, we find that CURATE offers strong average task performance and notably outperforms UED/DCD baselines that specialize in generalization.

### D.1 Average Task Performance for MiniGrid MultiRoom and PCS CoinRun

For both MultiRoom and CoinRun, we find evidence that CURATE curricula lead to strong performance across all tasks, not only the target tasks. As a result, CURATE usually yields the best metrics as compared to our baselines. Table 6 and Tab. 7 present these findings for MultiRoom and CoinRun in terms of summary statistics, respectively. These results correspond to learning curves shown in Fig. 5a and Fig. 5b. Heatmaps showing final per-task test returns are also shown in Fig. 9 and Fig. 10 for MultiRoom and CoinRun, respectively. For MultiRoom, we find evidence that CURATE-0.3 offers strong average task performance in this domain, where it is best achieving in terms of TTU and second in final test return only to the pseudo-oracle HC. For CoinRun, CURATE offers nearly identical average task performance in terms of TTU compared to best method, DR. However, CURATE leads to much stronger final reward.

### D.2 Average Task Performance for PCS Leaper through Off-Curriculum Sampling

Although we find that CURATE by itself is performant in solving the target tasks, CURATE by itself does not generalize well to other tasks within the task space for Leaper, leading to poor average task performance and offering contrary results to MultiRoom and CoinRun. These results for Leaper are shown in Tab. 8 (summary statistics), Fig. 5c (learning curves), and Fig. 11 (heatmaps showing final per-task test returns). In this case, the CURATE curriculum may lead to somewhat brittle and narrow representations, only leading to a specialist student that is adept at solving the hardest tasks, but does not generalize to others. To illustrate this, Fig. 11a shows that although CURATE has strong target task performance, CURATE performs poorly for tasks consisting of multiple water lanes and few road lanes. We hypothesize that these tasks are relatively difficult yet are sufficiently different from the target tasks, so the benefits of curricula do not transfer well without explicitly training on these tasks. Conversely, we further hypothesize that tasks consisting of multiple road lanes and few water lanes are less difficult, offering more margin for error than the demanding water lanes. Thus, some transfer is retained for these tasks.

Table 5: Hyperparameters used for experiments. Note that for CURATE, $N_r$ can take different values depending on whether it is the initial curriculum update or not. For PCS, Leaper Hard uses the same parameters, except $N_v$ is 256 per Cobbe et al. (2020).

| Hyperparameter | MultiRoom | PCS | BipedalWalker |
|---|---|---|---|
| Discount factor $\gamma$ | 0.995 | 0.999 | 0.99 |
| $\lambda_{GAE}$ | 0.95 | 0.95 | 0.9 |
| Rollout length | 256 | {512, 1024, 4096, 6144} | 2048 |
| Epochs | 5 | 3 | 5 |
| Minibatches per epoch | 1 | 16 | 32 |
| Clip range | 0.2 | 0.2 | 0.2 |
| Number of parallel workers $N_v$ | 32 | {easy/easier: 64, hard: 256} | 16 |
| Return normalization | no | yes | yes |
| Entropy bonus coefficient | 0.0 | 0.01 | 0.001 |
| Value loss coefficient | 0.5 | 0.5 | 0.5 |
| Max gradient norm | 0.5 | 0.5 | 0.5 |
| Adam learning rate | 0.0001 | 0.0005 | 0.0003 |
| Adam $\epsilon$ | 0.00001 | 0.00001 | 0.00001 |
| Recurrent agent | yes | no | no |
| Action space dimensionality $|\mathcal{A}|$ | 7 | 15 | 4 |
| Action type | discrete | discrete | continuous |
| Episode length | 80 | {500, 1000, 4000, 6000} | 2000 |
| Reward threshold $R_S$ | 0.7 | 8.0 | 200.0 |
| Target episodes per eval. $N_{te}$ | 128 | 128 | 10 |
| Task space dimensionality $|\Theta|$ | 1 | {1, 2, 3, 4} | 8 |
| Max. train frames $f_{max}$ $(\times 10^6)$ | 100.0 | at least 200.0 | at least 1,000.0 |
| Replay rate | 0.5 | 0.5 | 0.5 |
| PLR prioritization | rank | rank | rank |
| Temperature $\beta$ | 0.3 | 0.1 | 0.1 |
| Staleness coefficient $\rho$ | 0.3 | 0.1 | 0.5 |
| Replay buffer size | 4000 | 4000 | 1000 |
| Scoring function loss | positive value | L1 value | positive value |
| Edit rate | 1.0 | 1.0 | 1.0 |
| Replay rate | 0.8 | 0.8 | 0.9 |
| Number of edits | 3 | 3 | 3 |
| Edit method | random | random | random |
| Levels edited | easy | easy | easy |
| Number rounds $N_r$ | 4/2 | 4/2 | 8/2 |
| Samples per round $N_s$ | 8 | 16 | 64 |
| Episodes per eval. sample $N_{se}$ | 16 | 32 | 10 |
| Regularization hyperparameter $\lambda_\theta$ | auto | auto | auto |
| Distance hyperparameter $\lambda_d$ | auto | auto | auto |
| Curriculum update initialization | coarse | coarse | fine |
| REPS relative entropy bound $\epsilon$ | 0.75 | 0.75 | 0.75 |
| REPS minimum temperature $\eta$ | 0.05 | 0.05 | 0.05 |
| Min. updates before curr. update $N_{min}$ | 16 | 16 | 128 |
| Max. updates before curr. update $N_{max}$ | 128 | 128 | 256 |

Table 6: Average test performance results for MultiRoom across all tasks using 5,000 combined updates (student PPO and teacher, if used) for 10 trials. Summary statistics are shown in terms of success rate (if the average test performance across all tasks is at least $R_S$), time-to-threshold in updates for the average test performance across all tasks to exceed $R_S$ (TTU), and the final average test performance across all tasks (F. Test). Trials that do not solve the task still count towards summary statistics. TTU is $\times 10^3$. TTU and F. Test are shown in terms of IQM with 95% CI (Agarwal et al., 2021). The best approach for each metric is **bolded**, the second best approach is underlined, and an asterisk (*) denotes that the approach falls within the best approach's CI.

| Method | Success Rate | TTU ↓ | TTU CI | F. Test ↑ | F. Test CI |
|---|---|---|---|---|---|
| CURATE (ours) | **100.000%** (10) | 1.357 | (1.058, 1.542) | 0.827 | (0.825, 0.834) |
| CURATE-0.3 (ours) | **100.000%** (10) | **0.835** | (0.780, 0.930) | 0.841 | (0.828, 0.854) |
| Hand Curr. | **100.000%** (10) | 0.838* | (0.770, 0.906) | **0.852** | (0.846, 0.853) |
| Domain Rand. | **100.000%** (10) | 1.139 | (1.017, 1.920) | 0.827 | (0.824, 0.828) |
| Robust PLR | **100.000%** (10) | 1.581 | (1.277, 1.931) | 0.825 | (0.823, 0.827) |
| ACCEL | **100.000%** (10) | 0.955 | (0.865, 1.047) | 0.824 | (0.823, 0.826) |
| Target | 0.000% (0) | 5.000 | (5.000, 5.000) | 0.007 | (0.000, 0.042) |

Table 7: Average test performance results for PCS CoinRun (`CoinRun-Easy-2D`) across all tasks using 3,000 combined updates (student PPO and teacher, if used) for 6 trials. Summary statistics are shown in terms of success rate (if the average test performance across all tasks is at least $R_S$), time-to-threshold in updates for the average test performance across all tasks to exceed $R_S$ (TTU), and the final average test performance across all tasks (F. Test). Trials that do not solve the task still count towards summary statistics. TTU is $\times 10^3$. TTU and F. Test are shown in terms of IQM with 95% CI (Agarwal et al., 2021). The best approach for each metric is **bolded**, the second best approach is underlined, and an asterisk (*) denotes that the approach falls within the best approach's CI.

| Method | Success Rate | TTU ↓ | TTU CI | F. Test ↑ | F. Test CI |
|---|---|---|---|---|---|
| CURATE (ours) | **100.000%** (6) | 0.043* | (0.041, 0.046) | **9.665** | (9.622, 9.676) |
| Domain Rand. | **100.000%** (6) | **0.042** | (0.041, 0.046) | 9.306 | (9.275, 9.401) |
| Robust PLR | **100.000%** (6) | 0.073 | (0.066, 0.079) | 9.305 | (9.138, 9.384) |
| ACCEL | **100.000%** (6) | 0.150 | (0.132, 0.202) | 8.754 | (8.521, 8.966) |
| Target | 66.667% (4) | 0.812 | (0.075, 2.273) | 7.697 | (3.715, 9.701) |

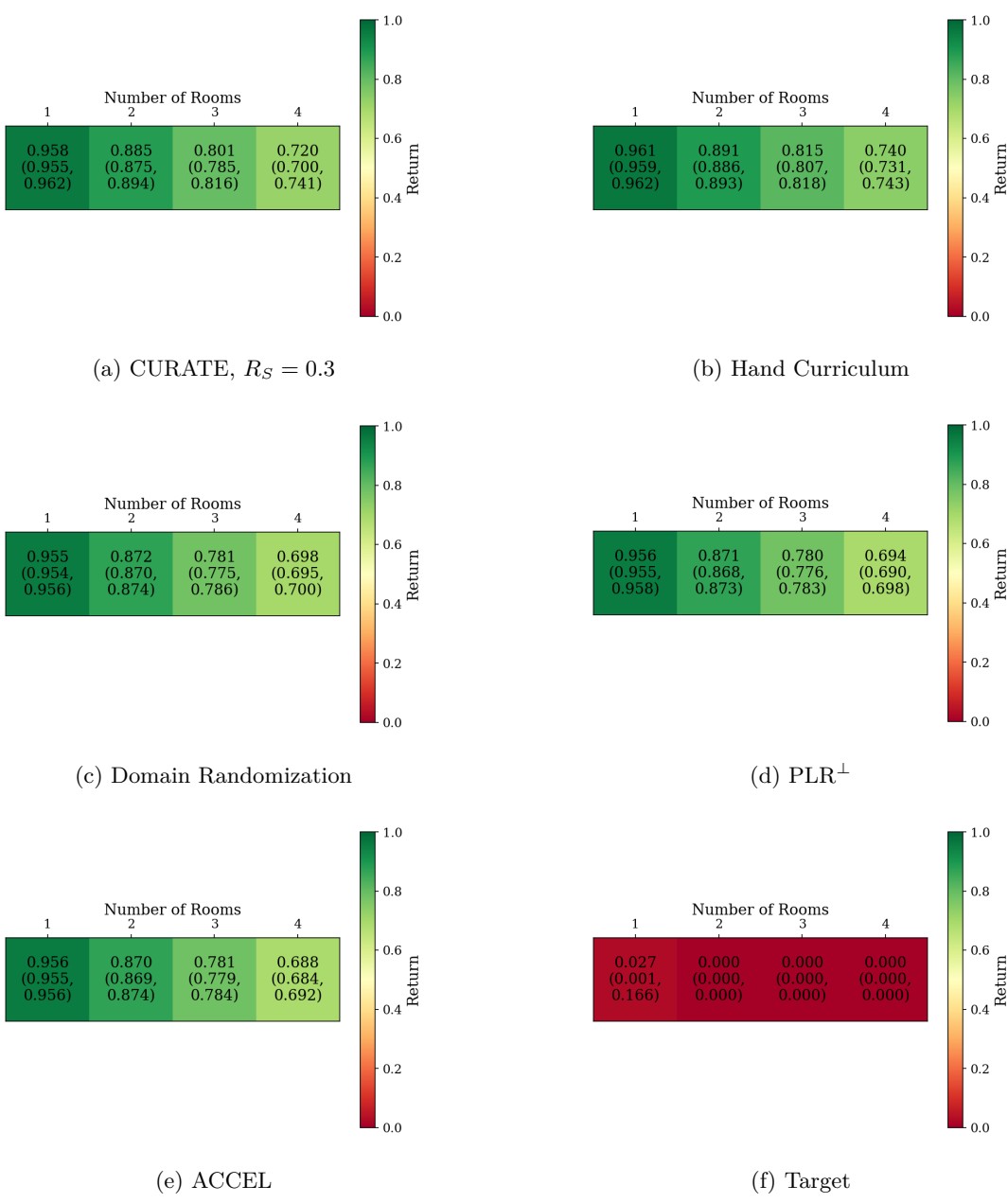

Figure 9: Final test return heatmaps for each approach and task in MultiRoom using 5,000 combined updates (student PPO and teacher, if used) for 10 trials. Test returns are shown in terms of IQM with 95% CI (Agarwal et al., 2021), where aggregation is done on a per-task basis.

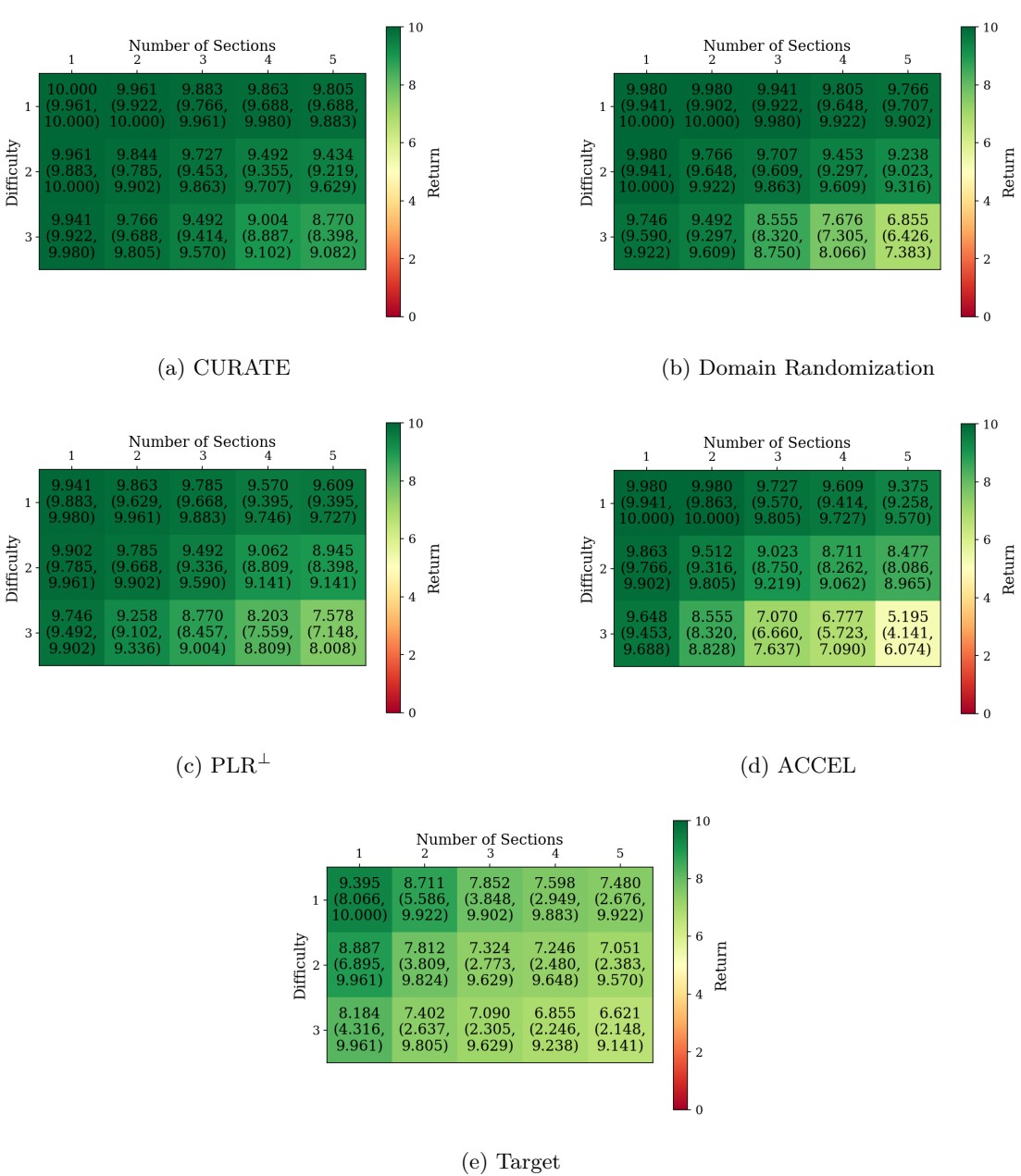

(a) CURATE

(b) Domain Randomization

(c) PLR$^\perp$

(d) ACCEL

(e) Target

Figure 10: Final test return heatmaps for each approach and task in PCS CoinRun (`CoinRun-Easy-2D`) using 3,000 combined updates (student PPO and teacher, if used) for 6 trials. Test returns are shown in terms of IQM with 95% CI (Agarwal et al., 2021), where aggregation is done on a per-task basis.

Table 8: Average test performance results for PCS Leaper (`Leaper-Easy-2D`) across all tasks using 5,000 combined updates (student PPO and teacher, if used) for 6 trials. Summary statistics are shown in terms of success rate (if the average test performance across all tasks is at least $R_S$), time-to-threshold in updates for the average test performance across all tasks to exceed $R_S$ (TTU), and the final average test performance across all tasks (F. Test). Trials that do not solve the task still count towards summary statistics. TTU is $\times 10^3$. TTU and F. Test are shown in terms of IQM with 95% CI (Agarwal et al., 2021). The best approach for each metric is **bolded**, the second best approach is underlined, and an asterisk (*) denotes that the approach falls within the best approach's CI.

| Method | Success Rate | TTU ↓ | TTU CI | F. Test ↑ | F. Test CI |
|---|---|---|---|---|---|
| CURATE (ours) | 0.000% (0) | 5.000 | (5.000, 5.000) | 5.686 | (5.139, 6.296) |
| CURATE, 37.5% OCS-S (ours) | **100.000%** (6) | 0.384 | (0.328, 0.553) | 9.160 | (9.055, 9.227) |
| Domain Rand. | **100.000%** (6) | **0.306** | (0.241, 0.379) | **9.271** | (9.191, 9.363) |
| Robust PLR | **100.000%** (6) | 0.422 | (0.301, 0.538) | 9.012 | (8.838, 9.122) |
| ACCEL | **100.000%** (6) | 1.025 | (0.785, 1.396) | 8.732 | (8.639, 8.918) |
| Target | 0.000% (0) | 5.000 | (5.000, 5.000) | 0.867 | (0.648, 0.886) |

However, we find that adding a straightforward off-curriculum sampling (OCS) strategy to CURATE boosts average task performance, significantly strengthening its performance so that it ranks second to DR. In this case, 37.5% of the parallel workers for CURATE randomly sample from tasks for which $\theta \prec \mu_\theta$ (OCS-S), with the remainder of the parallel workers sampling from CURATE's task distribution. However, this improved generalization comes at a price: we find that the use of 37.5% OCS-S reduces the time to threshold for solving the target tasks (IQM: 0.418, 95% CI: (0.353, 0.527)) as compared to CURATE without an off-curriculum strategy (IQM: 0.362, 95% CI: (0.305, 2.686)), likely due to less samples going towards pushing forward the learning frontier. Notably, we also find that CURATE with 37.5% OCS-S still outperforms DR in time-to-threshold for the target tasks (IQM: 0.540, CI: (0.412, 0.766)), even if DR is better in average task performance. Overall, this result confirms compatibility of CURATE with off-curriculum strategies and the associated benefits, e.g., improved generalization and better mitigation of forgetting.

# E    CURATE sensitivity and ablation experiments

In this section, we present four sensitivity and ablation experiments to understand the important components of CURATE and their contribution to performance. Note that the exact performance and importance of components may vary from domain to domain (and in PCS, game to game), so we select representative domains for each experiment where the effect of the component is pronounced.

## E.1    MultiRoom: Sensitivity of task solved threshold

Results for using a fixed value of $R_S$ throughout training (for both the intermediate and target tasks) are shown in Fig. 12a. We find that the relative ordering is generally consistent below $R_S = 0.5$, but this order changes for higher values of $R_S$. We find that HC usually performs best overall, followed by CURATE (except for 0.6). Note that the variance for CURATE increases significantly for $R_S$ values above 0.5. Other baselines except for HC generally see a similar increase at higher values of $R_S$.

If $R_S$ can differ between the intermediate and target tasks, as shown in Fig. 12b, we find that CURATE can achieve significant performance improvements, narrowing the gap with HC. Generally, we find that the lower the value of $R_S$ used by CURATE for training tasks, the higher the performance increase (except for the lowest $R_S$, 0.2). This finding may arise because a lower threshold allows the agent to progress more rapidly through the curriculum and not get stuck in the curriculum "bottleneck", where it can then devote its training budget to solving the target tasks. We find that the selection of $R_S$ used for HC usually does not lead to statistically significant differences, except for larger $R_S$, where a higher $R_S$ is better.

When choosing a fixed value of 0.7 for the target tasks, as in Fig. 12c, we find that CURATE has the overall best performance using an intermediate $R_S$ threshold of 0.3, again noting that a larger value leads to

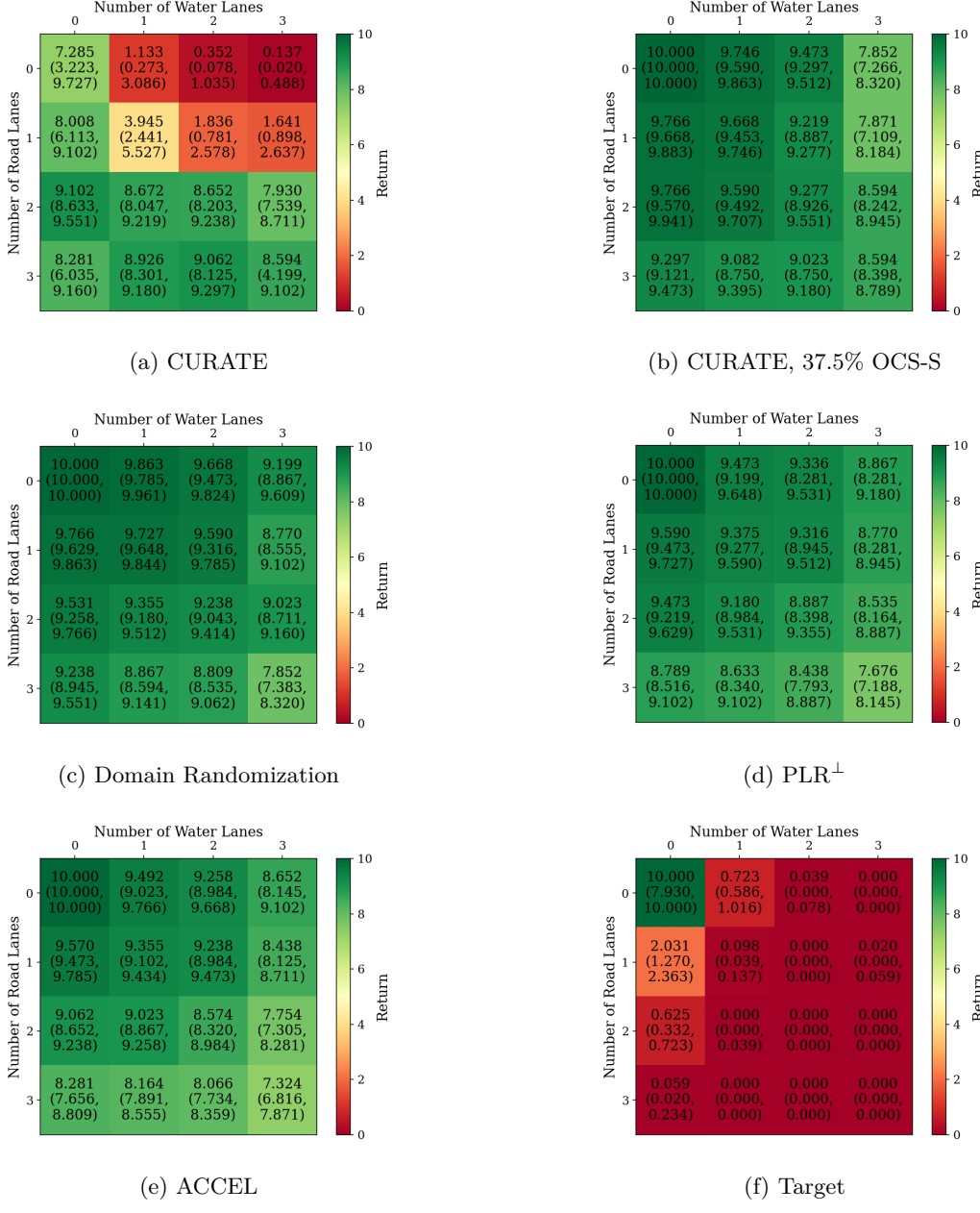

Figure 11: Final test return heatmaps for each approach and task in PCS Leaper (`Leaper-Easy-2D`) using 5,000 combined updates (student PPO and teacher, if used) for 6 trials. Test returns are shown in terms of IQM with 95% CI (Agarwal et al., 2021), where aggregation is done on a per-task basis.

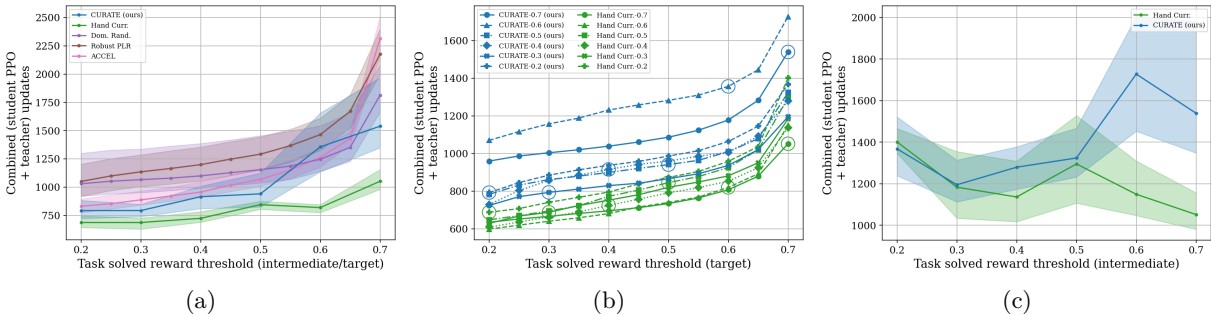

Figure 12: Ablation curves for MultiRoom to understand the importance of the task solved threshold $R_S$ and whether this value can be the same or different for both intermediate and target tasks. Curves are shown in terms of time to threshold in combined updates (student PPO and teacher, if used) until $\mathcal{M}_{\theta_t}$ is solved. For clarity, curves for Target (NC) are not shown as no trials solved $\mathcal{M}_{\theta_t}$. Curves are shown in terms of IQM with 95% CI (Agarwal et al., 2021). (a) Task solved threshold $R_S$ is the same for the target tasks and the intermediate tasks (CURATE, HC). (b) The task solved threshold $R_S$ is different for the target tasks ($x$-axis) and the intermediate tasks (varies for each signal of CURATE and HC). A circle annotation indicates that the task solved threshold is the same for both intermediate and target tasks. For clarity, the CIs are not shown. (c) Task solved threshold $R_S$ varies for the intermediate tasks ($x$-axis) in CURATE and HC, but a fixed target task threshold of 0.7 is used.

performance degradation. This value was used for the rest of our experiments for the CURATE-0.3 method. For HC, using the same value of 0.7 for both intermediate and final tasks was most performant.

### E.2 MultiRoom: Sensitivity of evaluations

Following Sec. E.1, which showed that CURATE is best performing with an intermediate task reward threshold of 0.3, we performed sensitivity experiments to understand the effect of evaluations used during the curriculum update for the MultiRoom domain. We primarily consider two hyperparameters: 1) number of episodes per evaluation sample $N_{ee}$, and 2) minimum number of student PPO updates before a curriculum update can be run $N_{min}$. Generally, increasing the number of episodes per evaulation sample improves the informativeness of a sample in terms of gauging the agent's competence, but it increases the number of teacher samples. Decreasing the minimum number of student updates means that the curriculum update can occur more frequently, but this also increases the teaching samples. We sweep number of parameters from $\{8, 16, 32\}$ for both hyperparameters.

Results are shown in Fig. 13, in terms of time to threshold to solve $\mathcal{M}_{\theta_t}$ for (a) student-only updates and (b) combined (student and teacher) updates. We find that moderate values of 16 for each of these two hyperparameters effectively balances between more informative, frequent updates and less infrequent and lower teacher samples. Importantly, the chosen hyperparameters also enable CURATE to match (with 95% CI) HC when only considering student updates.

In terms of number of episodes per evaluation sample, too few (8 episodes) may degrade the curriculum, selecting tasks that are not effective. When considering teacher samples, lower sample efficiency of too large episodes (32 episodes) can be explained by increasing the number of episodes overall in the curriculum update. However, there appears to be some sensitivity even for only considering the student samples alone. This mechanism is less understood, but it could be due to the training distribution induced by the curriculum overly collapsing. In terms of minimum student updates, there may be a benefit to not changing the curriculum as frequently (8 updates), which may better stabilize training. However, updating too infrequently (32 updates) also leads to poor sample efficiency by not pushing the curriculum forward.

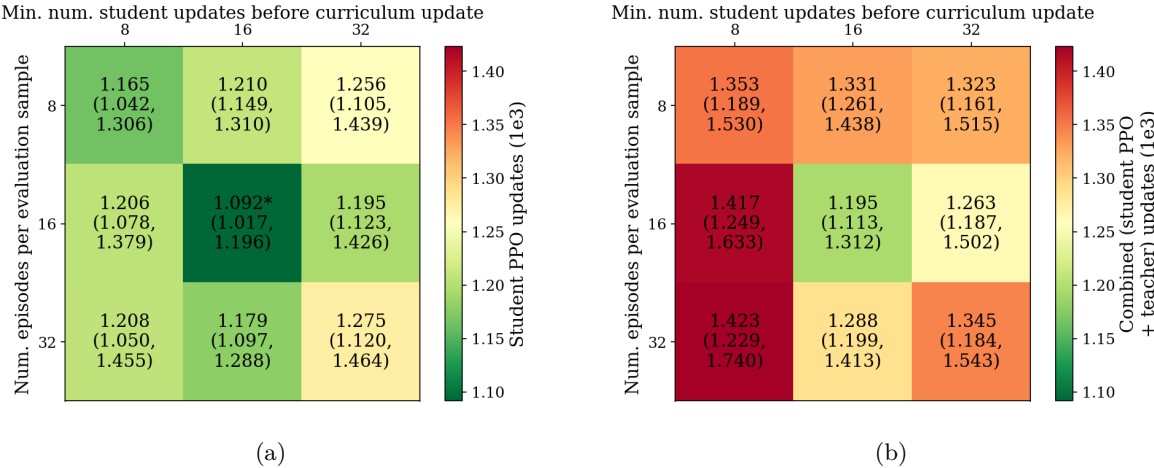

(a)

(b)

Figure 13: Sensitivity analysis of time to threshold in updates (TTU) to solve $\mathcal{M}_{\theta_t}$ for the MultiRoom domain when varying the hyperparameters of 1) number of episodes per evaluation sample $N_{ee}$ and 2) minimum number of student PPO updates before a curriculum update can be run $N_{min}$. TTU is shown in terms of IQM with 95% CI (Agarwal et al., 2021). Tables are shown (a) with student updates alone and (b) with cominbed (student and teacher) updates. An asterisk (*) denotes that the configuration falls within the 95% CI of TTU for HC.

### E.3 BipedalWalker: Ablating regularization loss

To better understand the effect on the regularization loss $\mathcal{L}_{diff}$, we conduct an ablation experiment in the BipedalWalker domain. Specifically, we evaluate CURATE, but without the $\mathcal{L}_{diff}$ term in the initial curriculum update. Without this term, the training task distribution offered by the curriculum policy $\pi_c$ represents an approximately uniform distribution over the task space. Figure 14 shows the results of this ablation experiment. Aside from under 2,500 updates, we find that removing $\mathcal{L}_{diff}$ significantly degrades performance for most of training. We hypothesize that the narrower training distribution when using $\mathcal{L}_{diff}$ boosts learning by focusing the training tasks into a smaller distribution. Interestingly, we find that the gap narrows towards the end of training, with the $\mathcal{L}_{diff}$ ablation slightly under the 95% CI of CURATE.

### E.4 Leaper Hard: Ablating distance loss and informed initialization

Lastly, we present ablations for the components of CURATE that provide task directedness. For our domain, we choose PCS Leaper Hard (`Leaper-Hard-2D`) for its challenge and importance of task-directed curricula.

Our findings in Tab. 9 show that removing $\mathcal{L}_{dist}$ and the informed initialization of $\Delta\mu_\theta$ both degrade performance, although of the two, $\Delta\mu_\theta$ has the stronger effect (at least in this domain). When removing $\mathcal{L}_{dist}$ alone, we find that the curricula tend to meander, leading to an increase in TTU and TTS, but having $\Delta\mu_\theta$ still enables enough task directedness for each trial to be successful. However, removing informed initialization via $\Delta\mu_\theta$ alone causes a notable performance degradation, leading to trials not being successful. We find that for the failed trials, their curricula generally stay aligned along $\vec{v}_{\theta_t-\theta_i}$, but the loss of the informed initialization mechanism to push the frontier forward leads to the student stagnating and being unable to solve the target tasks. When both $\mathcal{L}_{dist}$ and $\Delta\mu_\theta$ are removed, the resulting curricula meander widely throughout the task space, tending to require almost three times as many samples. Although the effect of these two components are ultimately task dependent, in domains such as Leaper Hard where task-directness is key, these components are critical for CURATE's best performance.

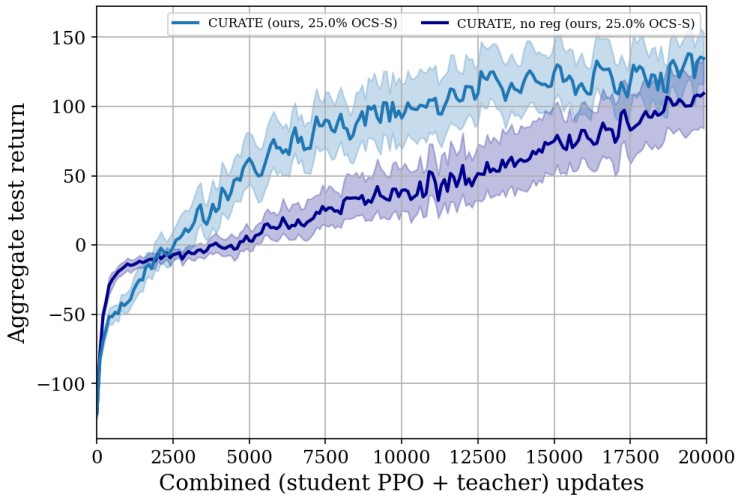

Figure 14: Learning curve in terms of aggregate test return for the seven test environments of interest in the BipedalWalker domain. CURATE is shown along with the ablation of CURATE without the regularization loss $\mathcal{L}_{diff}$. Curve is interpolated across methods and shown in terms of IQM over 10 trials with 95% CI (Agarwal et al., 2021).

Table 9: Target task performance results for CURATE in PCS Leaper Hard (`Leaper-Hard-2D`) using $500 \times 10^6$ training steps for 12 trials while ablating components that provide task directedness. Summary statistics are shown in terms of success rate (if the target tasks $\mathcal{M}_{\theta_t}$ were solved) and time-to-threshold both in combined updates (student PPO and teacher, TTU) or combined steps (student and teacher, TTS) until $\mathcal{M}_{\theta_t}$ is solved or maximum steps reached. Trials that do not solve the task still count towards summary statistics. TTU is $\times 10^3$, and TTS is $\times 10^6$. TTU and TTS are shown in terms of IQM with 95% CI (Agarwal et al., 2021). The best approach for each metric is **bolded**, the second best approach is underlined, and an asterisk (*) denotes that the approach falls within the best approach's CI.

| Method | Success Rate | TTU ↓ | TTU CI | TTS ↓ | TTS CI |
|---|---|---|---|---|---|
| CURATE | **100.0%** (12) | **1.201** | (1.123, 1.293) | **151.975** | (142.690, 163.230) |
| CURATE, $-\mathcal{L}_{dist}$ | **100.0%** (12) | 1.299 | (1.188, 1.614) | 162.855* | (149.418, 202.198) |
| CURATE, $-\Delta\mu_\theta$ | 66.667% (8) | 3.146 | (2.173, 3.945) | 390.764 | (269.139, 493.581) |
| CURATE, $-\mathcal{L}_{dist}, -\Delta\mu_\theta$ | 66.667% (8) | 3.393 | (3.012, 3.937) | 422.522 | (371.046, 491.600) |

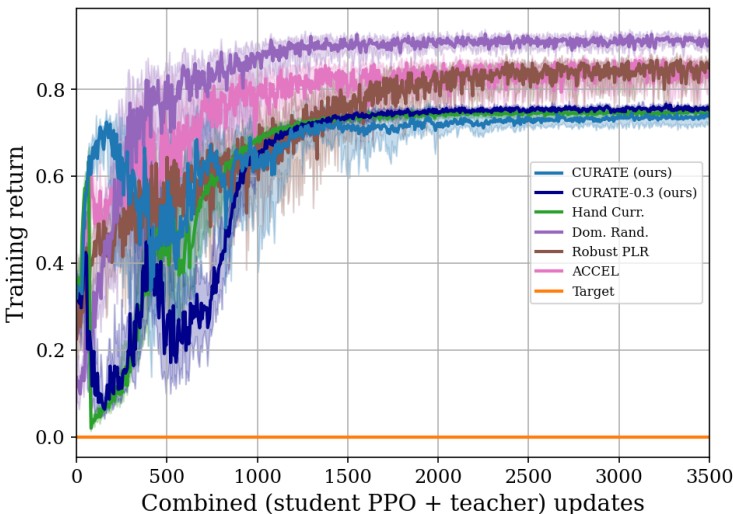

Figure 15: Learning curve in terms of training return for the MiniGrid MultiRoom domain. Curve is interpolated across methods with subsampling every 5 updates and shown in terms of IQM over 10 trials with 95% CI (Agarwal et al., 2021).

## F   Supplemental results for MiniGrid MultiRoom

The training learning curve for MultiRoom is shown in Fig. 15. Figure 16 presents a representative curricula and training/target learning curves for each approach. The representative trial for each approach is the closest trial to the median of all 10 trials used for that approach. We show results for CURATE with $R_S = 0.3$ as it was significantly better performing than with $R_S = 0.7$.

### F.1   Domain randomization experiment with evaluation on all tasks

To provide intuition for the key property that empowers CURATE curricula through the competence learning objective $\mathcal{R}_{comp}$, we conducted an experiment where we evaluate on all tasks after each PPO update, not just the target tasks. Figure 17 shows the results. Generally, we see that the evaluation returns on tasks that are nearby to those used for training increase inversely proportional to their difficulty. For example, an agent that is training on $\theta_2$ tasks will see some increase in evaluation return when evaluated on $\theta_3$ tasks and $\theta_4$ tasks due to transfer learning. However, this improvement is less for $\theta_4$ than $\theta_3$, as $\theta_4$ is more difficult than $\theta_3$. This property is leveraged by CURATE, so that as training tasks become solved, CURATE can select the easiest nearby tasks that are unsolved via optimizing $\mathcal{R}_{comp}$ in order to advance the curriculum.

## G   Supplemental results for the Procgen Curriculum Suite

Learning curves in terms of the training return and target return for all 16 games are shown in Fig. 18 and Fig. 19, respectively. We also present representative curricula and training/target learning curves for each approach for PCS Leaper (`Leaper-Easy-2D`); please refer to Fig. 20 and Fig. 21. The representative trial for each method is the closest trial to the median of all 6 trials for that method. Lastly, we also present summary statistics for each game in Tab. 10 through Tab. 25.

## H   Supplemental results for BipedalWalker

Supplemental results for the BipedalWalker domain are presented in this section. Figure 22 shows the training learning curve. In terms of test performance, Figure 23 presents the learning curve for each of the seven test environments, and Tab. 26 shows the full evaluation at the conclusion of training (expanding Tab. 3). We note that the test learning curves in Fig. 3c and Fig. 23 use 10 episodes per environment per

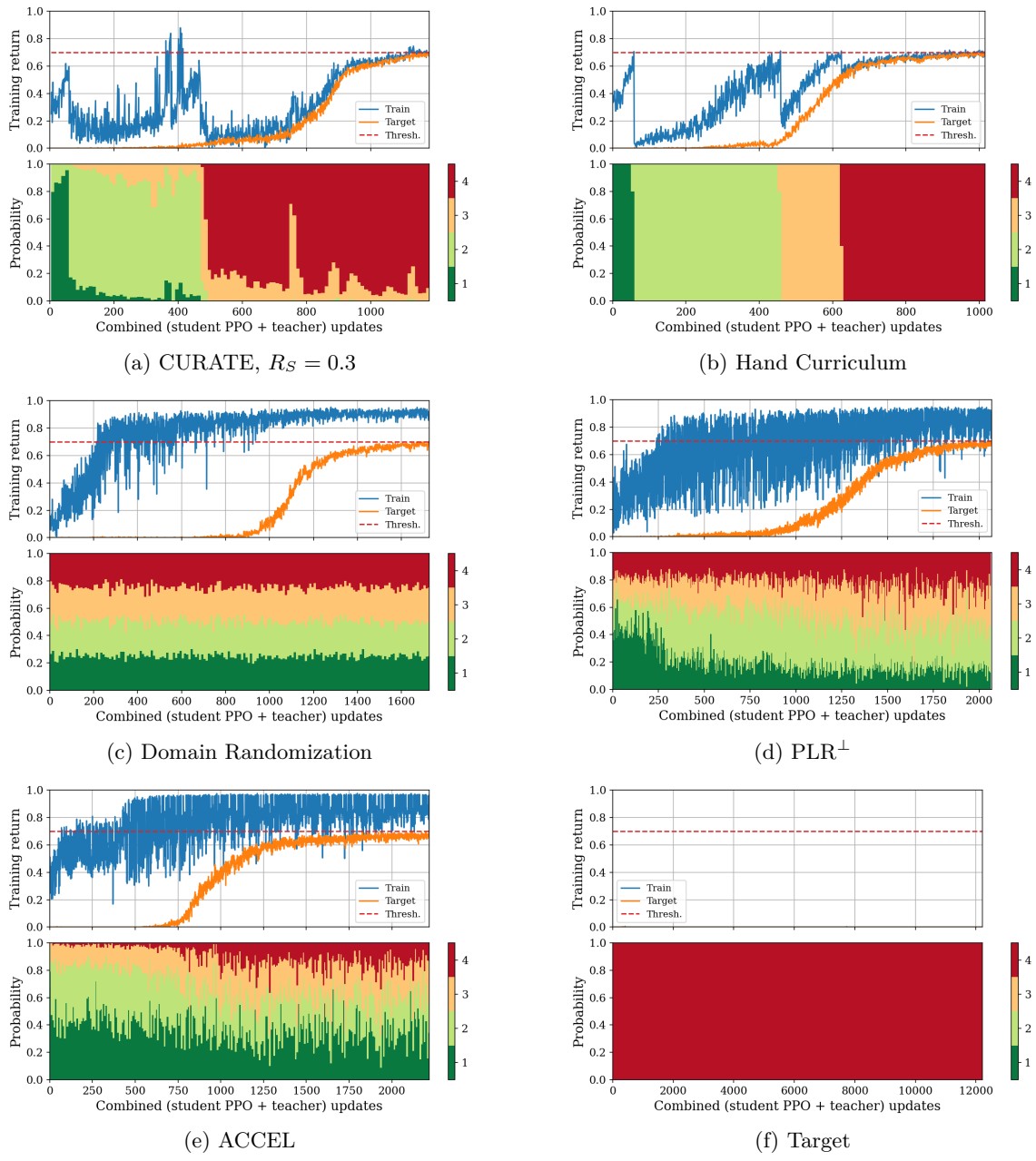

Figure 16: Representative curriculum learning time histories for each approach in MiniGrid MultiRoom. Each time history shows the trial that is closest to the median performance of all 10 trials for each approach. The top figure shows the time history of the return, shown for the training environments and the target task. The bottom figure shows the time history of the curriculum, with time average discretization of 10 updates to better show long-term trends.

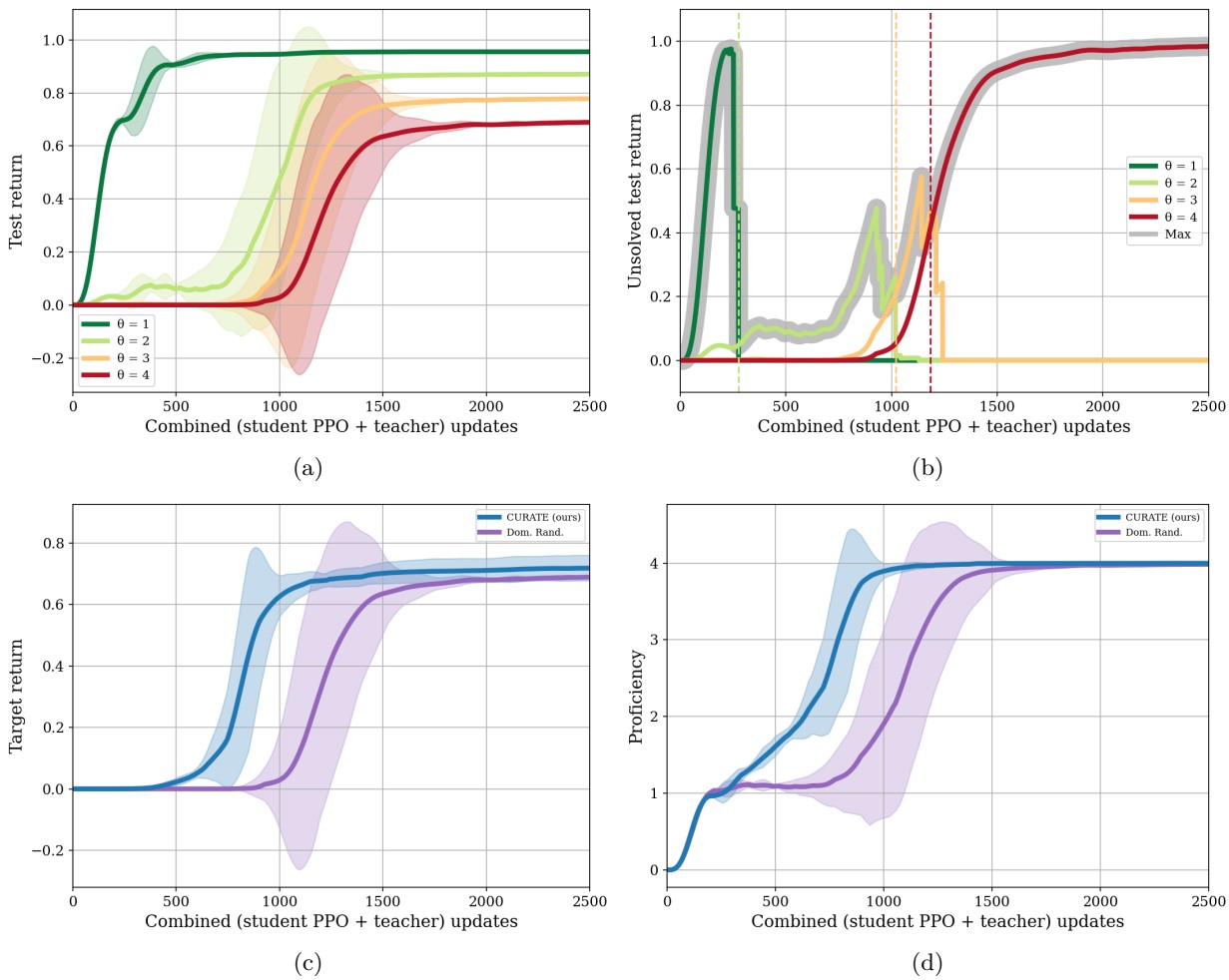

Figure 17: Experiment demonstrating the intuition behind CURATE's competence learning objective, $\mathcal{R}_{comp}$. (a) Even when using a random curriculum (DR), we observe that tasks that are nearby those used for training will exhibit performance increases (measured by evaluation return) that are inversely proportional to their difficulty. Thus, the temporal ordering of task mastery (i.e., when a task is solved) occurs by difficulty. (b) Using $\mathcal{R}_{comp}$, a measure of task competence in unsolved tasks (Eq. 5), a natural easy-to-hard task ordering can emerge by selecting tasks that maximize this measure. CURATE uses this competence learning objective $\mathcal{R}_{comp}$ to induce an approximately easiest-to-hardest curriculum that generally outperforms curricula that do not exploit this property, in terms of both (c) target task performance and (d) overall proficiency of the agent (i.e., the fraction of tasks it can solve). Experiments are from the MiniGrid MultiRoom domain (Sec. 5). Curves are calculating using median aggregation over 10 trials.

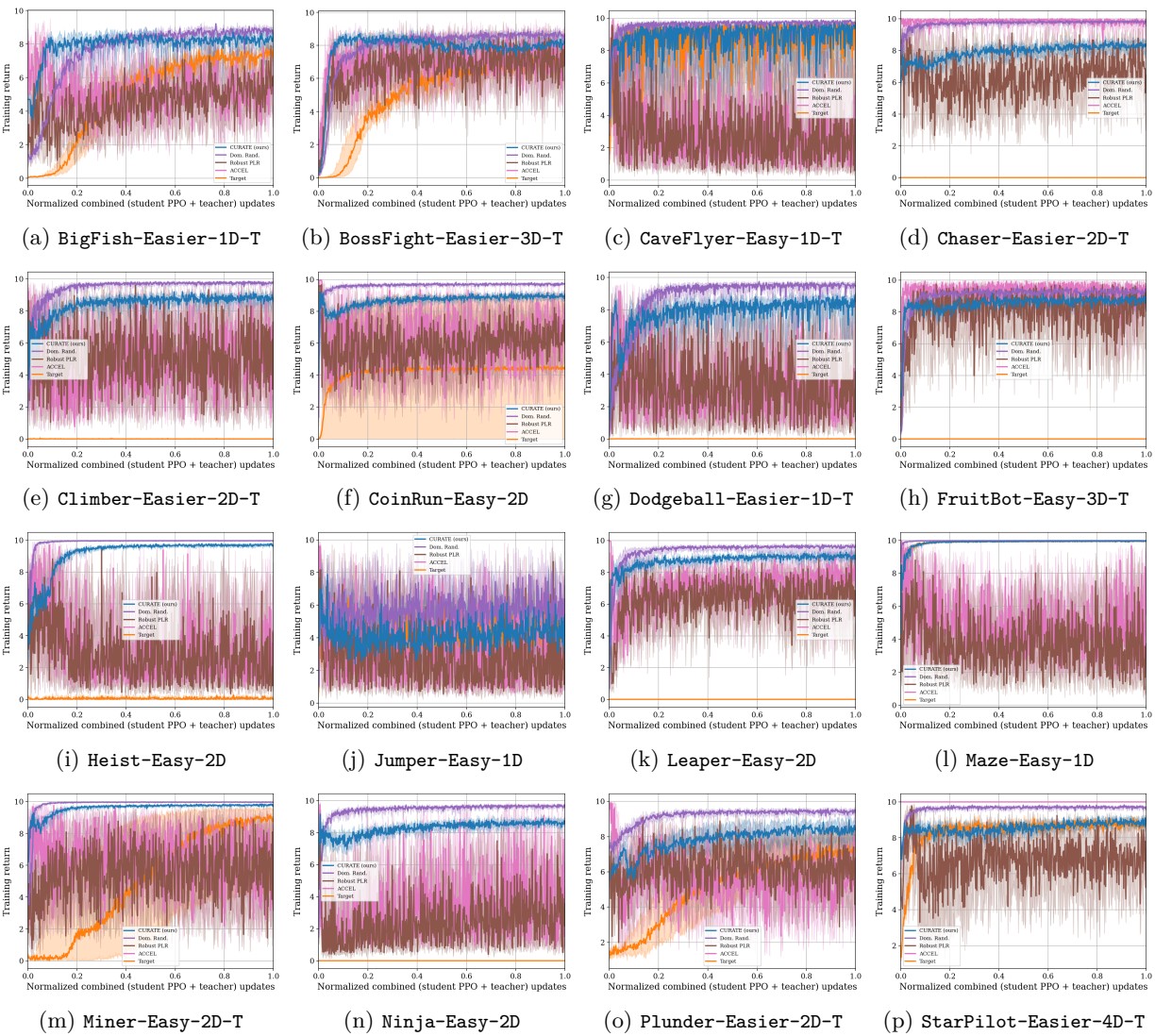

Figure 18: Learning curves in terms of training return for each of the 16 games in the Procgen Curriculum Suite. Curves are interpolated across methods and games with subsampling every 5 updates and shown in terms of IQM over six trials per game with 95% CI (Agarwal et al., 2021). Updates are normalized based on the allowable updates per game.

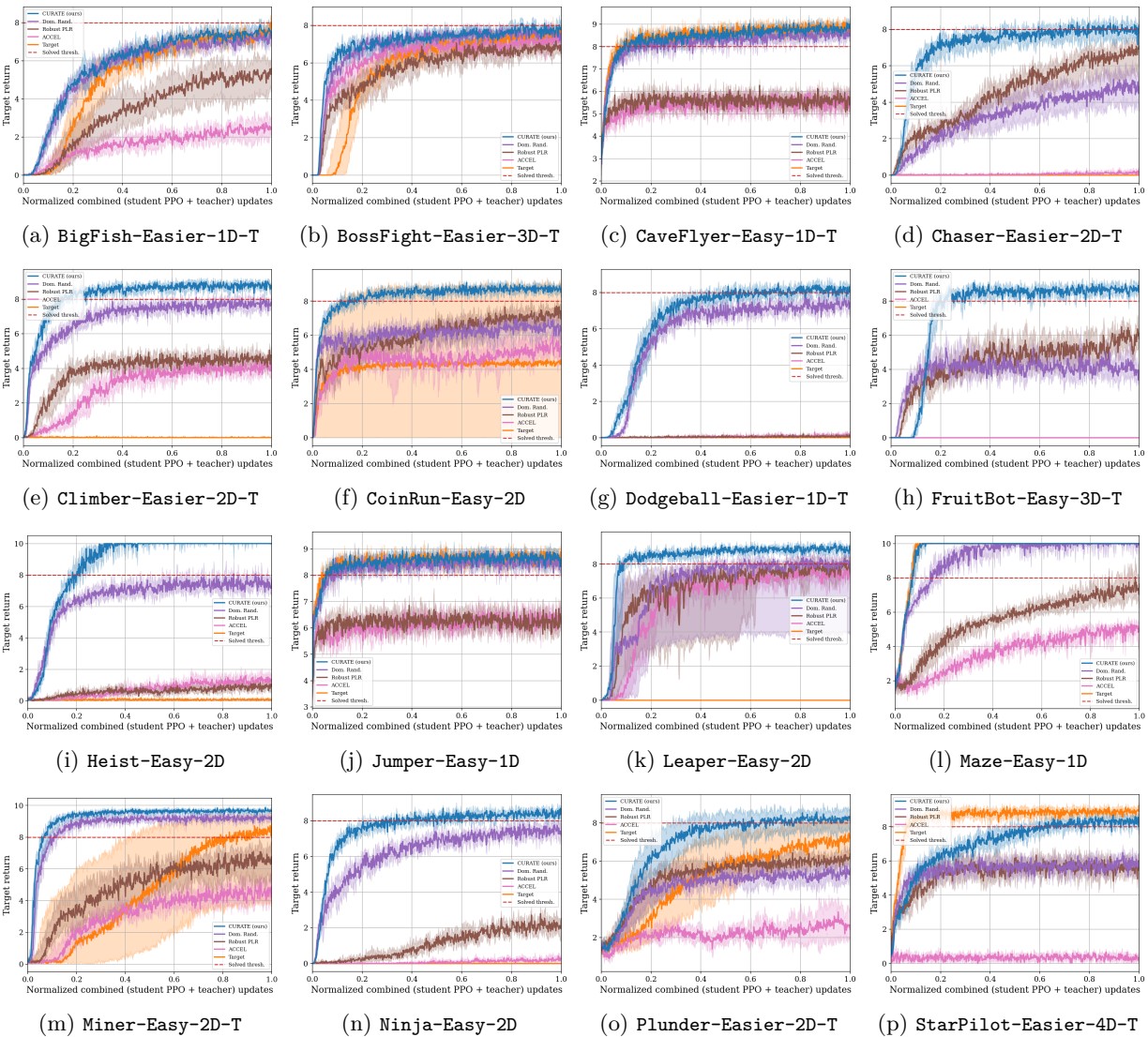

Figure 19: Learning curves in terms of target task return for each of the 16 games in the Procgen Curriculum Suite. Curves are interpolated across methods and games with subsampling every 5 updates and shown in terms of IQM over six trials per game with 95% CI (Agarwal et al., 2021). Updates are normalized based on the allowable updates per game.

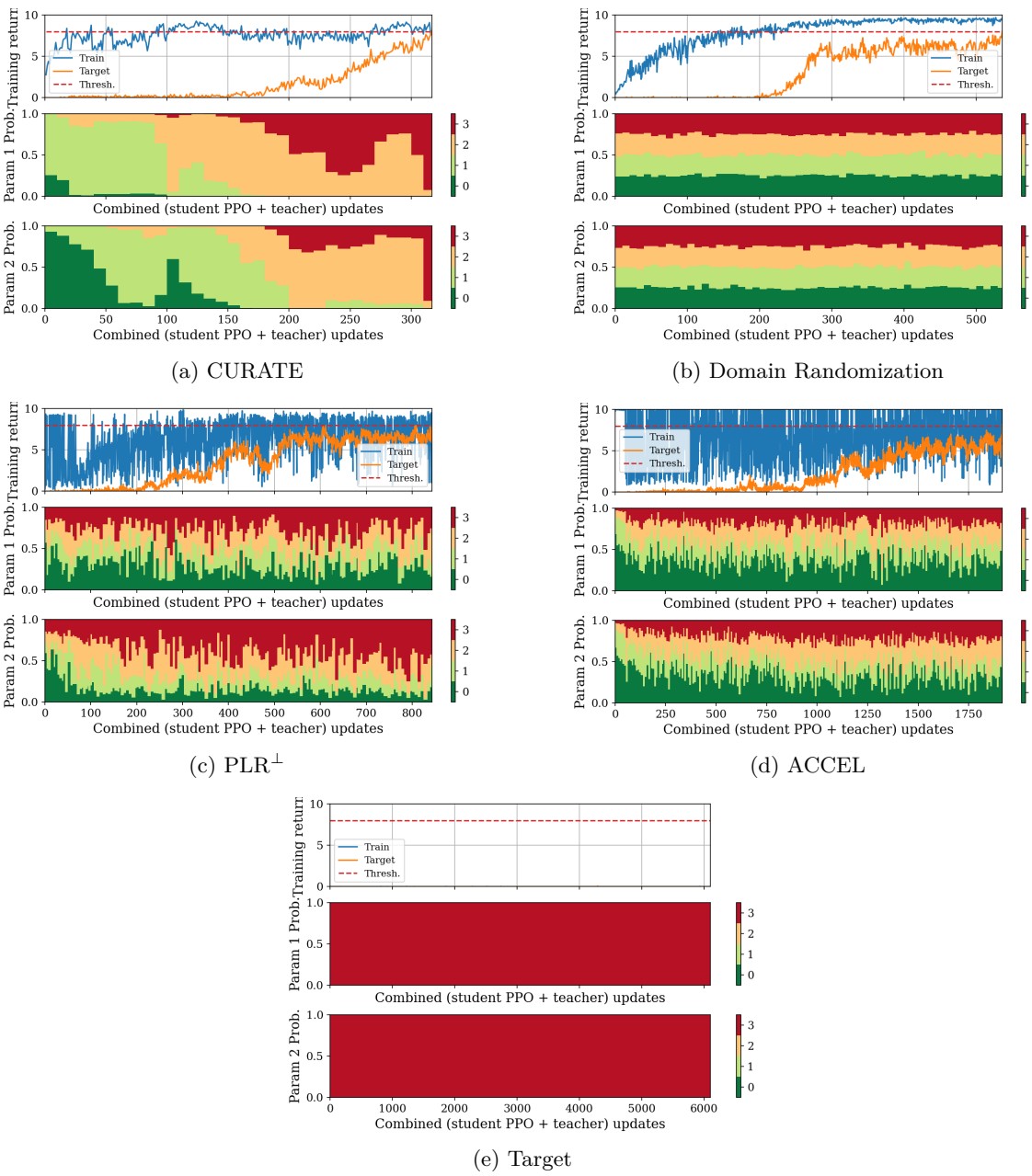

Figure 20: Representative curriculum learning time histories for each approach in PCS Leaper (`Leaper-Easy-2D`). Each time history shows the trial that is closest to the median performance of all 6 trials for each approach. The top figure shows the time history of the return, shown for the training environments and the target task. The bottom figures show the time history of the curriculum, with time average discretization of 10 updates to better show long-term trends.

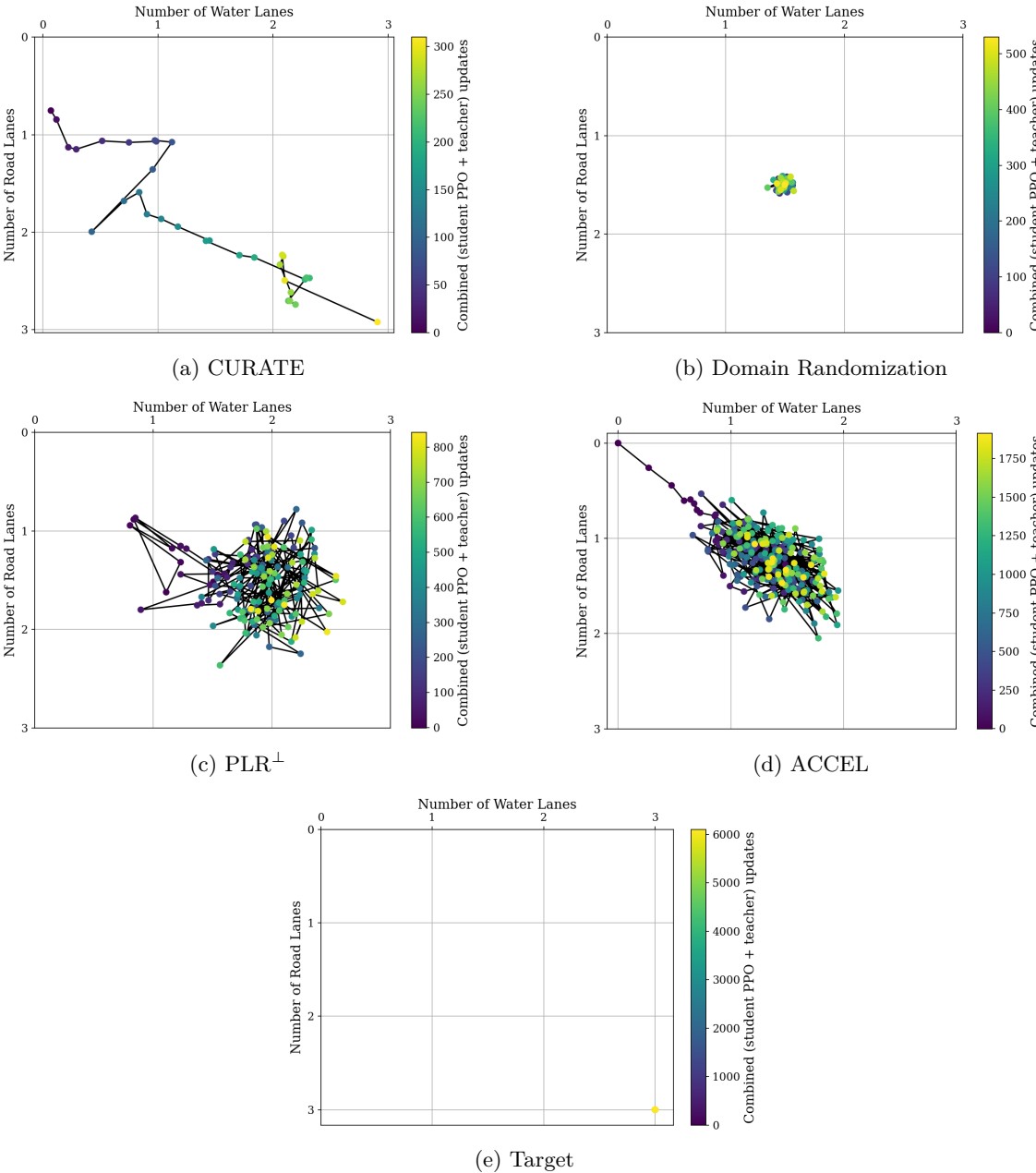

Figure 21: Representative curriculum for each approach in PCS Leaper (`Leaper-Easy-2D`) as represented by the mean environment parameters of the training tasks with time average discretization of 10 updates to better show long-term trends. The trial chosen for the curriculum is the closest to the median performance of all 6 trials of each approach. Note that the colorbar for each figure has a different maximum value.

Table 10: Target task performance results for `BigFish-Easier-1D-T` using at least $200\times10^6$ training steps for six trials. Updates are normalized to 508 updates. Summary statistics are shown in terms of success rate (if the target tasks $\mathcal{M}_{\theta_t}$ were solved), time to threshold in normalized updates until $\mathcal{M}_{\theta_t}$ is solved or maximum updates reached (TTUN), final training return (F. Train), and final target return (F. Target). Trials that do not solve the task still count towards summary statistics. TTUN, F. Train, and F. Target are shown in terms of IQM with 95% CI (Agarwal et al., 2021). The best approach for each metric is **bolded**, the second best approach is underlined, and an asterisk (*) denotes that the approach falls within the best approach's CI.

| Method | Success Rate | TTUN ↓ | TTUN CI | F. Train ↑ | F. Train CI | F. Target ↑ | F. Target CI |
|---|---|---|---|---|---|---|---|
| CURATE (ours) | **100.000%** (6) | **68.110%** | (52.854%, 85.039%) | 8.692* | (7.628, 9.053) | 7.305* | (6.992, 7.754) |
| Domain Rand. | **100.000%** (6) | 81.004%* | (72.293%, 91.880%) | **8.723** | (8.439, 9.041) | 7.148* | (6.602, 7.617) |
| Robust PLR | 0.000% (0) | 100.000% | (100.000%, 100.000%) | 5.600 | (3.303, 7.046) | 5.625 | (4.297, 6.250) |
| ACCEL | 0.000% (0) | 100.000% | (100.000%, 100.000%) | 4.926 | (2.274, 6.661) | 2.480 | (2.109, 3.066) |
| Target | **100.000%** (6) | 72.785%* | (59.301%, 83.957%) | 7.715 | (7.437, 7.911) | **7.656** | (7.148, 8.047) |

Table 11: Target task performance results for `BossFight-Easier-3D-T` using at least $200\times10^6$ training steps for six trials. Updates are normalized to 762 updates. Summary statistics are shown in terms of success rate (if the target tasks $\mathcal{M}_{\theta_t}$ were solved), time to threshold in normalized updates until $\mathcal{M}_{\theta_t}$ is solved or maximum updates reached (TTUN), final training return (F. Train), and final target return (F. Target). Trials that do not solve the task still count towards summary statistics. TTUN, F. Train, and F. Target are shown in terms of IQM with 95% CI (Agarwal et al., 2021). The best approach for each metric is **bolded**, the second best approach is underlined, and an asterisk (*) denotes that the approach falls within the best approach's CI.

| Method | Success Rate | TTUN ↓ | TTUN CI | F. Train ↑ | F. Train CI | F. Target ↑ | F. Target CI |
|---|---|---|---|---|---|---|---|
| CURATE (ours) | **100.000%** (6) | **27.756%** | (17.487%, 37.238%) | 8.119* | (7.771, 8.449) | **7.910** | (7.500, 8.320) |
| Domain Rand. | **100.000%** (6) | 30.840%* | (25.328%, 37.008%) | 8.371* | (8.132, 8.601) | 7.246 | (6.934, 7.852) |
| Robust PLR | 66.667% (4) | 95.571% | (78.904%, 99.114%) | **8.674** | (7.254, 9.443) | 6.504 | (6.211, 6.934) |
| ACCEL | 83.333% (5) | 59.219% | (41.503%, 86.680%) | 6.793 | (4.869, 8.543) | 7.227 | (6.484, 7.559) |
| Target | **100.000%** (6) | 54.298% | (44.226%, 78.478%) | 7.304* | (6.696, 7.792) | 7.441 | (7.227, 7.617) |

trial for evaluations, whereas Tab. 3 and Tab. 26 use 100 episodes per environment per trial for evaluations, so some discrepancies may exist due to the difference in sample size.

# I   Procgen Curriculum Suite

This work introduces the *Procgen Curriculum Suite*, an extension of Procgen (Cobbe et al., 2020) where each of the 16 games are assigned a structured task space $\Theta$ of varying dimensionality, from one-dimensional to four-dimensional (Tab. 27), such that the games are DOUPOMDPs. This task space definition is intended to 1) make games easier to learn with a curriculum and 2) promote benchmarking for advancing the field of curriculum learning by proposing standard task spaces. Moreover, defining these task spaces facilitates future work in learning these task spaces by comparing learned task spaces against these human-curated task spaces.

Under the hood, the procedure generation is changed by allowing causal interventions in specific parameters of interest. For example, the game of Leaper has a two-dimensional task space: 1) $\theta_1$: number of road lanes and 2) $\theta_2$: number of water lanes. Specifying that distributions of levels should be generated with 2 water lanes is done by setting $do(\theta_2 = 2)$ in the generation process. We note that for some games, these causal interventions are sparse and local. Thus, they are disentangled with the rest of the environment factors of variation. This allows for precise evaluation of causal counterfactuals, opening up new possibilities for future work. However, for other games, the environment generation process has some degree of entanglement.

Intuitively, the Procgen Curriculum Suite is conceptually similar to C-Procgen (Tan et al., 2024), which also exposes the parameters used for procedural generation. Although C-Procgen exposes significantly more parameters, it remains an open question for which parameters in particular are most useful or informative

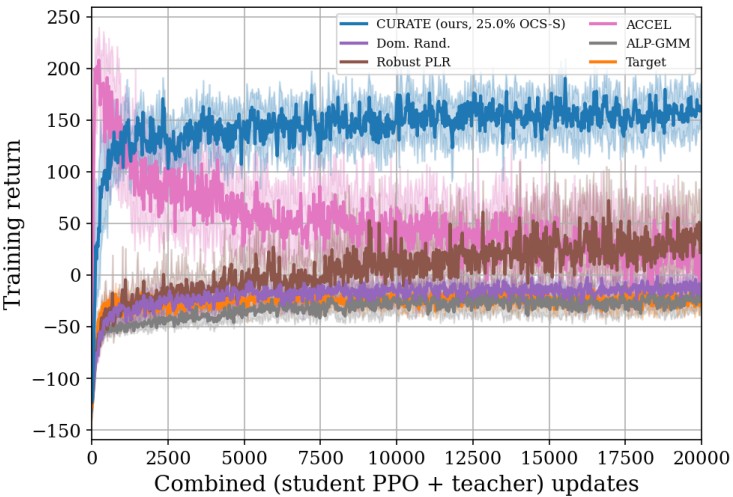

Figure 22: Learning curve in terms of training return for the BipedalWalker domain. Curve is interpolated across methods with subsampling every 25 updates and shown in terms of IQM over 10 trials with 95% CI (Agarwal et al., 2021).

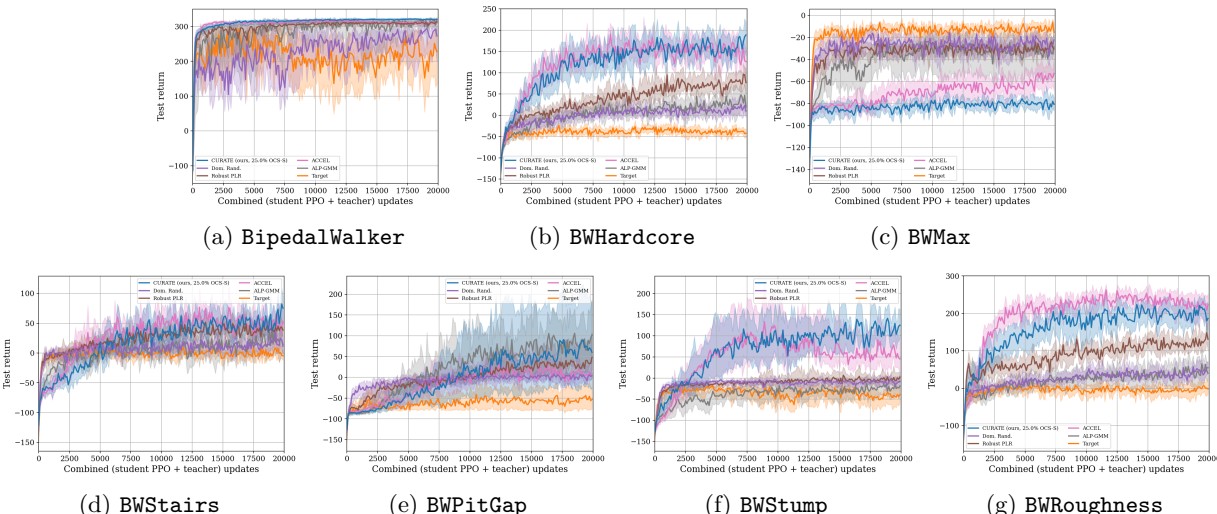

(a) `BipedalWalker`          (b) `BWHardcore`          (c) `BWMax`

(d) `BWStairs`     (e) `BWPitGap`     (f) `BWStump`     (g) `BWRoughness`

Figure 23: Learning curves in terms of test return for the seven test environments of interest in the BipedalWalker domain. The environments consist of two classic environments (`BipedalWalker`, `BipedalWalkerHardcore`); `BipedalWalkerMax`, the target environment $\mathcal{M}_{\theta_t}$ with maximum difficulty; and the four terrain challenges introduced in Parker-Holder et al. (2022a) (`BipedalWalkerStairs`, `BipedalWalkerPitGap`, `BipedalWalkerStump`, `BipedalWalkerRoughness`). Curves are interpolated across methods and shown in terms of IQM over 10 trials with 95% CI (Agarwal et al., 2021).

Table 12: Target task performance results for `CaveFlyer-Easy-1D-T` using at least $200\times10^6$ training steps for six trials. Updates are normalized to 3051 updates. Summary statistics are shown in terms of success rate (if the target tasks $\mathcal{M}_{\theta_t}$ were solved), time to threshold in normalized updates until $\mathcal{M}_{\theta_t}$ is solved or maximum updates reached (TTUN), final training return (F. Train), and final target return (F. Target). Trials that do not solve the task still count towards summary statistics. TTUN, F. Train, and F. Target are shown in terms of IQM with 95% CI (Agarwal et al., 2021). The best approach for each metric is **bolded**, the second best approach is underlined, and an asterisk (*) denotes that the approach falls within the best approach's CI.

| Method | Success Rate | TTUN ↓ | TTUN CI | F. Train ↑ | F. Train CI | F. Target ↑ | F. Target CI |
|---|---|---|---|---|---|---|---|
| CURATE (ours) | **100.000%** (6) | 5.293% | (4.769%, 5.851%) | 9.663* | (9.225, 9.790) | **8.828** | (8.711, 8.945) |
| Domain Rand. | **100.000%** (6) | 5.129% | (4.302%, 5.629%) | **9.701** | (9.324, 9.820) | 8.594 | (8.340, 8.770) |
| Robust PLR | 0.000% (0) | 100.000% | (100.000%, 100.000%) | 2.759 | (1.230, 5.836) | 5.723 | (5.312, 6.016) |
| ACCEL | 0.000% (0) | 100.000% | (100.000%, 100.000%) | 1.796 | (0.611, 4.824) | 5.508 | (5.098, 5.664) |
| Target | **100.000%** (6) | **4.425%** | (3.507%, 5.015%) | 8.335 | (6.112, 9.282) | 8.672 | (8.516, 8.906) |

Table 13: Target task performance results for `Chaser-Easier-2D-T` using at least $200\times10^6$ training steps for six trials. Updates are normalized to 3051 updates. Summary statistics are shown in terms of success rate (if the target tasks $\mathcal{M}_{\theta_t}$ were solved), time to threshold in normalized updates until $\mathcal{M}_{\theta_t}$ is solved or maximum updates reached (TTUN), final training return (F. Train), and final target return (F. Target). Trials that do not solve the task still count towards summary statistics. TTUN, F. Train, and F. Target are shown in terms of IQM with 95% CI (Agarwal et al., 2021). The best approach for each metric is **bolded**, the second best approach is underlined, and an asterisk (*) denotes that the approach falls within the best approach's CI.

| Method | Success Rate | TTUN ↓ | TTUN CI | F. Train ↑ | F. Train CI | F. Target ↑ | F. Target CI |
|---|---|---|---|---|---|---|---|
| CURATE (ours) | **100.000%** (6) | **18.355%** | (14.192%, 34.415%) | 8.308 | (7.734, 8.590) | **8.223** | (7.754, 8.594) |
| Domain Rand. | 0.000% (0) | 100.000% | (100.000%, 100.000%) | **9.857** | (9.728, 9.915) | 4.844 | (3.945, 5.664) |
| Robust PLR | 83.333% (5) | 89.684% | (80.039%, 96.788%) | 6.430 | (4.967, 8.207) | 7.070 | (5.898, 7.461) |
| ACCEL | 0.000% (0) | 100.000% | (100.000%, 100.000%) | 9.548 | (8.640, 9.959) | 0.137 | (0.039, 0.254) |
| Target | 0.000% (0) | 100.000% | (100.000%, 100.000%) | 0.000 | (0.000, 0.011) | 0.000 | (0.000, 0.000) |

to use for curriculum learning. Thus, we suggest that the Procgen Curriculum Suite is distinguished in this regard by the definition of task spaces, so that other approaches in curriculum learning can be benchmarked using these task spaces. However, the greater breadth of control that C-Procgen offers is also useful for future work, such as learning the task spaces. Thus, although an overlap exists in capabilities, both frameworks offer important contributions to the curriculum learning literature.

Table 28 summarizes the games by maximum episode length. These values were used as defined within the original Procgen code. When selecting the PPO rollout length for these games, note that unlike the evaluation of Procgen for Cobbe et al. (2020), we do not use a fixed episode length of 256 for all games. We instead select a nearest value to each game's maximum episode length, e.g., 512 for 500 step games, 1024 for 1000 step games, 4096 for 4000 step games, and 6144 for 6000 step games. Our rationale is that it is reasonable to allow an agent the entire episode length to solve the level and collect the level completion reward, and to not end the episode earlier than what the game code defines.

### I.1 Task space definition

Table 29 (1D), Tab. 30 (2D), Tab. 31 (3D), and Tab. 32 (4D) present the task space $\Theta$ for each game based on dimensionality, along with a description of the task space parameters. Note that games may have differing task spaces between the easy and hard distribution modes. Usually, only the minimum and maximum values change between distribution modes, but two games have differing dimensionality due to the introduction of new mechanics (FruitBot: locked gates, Plunder: panels).

We also found that even with the easy distribution modes, some games were too difficult to consistently solve the target tasks using current curriculum methods. Thus, for select games, we introduce an "easier"

Table 14: Target task performance results for `Climber-Easier-2D-T` using at least $200{\times}10^6$ training steps for six trials. Updates are normalized to 3051 updates. Summary statistics are shown in terms of success rate (if the target tasks $\mathcal{M}_{\theta_t}$ were solved), time to threshold in normalized updates until $\mathcal{M}_{\theta_t}$ is solved or maximum updates reached (TTUN), final training return (F. Train), and final target return (F. Target). Trials that do not solve the task still count towards summary statistics. TTUN, F. Train, and F. Target are shown in terms of IQM with 95% CI (Agarwal et al., 2021). The best approach for each metric is **bolded**, the second best approach is underlined, and an asterisk (*) denotes that the approach falls within the best approach's CI.

| Method | Success Rate | TTUN ↓ | TTUN CI | F. Train ↑ | F. Train CI | F. Target ↑ | F. Target CI |
|---|---|---|---|---|---|---|---|
| CURATE (ours) | **100.000%** (6) | **10.439%** | (9.276%, 12.053%) | 8.621 | (8.076, 8.963) | **8.652** | (8.555, 9.023) |
| Domain Rand. | **100.000%** (6) | 30.826% | (27.606%, 34.071%) | **9.840** | (9.734, 9.910) | 7.695 | (7.363, 7.949) |
| Robust PLR | 0.000% (0) | 100.000% | (100.000%, 100.000%) | 5.292 | (3.890, 7.665) | 4.688 | (4.590, 4.863) |
| ACCEL | 0.000% (0) | 100.000% | (100.000%, 100.000%) | 6.979 | (2.578, 9.566) | 3.867 | (3.398, 4.297) |
| Target | 0.000% (0) | 100.000% | (100.000%, 100.000%) | 0.009 | (0.000, 0.032) | 0.000 | (0.000, 0.000) |

Table 15: Target task performance results for `CoinRun-Easy-2D` using at least $200{\times}10^6$ training steps for six trials. Updates are normalized to 3051 updates. Summary statistics are shown in terms of success rate (if the target tasks $\mathcal{M}_{\theta_t}$ were solved), time to threshold in normalized updates until $\mathcal{M}_{\theta_t}$ is solved or maximum updates reached (TTUN), final training return (F. Train), and final target return (F. Target). Trials that do not solve the task still count towards summary statistics. TTUN, F. Train, and F. Target are shown in terms of IQM with 95% CI (Agarwal et al., 2021). The best approach for each metric is **bolded**, the second best approach is underlined, and an asterisk (*) denotes that the approach falls within the best approach's CI.

| Method | Success Rate | TTUN ↓ | TTUN CI | F. Train ↑ | F. Train CI | F. Target ↑ | F. Target CI |
|---|---|---|---|---|---|---|---|
| CURATE (ours) | **100.000%** (6) | **7.915%** | (6.957%, 8.473%) | 9.061 | (8.907, 9.273) | **8.730** | (8.281, 9.062) |
| Domain Rand. | 50.000% (3) | 91.667% | (68.346%, 100.000%) | **9.729** | (9.653, 9.823) | 6.445 | (6.035, 6.660) |
| Robust PLR | 83.333% (5) | 69.993% | (58.956%, 85.906%) | 6.008 | (4.053, 8.046) | 7.285 | (6.699, 7.969) |
| ACCEL | 0.000% (0) | 100.000% | (100.000%, 100.000%) | 6.733 | (4.207, 8.141) | 5.371 | (4.844, 5.703) |
| Target | 50.000% (3) | 53.909% | (6.096%, 100.000%) | 4.521 | (0.000, 9.109) | 4.375 | (0.000, 9.062) |

distribution mode, which uses the easy mode mechanics, but further reduces the task space. To select the task spaces for these easier modes, we start with the easier modes and systematically reduce the value of dimensions by binary search until the game becomes solvable by either CURATE, DR, or NC.

Task spaces were designed with several desiderata. First, task spaces were defined such that the hardest levels that occur at $\theta_t = \max(\Theta)$ can still be encountered in the standard version of Procgen. Put differently, the task spaces cannot yield levels that are more difficult than would have been generated normally for a particular distribution mode. Second, we designed the tasks spaces to have a diversity of dimensionality. For games where some options exist for which parameters should be used, we selected them based on balancing the overall dimensionality distribution. Third, some games, such as BigFish, had no obvious candidates for the parameters of the level procedural generation, so we occasionally define new sources of variation that can be controlled through the task space parameters. In the case of BigFish, we control the number of fish needed to solve the level. Fourth, we design the task space axes such that they are difficulty aligned, where increasing the value of the task space parameters leads to harder levels (in expectation). For cases where the variable in question in inversely proportional to difficulty, we reparameterize the variable such that it becomes proportional to difficulty. Fifth, in setting the lower bounds of the task spaces, we aimed to match the same distribution of easy levels that would have been encountered normally. Where possible, we avoided making the easiest levels trivially easy. There are some exceptions, e.g., StarPilot at very low finish line spawn times can be solved regardless of any agent action.

## I.2 Terminal reward mode

For the original 16 games, only six (CoinRun, Heist, Jumper, Leaper, Maze, Ninja) provide episode rewards only upon successfully completing a level. The 10 other games (BigFish, BossFight, CaveFlyer, Chaser,

Table 16: Target task performance results for `Dodgeball-Easier-1D-T` using at least $200 \times 10^6$ training steps for six trials. Updates are normalized to 3051 updates. Summary statistics are shown in terms of success rate (if the target tasks $\mathcal{M}_{\theta_t}$ were solved), time to threshold in normalized updates until $\mathcal{M}_{\theta_t}$ is solved or maximum updates reached (TTUN), final training return (F. Train), and final target return (F. Target). Trials that do not solve the task still count towards summary statistics. TTUN, F. Train, and F. Target are shown in terms of IQM with 95% CI (Agarwal et al., 2021). The best approach for each metric is **bolded**, the second best approach is underlined, and an asterisk (*) denotes that the approach falls within the best approach's CI.

| Method | Success Rate | TTUN ↓ | TTUN CI | F. Train ↑ | F. Train CI | F. Target ↑ | F. Target CI |
|---|---|---|---|---|---|---|---|
| CURATE (ours) | **100.000%** (6) | **26.450%** | (22.321%, 31.572%) | 8.496 | (7.162, 9.085) | **8.105** | (7.676, 8.516) |
| Domain Rand. | **100.000%** (6) | 40.216% | (37.455%, 43.478%) | **9.452** | (9.355, 9.574) | 7.422 | (6.738, 7.871) |
| Robust PLR | 0.000% (0) | 100.000% | (100.000%, 100.000%) | 1.102 | (0.603, 3.671) | 0.098 | (0.020, 0.234) |
| ACCEL | 0.000% (0) | 100.000% | (100.000%, 100.000%) | 2.610 | (0.964, 6.050) | 0.137 | (0.098, 0.156) |
| Target | 0.000% (0) | 100.000% | (100.000%, 100.000%) | 0.009 | (0.000, 0.024) | 0.000 | (0.000, 0.000) |

Table 17: Target task performance results for `FruitBot-Easy-3D-T` using at least $200 \times 10^6$ training steps for six trials. Updates are normalized to 3051 updates. Summary statistics are shown in terms of success rate (if the target tasks $\mathcal{M}_{\theta_t}$ were solved), time to threshold in normalized updates until $\mathcal{M}_{\theta_t}$ is solved or maximum updates reached (TTUN), final training return (F. Train), and final target return (F. Target). Trials that do not solve the task still count towards summary statistics. TTUN, F. Train, and F. Target are shown in terms of IQM with 95% CI (Agarwal et al., 2021). The best approach for each metric is **bolded**, the second best approach is underlined, and an asterisk (*) denotes that the approach falls within the best approach's CI.

| Method | Success Rate | TTUN ↓ | TTUN CI | F. Train ↑ | F. Train CI | F. Target ↑ | F. Target CI |
|---|---|---|---|---|---|---|---|
| CURATE (ours) | **100.000%** (6) | **16.822%** | (15.978%, 18.420%) | 8.777 | (8.501, 9.173) | **8.652** | (7.891, 8.945) |
| Domain Rand. | 0.000% (0) | 100.000% | (100.000%, 100.000%) | **9.318** | (9.044, 9.502) | 3.965 | (3.711, 4.258) |
| Robust PLR | 33.333% (2) | 97.583% | (76.278%, 100.000%) | 8.063 | (5.385, 9.578) | 5.781 | (5.039, 6.211) |
| ACCEL | 0.000% (0) | 100.000% | (100.000%, 100.000%) | 9.035 | (8.602, 9.597) | 0.000 | (0.000, 0.000) |
| Target | 0.000% (0) | 100.000% | (100.000%, 100.000%) | 0.000 | (0.000, 0.000) | 0.000 | (0.000, 0.000) |

Climber, Dodgeball, FruitBot, Miner, Plunder, StarPilot) provide intermediate feedback of varying frequency. Some feedback can still be quite sparse (e.g., destroying targets in CaveFlyer) or relatively dense (e.g., collecting orbs in Chaser). This intermediate feedback may be sufficiently informative such that exploration is no longer a concern, and the agent can train directly in the target tasks without a curricula. Therefore, for these 10 games, we introduce a *terminal reward mode*, which "sparsifies" these games such that a reward is only provided upon successfully completing the level. To avoid confusion, we append the word "Terminal" when referring to the terminal reward version of a game (e.g., Miner Terminal), or simply "-T" for brevity (e.g., Miner-T).

Generally, all other game mechanics are unchanged when in terminal reward mode, with the exception of FruitBot. Normally, in FruitBot, eating a fruit yields +1 reward, and eating a food item that isn't fruit yields a −4 penalty. An agent that is maximizing the sum of discounted returns would learn to eat fruit and avoid non-fruit. However, for FruitBot Terminal, the agent should only receive the completion bonus upon reaching the end of the level, regardless of the fruit consumed. Therefore, to preserve the incentive structure in FruitBot Terminal to eat fruit and avoid non-fruit, we introduce a health point system. The agent starts with three health points that are displayed in the top-right of the observation frame so that this information is observable to the agent. Eating a fruit adds a health point, up to a maximum of three. Eating a food item that isn't fruit removes a health point. If the agent reaches zero health points, the episode ends.

Table 18: Target task performance results for `Heist-Easy-2D` using at least $200 \times 10^6$ training steps for six trials. Updates are normalized to 3051 updates. Summary statistics are shown in terms of success rate (if the target tasks $\mathcal{M}_{\theta_t}$ were solved), time to threshold in normalized updates until $\mathcal{M}_{\theta_t}$ is solved or maximum updates reached (TTUN), final training return (F. Train), and final target return (F. Target). Trials that do not solve the task still count towards summary statistics. TTUN, F. Train, and F. Target are shown in terms of IQM with 95% CI (Agarwal et al., 2021). The best approach for each metric is **bolded**, the second best approach is underlined, and an asterisk (*) denotes that the approach falls within the best approach's CI.

| Method | Success Rate | TTUN ↓ | TTUN CI | F. Train ↑ | F. Train CI | F. Target ↑ | F. Target CI |
|---|---|---|---|---|---|---|---|
| CURATE (ours) | **100.000%** (6) | **15.085%** | (13.938%, 16.929%) | 9.662 | (9.590, 9.740) | **10.000** | (10.000, 10.000) |
| Domain Rand. | **100.000%** (6) | 30.719% | (28.761%, 31.990%) | **9.974** | (9.949, 9.984) | 7.656 | (7.012, 8.906) |
| Robust PLR | 0.000% (0) | 100.000% | (100.000%, 100.000%) | 2.084 | (0.587, 5.036) | 0.898 | (0.801, 1.035) |
| ACCEL | 0.000% (0) | 100.000% | (100.000%, 100.000%) | 1.258 | (0.833, 3.167) | 1.230 | (1.074, 1.445) |
| Target | 0.000% (0) | 100.000% | (100.000%, 100.000%) | 0.000 | (0.000, 0.000) | 0.098 | (0.020, 0.156) |

Table 19: Target task performance results for `Jumper-Easy-1D` using at least $200 \times 10^6$ training steps for six trials. Updates are normalized to 3051 updates. Summary statistics are shown in terms of success rate (if the target tasks $\mathcal{M}_{\theta_t}$ were solved), time to threshold in normalized updates until $\mathcal{M}_{\theta_t}$ is solved or maximum updates reached (TTUN), final training return (F. Train), and final target return (F. Target). Trials that do not solve the task still count towards summary statistics. TTUN, F. Train, and F. Target are shown in terms of IQM with 95% CI (Agarwal et al., 2021). The best approach for each metric is **bolded**, the second best approach is underlined, and an asterisk (*) denotes that the approach falls within the best approach's CI.

| Method | Success Rate | TTUN ↓ | TTUN CI | F. Train ↑ | F. Train CI | F. Target ↑ | F. Target CI |
|---|---|---|---|---|---|---|---|
| CURATE (ours) | **100.000%** (6) | 3.532% | (2.852%, 4.236%) | **5.444** | (4.093, 7.065) | 8.594* | (8.203, 8.809) |
| Domain Rand. | **100.000%** (6) | 3.663% | (2.524%, 4.171%) | 5.155* | (4.032, 7.714) | **8.633** | (8.418, 8.926) |
| Robust PLR | 0.000% (0) | 100.000% | (100.000%, 100.000%) | 3.062 | (0.925, 6.944) | 6.152 | (5.723, 6.680) |
| ACCEL | 0.000% (0) | 100.000% | (100.000%, 100.000%) | 2.081 | (0.677, 4.995) | 6.328 | (5.859, 6.875) |
| Target | **100.000%** (6) | **2.417%** | (1.704%, 2.942%) | 4.978* | (3.768, 6.064) | 8.438* | (8.223, 8.613) |

Table 20: Target task performance results for `Leaper-Easy-2D` using at least $200 \times 10^6$ training steps for six trials. Updates are normalized to 6103 updates. Summary statistics are shown in terms of success rate (if the target tasks $\mathcal{M}_{\theta_t}$ were solved), time to threshold in normalized updates until $\mathcal{M}_{\theta_t}$ is solved or maximum updates reached (TTUN), final training return (F. Train), and final target return (F. Target). Trials that do not solve the task still count towards summary statistics. TTUN, F. Train, and F. Target are shown in terms of IQM with 95% CI (Agarwal et al., 2021). The best approach for each metric is **bolded**, the second best approach is underlined, and an asterisk (*) denotes that the approach falls within the best approach's CI.

| Method | Success Rate | TTUN ↓ | TTUN CI | F. Train ↑ | F. Train CI | F. Target ↑ | F. Target CI |
|---|---|---|---|---|---|---|---|
| CURATE (ours) | **100.000%** (6) | **6.128%** | (4.993%, 7.935%) | 8.807 | (8.598, 9.048) | **8.867** | (8.672, 8.965) |
| Domain Rand. | 83.333% (5) | 14.309% | (7.791%, 58.807%) | **9.588** | (9.308, 9.703) | 8.008 | (3.945, 8.281) |
| Robust PLR | **100.000%** (6) | 15.709% | (11.052%, 39.927%) | 6.770 | (3.951, 8.477) | 7.500 | (6.875, 8.047) |
| ACCEL | **100.000%** (6) | 30.362% | (24.807%, 43.380%) | 8.191 | (7.079, 9.048) | 7.676 | (7.246, 8.027) |
| Target | 0.000% (0) | 100.000% | (100.000%, 100.000%) | 0.000 | (0.000, 0.000) | 0.000 | (0.000, 0.000) |

Table 21: Target task performance results for `Maze-Easy-1D` using at least $200 \times 10^6$ training steps for six trials. Updates are normalized to 6103 updates. Summary statistics are shown in terms of success rate (if the target tasks $\mathcal{M}_{\theta_t}$ were solved), time to threshold in normalized updates until $\mathcal{M}_{\theta_t}$ is solved or maximum updates reached (TTUN), final training return (F. Train), and final target return (F. Target). Trials that do not solve the task still count towards summary statistics. TTUN, F. Train, and F. Target are shown in terms of IQM with 95% CI (Agarwal et al., 2021). The best approach for each metric is **bolded**, the second best approach is underlined, and an asterisk (*) denotes that the approach falls within the best approach's CI.

| Method | Success Rate | TTUN ↓ | TTUN CI | F. Train ↑ | F. Train CI | F. Target ↑ | F. Target CI |
|---|---|---|---|---|---|---|---|
| CURATE (ours) | **100.000%** (6) | 6.300% | (5.678%, 6.968%) | 9.948 | (9.934, 9.960) | **10.000** | (10.000, 10.000) |
| Domain Rand. | **100.000%** (6) | 9.684% | (8.914%, 10.163%) | **9.987** | (9.981, 9.993) | **10.000** | (10.000, 10.000) |
| Robust PLR | **100.000%** (6) | 67.819% | (55.989%, 80.215%) | 3.712 | (1.046, 8.022) | 7.656 | (6.953, 8.145) |
| ACCEL | 0.000% (0) | 100.000% | (100.000%, 100.000%) | 5.466 | (1.851, 9.472) | 5.117 | (4.961, 5.430) |
| Target | **100.000%** (6) | **5.747%** | (5.366%, 5.997%) | 9.946 | (9.900, 9.962) | **10.000** | (10.000, 10.000) |

Table 22: Target task performance results for `Miner-Easy-2D-T` using at least $200 \times 10^6$ training steps for six trials. Updates are normalized to 3051 updates. Summary statistics are shown in terms of success rate (if the target tasks $\mathcal{M}_{\theta_t}$ were solved), time to threshold in normalized updates until $\mathcal{M}_{\theta_t}$ is solved or maximum updates reached (TTUN), final training return (F. Train), and final target return (F. Target). Trials that do not solve the task still count towards summary statistics. TTUN, F. Train, and F. Target are shown in terms of IQM with 95% CI (Agarwal et al., 2021). The best approach for each metric is **bolded**, the second best approach is underlined, and an asterisk (*) denotes that the approach falls within the best approach's CI.

| Method | Success Rate | TTUN ↓ | TTUN CI | F. Train ↑ | F. Train CI | F. Target ↑ | F. Target CI |
|---|---|---|---|---|---|---|---|
| CURATE (ours) | **100.000%** (6) | **5.072%** | (4.695%, 6.817%) | 9.715 | (9.645, 9.793) | **9.668** | (9.434, 9.766) |
| Domain Rand. | **100.000%** (6) | 8.325% | (7.604%, 8.940%) | **9.964** | (9.950, 9.976) | 9.297 | (9.082, 9.492) |
| Robust PLR | 66.667% (4) | 78.597% | (51.442%, 95.657%) | 3.170 | (1.446, 5.718) | 6.621 | (5.820, 7.344) |
| ACCEL | 0.000% (0) | 100.000% | (100.000%, 100.000%) | 8.520 | (6.531, 9.683) | 4.902 | (3.945, 5.879) |
| Target | 83.333% (5) | 63.684% | (26.409%, 92.593%) | 8.889 | (4.216, 9.564) | 8.828 | (4.336, 9.375) |

Table 23: Target task performance results for `Ninja-Easy-2D` using at least $200 \times 10^6$ training steps for six trials. Updates are normalized to 3051 updates. Summary statistics are shown in terms of success rate (if the target tasks $\mathcal{M}_{\theta_t}$ were solved), time to threshold in normalized updates until $\mathcal{M}_{\theta_t}$ is solved or maximum updates reached (TTUN), final training return (F. Train), and final target return (F. Target). Trials that do not solve the task still count towards summary statistics. TTUN, F. Train, and F. Target are shown in terms of IQM with 95% CI (Agarwal et al., 2021). The best approach for each metric is **bolded**, the second best approach is underlined, and an asterisk (*) denotes that the approach falls within the best approach's CI.

| Method | Success Rate | TTUN ↓ | TTUN CI | F. Train ↑ | F. Train CI | F. Target ↑ | F. Target CI |
|---|---|---|---|---|---|---|---|
| CURATE (ours) | **100.000%** (6) | **13.471%** | (11.382%, 14.397%) | 8.552 | (8.137, 8.845) | **8.555** | (8.164, 8.828) |
| Domain Rand. | **100.000%** (6) | 51.016% | (48.296%, 55.965%) | **9.615** | (9.505, 9.724) | 7.188 | (6.641, 7.754) |
| Robust PLR | 0.000% (0) | 100.000% | (100.000%, 100.000%) | 3.557 | (0.723, 6.183) | 2.500 | (1.914, 3.086) |
| ACCEL | 0.000% (0) | 100.000% | (100.000%, 100.000%) | 4.513 | (1.614, 7.924) | 0.234 | (0.098, 0.430) |
| Target | 0.000% (0) | 100.000% | (100.000%, 100.000%) | 0.000 | (0.000, 0.000) | 0.000 | (0.000, 0.000) |

Table 24: Target task performance results for `Plunder-Easier-2D-T` using at least $200\times10^6$ training steps for six trials. Updates are normalized to 762 updates. Summary statistics are shown in terms of success rate (if the target tasks $\mathcal{M}_{\theta_t}$ were solved), time to threshold in normalized updates until $\mathcal{M}_{\theta_t}$ is solved or maximum updates reached (TTUN), final training return (F. Train), and final target return (F. Target). Trials that do not solve the task still count towards summary statistics. TTUN, F. Train, and F. Target are shown in terms of IQM with 95% CI (Agarwal et al., 2021). The best approach for each metric is **bolded**, the second best approach is underlined, and an asterisk (*) denotes that the approach falls within the best approach's CI.

| Method | Success Rate | TTUN ↓ | TTUN CI | F. Train ↑ | F. Train CI | F. Target ↑ | F. Target CI |
|---|---|---|---|---|---|---|---|
| CURATE (ours) | **100.000%** (6) | **29.199%** | (20.079%, 39.895%) | 8.390 | (7.447, 8.891) | **8.105** | (7.656, 8.848) |
| Domain Rand. | 0.000% (0) | 100.000% | (100.000%, 100.000%) | **9.407** | (9.220, 9.601) | 5.586 | (4.609, 5.938) |
| Robust PLR | 0.000% (0) | 100.000% | (100.000%, 100.000%) | 6.807 | (4.539, 8.837) | 6.387 | (5.859, 6.895) |
| ACCEL | 0.000% (0) | 100.000% | (100.000%, 100.000%) | 2.931 | (1.747, 4.096) | 2.441 | (2.031, 2.910) |
| Target | 50.000% (3) | 86.417% | (53.084%, 100.000%) | 7.267 | (6.068, 8.390) | 7.168 | (5.918, 8.438) |

Table 25: Target task performance results for `StarPilot-Easier-4D-T` using at least $200\times10^6$ training steps for six trials. Updates are normalized to 3051 updates. Summary statistics are shown in terms of success rate (if the target tasks $\mathcal{M}_{\theta_t}$ were solved), time to threshold in normalized updates until $\mathcal{M}_{\theta_t}$ is solved or maximum updates reached (TTUN), final training return (F. Train), and final target return (F. Target). Trials that do not solve the task still count towards summary statistics. TTUN, F. Train, and F. Target are shown in terms of IQM with 95% CI (Agarwal et al., 2021). The best approach for each metric is **bolded**, the second best approach is underlined, and an asterisk (*) denotes that the approach falls within the best approach's CI.

| Method | Success Rate | TTUN ↓ | TTUN CI | F. Train ↑ | F. Train CI | F. Target ↑ | F. Target CI |
|---|---|---|---|---|---|---|---|
| CURATE (ours) | **100.000%** (6) | 31.768% | (27.073%, 38.651%) | 9.033 | (8.956, 9.178) | 8.418 | (7.891, 8.711) |
| Domain Rand. | 0.000% (0) | 100.000% | (100.000%, 100.000%) | 9.569 | (9.477, 9.640) | 5.664 | (5.352, 5.801) |
| Robust PLR | 16.667% (1) | 100.000% | (71.501%, 100.000%) | 6.683 | (5.175, 7.969) | 5.605 | (4.688, 6.328) |
| ACCEL | 0.000% (0) | 100.000% | (100.000%, 100.000%) | **10.000** | (10.000, 10.000) | 0.371 | (0.156, 0.605) |
| Target | **100.000%** (6) | **5.302%** | (5.138%, 6.293%) | 8.745 | (8.627, 8.798) | **9.219** | (9.082, 9.414) |

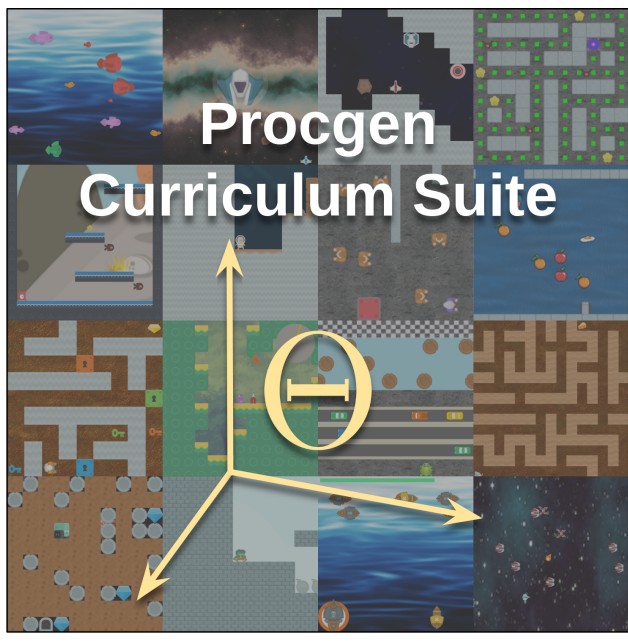

Figure 24: The Procgen Curriculum Suite is a variation of Procgen (Cobbe et al., 2020), where a structured task space is defined for each of the 16 games.

Table 26: Test performance for the BipedalWalker domain in terms of seven environments of interest. Tests were conducted at 20,000 student updates across 10 trials and 100 episodes per environment per trial. Individual scores of each environment are shown as well as an aggregate score of all seven environments. The environments consist of two classic environments (`BipedalWalker`, `BipedalWalkerHardcore`); `BipedalWalkerMax`, the target environment $\mathcal{M}_{\theta_t}$ with maximum difficulty; and the four terrain challenges introduced in Parker-Holder et al. (2022a) (`BipedalWalkerStairs`, `BipedalWalkerPitGap`, `BipedalWalkerStump`, `BipedalWalkerRoughness`). Scores are shown in terms of IQM with 95% CI (Agarwal et al., 2021). CI-L is lower CI bound, and CI-U is upper CI bound. The best approach for each metric is **bolded**, the second best approach is underlined, and an asterisk (*) denotes that the approach falls within the best approach's CI.

| Method | | BW | BWHardcore | BWMax | BWStairs | BWPitGap | BWStump | BWRoughness | **Aggregate** |
|---|---|---|---|---|---|---|---|---|---|
| CURATE, | IQM | 320.266 | **192.681** | -86.413 | **41.399** | **56.949** | **123.075** | 237.478* | **132.177** |
| 25% OCS-S (ours) | CI-L | 320.016 | 178.037 | -88.115 | 35.306 | 40.308 | 110.369 | 225.784 | 127.392 |
| | CI-U | 320.528 | 206.835 | -84.501 | 47.879 | 74.238 | 135.974 | 248.666 | 136.960 |
| | IQM | 296.175 | 8.654 | -19.791 | 11.362 | 3.207 | -6.919 | 37.547 | 10.340 |
| Domain Rand. | CI-L | 290.305 | 6.068 | -22.394 | 8.665 | 2.255 | -7.721 | 34.173 | 9.493 |
| | CI-U | 298.711 | 11.329 | -17.494 | 14.144 | 4.351 | -6.114 | 41.142 | 11.210 |
| | IQM | 313.420 | 63.925 | -24.568 | 26.398 | 23.750 | -5.512 | 119.825 | 42.818 |
| Robust PLR | CI-L | 313.077 | 56.112 | -25.948 | 22.304 | 20.304 | -7.077 | 110.354 | 40.588 |
| | CI-U | 313.751 | 72.264 | -23.230 | 30.741 | 27.518 | -3.850 | 129.805 | 45.158 |
| | IQM | 312.717 | 136.334 | -55.408 | 27.088 | -10.364 | 19.389 | **244.894** | 75.051 |
| ACCEL | CI-L | 312.465 | 122.113 | -58.257 | 22.049 | -13.685 | 14.715 | 234.461 | 71.187 |
| | CI-U | 312.972 | 150.401 | -52.599 | 32.323 | -7.015 | 24.524 | 254.662 | 78.942 |
| | IQM | **323.939** | 12.139 | -30.025 | 8.767 | 49.605* | -16.855 | 28.119 | 16.573 |
| ALP-GMM | CI-L | 323.475 | 7.856 | -33.394 | 5.596 | 42.154 | -19.385 | 24.115 | 14.888 |
| | CI-U | 324.386 | 16.643 | -27.074 | 12.083 | 57.799 | -14.767 | 32.252 | 18.310 |
| | IQM | 270.901 | -29.688 | **-8.889** | -0.472 | -48.939 | -34.253 | 5.614 | -9.691 |
| Target | CI-L | 262.808 | -33.243 | -9.849 | -2.061 | -53.509 | -38.045 | 3.242 | -10.554 |
| | CI-U | 277.803 | -26.269 | -7.910 | 1.129 | -44.374 | -30.483 | 7.836 | -8.863 |

Table 27: Procgen Curriculum Suite games by task space dimensionality.

| $|\Theta| = 1$ | $|\Theta| = 2$ | $|\Theta| = 3$ | $|\Theta| = 4$ |
|---|---|---|---|
| BigFish | Chaser | BossFight | FruitBot (hard) |
| CaveFlyer | Climber | FruitBot (easy) | StarPilot |
| Dodgeball | CoinRun | Plunder (hard) | |
| Jumper | Heist | | |
| Maze | Leaper | | |
| | Miner | | |
| | Ninja | | |
| | Plunder (easy) | | |

Table 28: Procgen Curriculum Suite games by maximum episode length. A majority (11) of games have a maximum episode length $T$ of 1000. Two games each use $T = 500$ and $T = 4000$. The only game with $T = 6000$ is BigFish.

| $T = 500$ | $T = 1000$ | $T = 4000$ | $T = 6000$ |
|---|---|---|---|
| Maze | CaveFlyer | BossFight | BigFish |
| Leaper | Chaser | Plunder | |
| | Climber | | |
| | CoinRun | | |
| | Dodgeball | | |
| | FruitBot | | |
| | Heist | | |
| | Jumper | | |
| | Miner | | |
| | Ninja | | |
| | StarPilot | | |

Table 29: Definition of task spaces for games in the Procgen Curriculum Suite that have a one-dimensional task space ($|\Theta| = 1$). E $\downarrow$ and E $\uparrow$ mean the lower and upper bound respectively in easy mode, and H $\downarrow$ and H $\uparrow$ mean the lower and upper bound respectively in hard mode. Games with "easier" modes (easy mode mechanics, but easier task spaces) have these parameters indicated in parentheses.

| | E $\downarrow$ | E $\uparrow$ | H $\downarrow$ | H $\uparrow$ | Description |
|---|---|---|---|---|---|
| | | | **BigFish** | | |
| $\theta_1$ | 1 | 30 (10) | 1 | 30 | Number of fish needed to be eaten in order to complete the level. |
| | | | **CaveFlyer** | | |
| $\theta_1$ | 0 | 3 | 0 | 3 | Number of objects per chunk. The first objects are asteroids, the second objects are targets, and the third objects are enemy spaceships. |
| | | | **Dodgeball** | | |
| $\theta_1$ | 3 (0) | 6 (3) | 3 | 6 | Number of enemies. |
| | | | **Jumper** | | |
| $\theta_1$ | 0 | 20 | 0 | 20 | Spike spawn probability in percentage. |
| | | | **Maze** | | |
| $\theta_1$ | 0 | 6 | 0 | 11 | Size of the maze. |

Table 30: Definition of task spaces for games in the Procgen Curriculum Suite that have a two-dimensional task space ($|\Theta| = 2$). E $\downarrow$ and E $\uparrow$ mean the lower and upper bound respectively in easy mode, and H $\downarrow$ and H $\uparrow$ mean the lower and upper bound respectively in hard mode. Games with "easier" modes (easy mode mechanics, but easier task spaces) have these parameters indicated in parentheses.

| | E $\downarrow$ | E $\uparrow$ | H $\downarrow$ | H $\uparrow$ | Description |
|---|---|---|---|---|---|
| | | | **Chaser** | | |
| $\theta_1$ | 0 | 3 | 0 | 3 | Number of enemies. |
| $\theta_2$ | 1 | 100 (75) | 1 | 100 | Number of orbs to collect before the level is complete. |
| | | | **Climber** | | |
| $\theta_1$ | 1 | 10 (5) | 1 | 10 | Number of platforms. |
| $\theta_2$ | 0 | 20 | 0 | 50 | Enemy spawn probability in percentage. |
| | | | **CoinRun** | | |
| $\theta_1$ | 1 | 3 | 1 | 3 | Level difficulty. |
| $\theta_2$ | 1 | 5 | 1 | 5 | Number of sections within a level. |
| | | | **Heist** | | |
| $\theta_1$ | 0 | 2 | 0 | 4 | Level difficulty. |
| $\theta_2$ | 0 | 3 | 0 | 3 | Number of locks/keys that prevent agent progress. |
| | | | **Leaper** | | |
| $\theta_1$ | 0 | 3 | 0 | 5 | Number of road lanes. |
| $\theta_2$ | 0 | 3 | 0 | 5 | Number of water lanes. |
| | | | **Miner** | | |
| $\theta_1$ | 0 | 3 | 0 | 12 | Number of diamonds. |
| $\theta_2$ | 0 | 20 | 0 | 80 | Number of boulders. |
| | | | **Ninja** | | |
| $\theta_1$ | 1 | 3 | 1 | 3 | Level difficulty. |
| $\theta_2$ | 1 | 5 | 1 | 5 | Number of sections within a level. |
| | | | **Plunder Easy** | | |
| $\theta_1$ | 1 | 20 (8) | - | - | Number of ships to defeat to complete the level. |
| $\theta_2$ | 1 | 10 | - | - | Juice penalty when defeating a friendly ship. |

Table 31: Definition of task spaces for games in the Procgen Curriculum Suite that have a three-dimensional task space ($|\Theta| = 3$). E $\downarrow$ and E $\uparrow$ mean the lower and upper bound respectively in easy mode, and H $\downarrow$ and H $\uparrow$ mean the lower and upper bound respectively in hard mode. Games with "easier" modes (easy mode mechanics, but easier task spaces) have these parameters indicated in parentheses.

| | E $\downarrow$ | E $\uparrow$ | H $\downarrow$ | H $\uparrow$ | Description |
|---|---|---|---|---|---|
| | | | **BossFight** | | |
| $\theta_1$ | 1 | 9 (2) | 1 | 9 | Health points of the boss per round. |
| $\theta_2$ | 1 | 5 (2) | 1 | 5 | Number of rounds for the fight. |
| $\theta_3$ | 2 | 3 | 2 | 5 | Duration of invulnerability at the beginning of each round. |
| | | | **FruitBot Easy** | | |
| $\theta_1$ | 1 | 5 | - | - | Number of walls. |
| $\theta_2$ | 0 | 60 | - | - | Inverse of gap distribution used for the walls ($80\% - \theta_2$). Difficulty is inverted such that increasing $\theta_2$ yields smaller wall gaps and more difficult levels. |
| $\theta_3$ | 0 | 10 | - | - | Number of bad food items. |
| | | | **Plunder Hard** | | |
| $\theta_1$ | - | - | 1 | 20 | Number of ships to defeat to complete the level. |
| $\theta_2$ | - | - | 1 | 10 | Juice penalty when defeating a friendly ship. |
| $\theta_3$ | - | - | 0 | 3 | Number of panels that block the agent's line of fire. |

Table 32: Definition of task spaces for games in the Procgen Curriculum Suite that have a four-dimensional task space ($|\Theta| = 4$). E $\downarrow$ and E $\uparrow$ mean the lower and upper bound respectively in easy mode, and H $\downarrow$ and H $\uparrow$ mean the lower and upper bound respectively in hard mode. Games with "easier" modes (easy mode mechanics, but easier task spaces) have these parameters indicated in parentheses.

| | E $\downarrow$ | E $\uparrow$ | H $\downarrow$ | H $\uparrow$ | Description |
|---|---|---|---|---|---|
| | | | **FruitBot Hard:** $|\Theta| = 4$ | | |
| $\theta_1$ | - | - | 1 | 10 | Number of walls. |
| $\theta_2$ | - | - | 0 | 70 | Inverse of gap distribution used for the walls ($80\% - \theta_2$). Difficulty is inverted such that increasing $\theta_2$ yields smaller wall gaps and more difficult levels. |
| $\theta_3$ | - | - | 0 | 10 | Number of bad food items. |
| $\theta_4$ | - | - | 0 | 5 | Probability that a wall will be locked, in steps of 2.5%. |
| | | | **StarPilot:** $|\Theta| = 4$ | | |
| $\theta_1$ | 1 | 500 (250) | 1 | 500 | Finish line spawn time. |
| $\theta_2$ | 1 | 20 (10) | 1 | 20 | Inverse of minimum time between enemies ($30 - \theta_2$). Difficulty is inverted such that increasing $\theta_2$ yields less time between enemies and more difficult levels. |
| $\theta_3$ | 1 | 5 (3) | 1 | 5 | Maximum group size for flyer enemies. |
| $\theta_4$ | 1 | 90 (45) | 1 | 90 | Inverse of minimum time between flyer shots ($100 - \theta_4$). Difficulty is inverted such that increasing $\theta_4$ yields less time between flyer shots and more difficult levels. |

