# OpenReview forum: "CURATE: Automatic Curriculum Learning for Reinforcement Learning Agents through Competence-Based Curriculum Policy Search in Structured Task Spaces"
_TMLR — Decision pending for TMLR_

### Review · Reviewer_Evta · 2026-04-28

**Summary Of Contributions:**

In this work, the authors propose a new algorithm, CURATE, an automated curriculum learning approach in reinforcement learning designed to learn policies for tasks that have sparse rewards in structured task spaces. Their approach is to use an exploration by exploitation strategy, where the agent’s current task solving ability is used to dynamically scale task difficulty. To achieve this, the authors formalize a subclass of underspecified POMDPs known as DOUPOMDPs, where environment parameters are monotonically difficulty-ordered. CURATE conducts a nonlinear, sample-based policy search using Relative Entropy Policy Search (REPS) to identify a high-reward training distribution of unsolved tasks. One additional contribution of this paper is to establish the Procgen Curriculum Suite (PCS), providing standard, structured task spaces for 16 Procgen games as a testing benchmark in the field.

Strengths:

1. Autonomous Scheduling: CURATE effectively aligns the curriculum in an easiest-to-hardest format, without requiring human-designed schedules, optimal initializations, or expert-chosen difficulty increments.
2. Empirical Breadth: The authors validate the framework across three diverse domains of MiniGrid, Procgen, and Bipedal Walker, encompassing discrete/continuous actions, image-based observations, and task spaces ranging from 1D to 8D.
3. Benchmarking: The introduction of the Procgen Curriculum Suite (PCS) provides a human-curated, causal task spaces for systematic curriculum benchmarking.

Weaknesses:
1. Monotonicity Dependency: My biggest concern is that CURATE is explicitly designed for task spaces that exhibit monotonic difficulty along the axes of environment parameters. This limits its applicability to unstructured or non-monotonic task spaces. Such an approach would struggle in lunar lander or other environments where gravity is present, and both the minimum and maximum values of the task parameter are difficult, as it assumes the easiest tasks always reside at the parameter minimums.
2. Generalization Fragility: In certain domains like Leaper, the curriculum can lead to narrow representations where the agent becomes a specialist in the hardest tasks but fails to solve easier ones. Authors address that more techniques like off-curriculum sampling might be needed.
3. Optimization Bottlenecks: In a few instances such as StarPilot, domain randomization on the target task outperforms CURATE, suggesting that the curriculum optimization might sometimes not be the best option available.

**Audience:**

Yes

**Audience Explanation:**

Yes, this work is highly relevant to researchers in curriculum learning, auto reinforcement-learning and reinforcement learning. The practical lessons regarding off-curriculum sampling provide reusable knowledge for the broader TMLR community.

**Broader Impact Concerns:**

A broader impact section is present. The section accurately addresses the impact.

**Claims And Evidence:**

Yes

**Claims Explanation:**

The authors provide enough empirical evidence through extensive learning curves and summary statistics. Performance claims are supported by interquartile mean (IQM) metrics with 95% confidence intervals across multiple trials. The authors also include targeted ablation studies that confirm the necessity of the regularization loss and the distance loss for maintaining task direction and focusing the learning frontier.

**Requested Changes:**

1. One minor change is to soften the claims regarding broadly capable agents in the abstract to reflect the findings in domains like Leaper, where the agent became a specialist and required manual OCS intervention to generalize.

2. Latent Space Discussion: If possible, please expand on the future work regarding the learning of latent task spaces, as the current requirement for predefined and disentangled parameters is a significant barrier to applying CURATE to real-world, unstructured data.

2. Hyperparameter Heuristics: Can the authors provide more detail in the main text on how the automated heuristics for $\lambda_\theta$​ and $\lambda_d$​ were derived, as these are critical for the automatic claim of the framework.

---

> ### Author Response · Authors · 2026-05-18
> **Thank you for your feedback: Our initial response**
>
> We extend our thanks for your time to review our submission and provide your helpful comments. We are happy to hear positive feedback for CURATE, our empirical breadth and evidence, and the Procgen Curriculum Suite. Overall, we are pleased to hear that you feel the claims of the paper are substantiated and justified. Indeed, we find that CURATE offers strong performance in our chosen experiments through the emergence of approximately easy-to-hard curricula through the learning objectives, in particular the competence learning objective. We also hope that the Procgen Curriculum Suite leads to further work in curriculum learning for reinforcement learning agents.
>
> We acknowledge that the assumptions that CURATE uses for the DOUPOMDP will not apply to every domain. However, when these assumptions hold, CURATE leads to the emergence of approximately easy-to-hard curricula that outperform prior methods. A main goal of curriculum learning is to help agents advance on the most difficult tasks at the frontier of agent capability. CURATE is well-designed for this goal. As mentioned, in some domains, this may require a tradeoff to provide less focus on easier tasks and thus yields more narrow representations. However, our experiments show that off-curriculum sampling can help, but better techniques may be used, e.g., other sampling or representation learning techniques. It is also the case that due to the training dynamics, that a curriculum might not be beneficial, e.g., in CaveFlyer, Jumper, Maze, and StarPilot, it is best to just train on the target task. This can happen if the target task is relatively easy, and thus training directly on the target task is tractable, rather than using transfer learning through curricula to progress the policy from easier tasks to the target task.
>
> We are currently investigating additional experiments and revisions to address the overall comments, which will include your requested changes to the text of the paper.

---

> > ### Author Response · Authors · 2026-05-26
> > **New version of paper is available**
> >
> > Dear Reviewer Evta,
> >
> > We wanted to inform you that we have uploaded a new version of the paper. We have listed the full changes in the paper at the top of this page, but we wanted to specifically mention 3 changes:
> > 1. As requested, we have softened the claims of broadly capable agents, e.g., in the abstract and introduction.
> > 2. We have added a discussion of latent space learning in the expanded future work (Sec. 6.2).
> > 3. We have expanded the discussion in App. B.4 for the automatic determination of hyperparameters.
> >
> > We hope these changes address your outstanding concerns. Could you please review the new changes and inform us if there are any remaining concerns? Thank you again for your time.

---

### Review · Reviewer_3zTD · 2026-05-03

**Summary Of Contributions:**

# Contributions:
- Paper introduces CURATE, a curriculum learning algorithm for structured task spaces with monotonic difficulty along disentangled task specification dimensions
- Establish experimentally across 3 different environments:
    1. CURATE produces curricula which move through the task space in the expected manner (i.e. easiest to hardest, directed towards the target [hardest] task)
    2. CURATE performs equally or better than baselines
    3.
- Introduces Procgen Curriculum Suite

# Strengths
- The paper is clearly structured and well written.
- The research questions are well-posed, and mostly clearly evidenced by well-constructed experiments.
- Sufficient detail is provided about the method that a replication seems feasible.
- The paper is clear about the assumptions and limitations of the method.

# Weaknesses
- CURATE has a few design choices that are not fully explored by ablations. Exploring them in this paper may be too much, but perhaps these are worth acknowledging in the future work section.
- Some of the claims could be better supported by evidence (see below).

**Audience:**

Yes

**Audience Explanation:**

Curriculum learning is an important research topic in reinforcement learning. CURATE provides a simple novel approach to this.

**Claims And Evidence:**

No

**Claims Explanation:**

I'm mostly happy with the claims made in this paper, but some aspects need revision. I will be happy to change my answer to this question based on the authors' response. In particular:
- RQ1 is demonstrated convincingly, with the exception of "diverse starting task distributions and therefore resulting curricula". Firstly, it's not clear to me how Fig 3. shows diverse starting task distributions (unless I misunderstand what is meant by 'starting task'?). Secondly, it's not clear how Fig 3. matches with the described initialisation scheme in B.3
- RQ2 is demonstrated fairly convincingly. I'm not sure about the explanation of the `StarPilot-Easier-4D-T` results, but it is difficult to establish this kind of reason and the authors make a reasonable attempt
- RQ3 shows that CURATE has a better TTU & final return aggregated across different tasks from the task space, showing that the policies that CURATE produces can generalise to different tasks. I think this would be more convincingly demonstrated by the addition of heatmaps showing performance (TTU or final return) across the task space (for each (task, baseline) pair there would be one heatmap), as some information relevant to interpreting the generalisation capability is lost by aggregating.
- RQ4 has a reasonable design, but the results are slightly confusing. Table 3 shows quite a large performance gap between CURATE and ACCEL, but Figs 3 and 20 do not show such a large gap. I'm not sure how this discrepancy is explained. I also think 'probability of improvement' metrics would be useful in this case — I imagine ACCEL and CURATE would perform similarly under such a metric, which is fine (my recommendation will not be based on the relative performance of the approaches here) but I think would give a more complete picture about the performance of CURATE.
- RQ5: I think key highlights of the ablation results should be moved into the main text. As far as I understand it the 12-page limit isn't strict for TMLR?

**Requested Changes:**

# Critical
- Addition of heatmaps for RQ3
- Clarification of results for RQ4


# Recommended
- Inclusion of at least key results for RQ5 in main text
- Expansion of future work section to highlight more open questions regarding the design of CURATE. Possibly in the appendix if space constrained. In particular, things like:
    - Why $\mathcal{L}_\text{diff}$ and $\mathcal{L}_\text{dist}$ are treated in a binary manner instead of e.g. smoothly varying them
    - OCS-S fraction
    - etc
- Discussion of the 'scale' of the dimensions in the task space. Some of the penalties are

---

> ### Author Response · Authors · 2026-05-18
> **Thank you for your feedback: Our initial response**
>
> Thank you for your time to review our submission and for your helpful comments. We appreciate the positive feedback regarding that the paper was well-structured and easy to read, including sufficient details for replication and assumptions. We also appreciate that the research questions seemed well-posed and generally supported.
>
> We are happy to hear that the claims were generally reasonable, and, indeed, we find that CURATE offers strong performance for our comprehensive set of experiments. The learning objectives for CURATE are able to induce approximately easy-to-hard curricula, particularly with the competence learning objective, which we find can lead to performant agents. We also hope that the Procgen Curriculum Suite can help drive more research towards curriculum learning.
>
> To address your concerns about revising and strengthen the claims:
> 1. For RQ1, the initial task distributions shown in Figure 4 are from the curriculum policy _after_ the first curriculum learning update. The initialization scheme in B.3 (`InitializeRandomCurriculumPolicy`) is used _as input to_ the first curriculum learning update (e.g., before Figure 4). The initial curriculum learning update starts from the Normal distribution that best represents the uniform distribution over the task space (from `InitializeRandomCurriculumPolicy`, B.3), then through the first curriculum learning update, learns a smaller task distribution (shown in Figure 4). The reason why CURATE yields diverse starting curricula is that, in principle, the starting task distributions are specialized to the student, e.g., which tasks have the highest curriculum reward according to the student’s capabilities. This differs from other curriculum learning algorithms that may require a predefined starting distribution that thus would be used for all students.
> 2. For RQ3, we are currently investigating adding additional heatmaps to the paper.
> 3. For RQ4, the discrepancy may seem to exist because for most of training, CURATE and ACCEL appear comparable (according to Figures 3 and 20). However, we find that the performance improvements of ACCEL tend to reduce over time, and even regress for some environments (e.g., BipedalWalkerStump). Thus, CURATE begins to overtake ACCEL by 15,000 teacher-student updates. For the aggregate plot in Figure 3, at 20,000 teacher-student updates, CURATE has an IQM of about 135, and ACCEL has an IQM of about 85. We believe this is generally in-distribution to the evaluation results in Table 3, where CURATE has an IQM of 132.177 and ACCEL has an IQM of 75.051, given that Figures 3 and 20 use 10 trials with 10 episodes per environment per trial, whereas the results in Table 3 use 10 trials with 100 episodes per environment per trial. So, Table 3 is a more statistically rigorous evaluation and thus some small discrepancies may exist due to the greater sample size.
> 4. For RQ5, we will move the ablation results to the main text.
> 5. We will also update the paper as suggested to update the future work and discuss the scale of the task space dimensions.

---

> > ### Author Response · Authors · 2026-05-26
> > **New version of paper is available**
> >
> > Dear Reviewer 3zTD,
> >
> > We wanted to let you know that we have uploaded a new version of the paper. The full changes in the paper are described at the top of this page, but we wanted to bring to your attention several changes that we believe should address many of your concerns:
> > 1. We updated the caption in Figure 4 to clarify the starting task distributions (RQ1).
> > 2. We have added the heatmaps as requested for RQ3. These are available in App. D, specifically, Figures 9-11. We also add some discussion for the case of Leaper.
> > 3. For RQ4, we added a clarification note in App. H about the difference in sample size and how this could lead to differences between the figures and the tables.
> > 4. As recommended, we provided the key findings of the sensitivity and ablation experiments (App. E) within the main text (RQ5). The new changes are in Sec. 5.5.
> > 5. We added discussion of the scale of the task space to App. B.3.
> >
> > We hope that these changes and our initial response address many if not all of your concerns. Could you please review the new changes and let us know if there are any outstanding concerns? Thank you again for your time.

---

> > > ### Comment · Reviewer_3zTD · 2026-06-02
> > > **Response to revisions**
> > >
> > > Thank you to the authors for your work in addressing my questions. I think the paper is now clearer, and overall I'm happy with the paper.
> > >
> > > Re. RQ3: The heatmaps are an enlightening addition: in particular, Fig. 11a and Fig. 4g perhaps point towards the issue being that the high-water-low-road scenarios being undersampled by CURATE. I'm not entirely convinced by the explanation that "high-water-low-lanes is harder" because DR seems to perform similarly on both high-water-low-lanes and high-lanes-low-water. My guess is that it's due to the path taken in Fig. 4g not sampling many tasks where water is the first lane. But I'm not really sure, and don't feel strongly about this.
> > >
> > > Re. RQ4: I see now that the scale of divergence at the end of Fig. 3 indeed reflects that in Tab. 3, so thank you for the clarification. My final ask would be to provide 'probability of improvement' charts (like Agarwal et al 2021) in the appendix for the results that produced Tab. 3. I won't hold up the review process any longer if this is for some reason not possible, as I don't think it's *crucial*, but reporting these values would paint a clearer picture of the relative performance of CURATE.

---

> > > > ### Author Response · Authors · 2026-06-04
> > > > **Follow-up response and new probability of improvement analysis**
> > > >
> > > > Dear Reviewer 3zTD,
> > > >
> > > > Thank you for your response and for your time to review our updated paper. We are happy to hear that you find the paper clearer and stronger.
> > > >
> > > > Re. RQ3: Agreed and indeed, whether or not the tasks are sampled will play a factor in their performance. With the 37.5% off-curriculum sampling, we see that CURATE’s task performance improves for the high-water-low-road tasks because of the increased sampling of these tasks. One thing to note with the heatmaps is that they are generated at 5,000 combined updates (e.g., essentially at the end of training), and because CURATE quickly reaches the target tasks, by 5,000 combined updates, most of the combined updates will have been spent sampling at or near the target tasks. So, we hypothesize that the heatmap results would also be influenced by the degree to which training on the target tasks can transfer to the tasks that aren’t being sampled, along with the curriculum path taken to reach the target tasks as mentioned.
> > > >
> > > > Re. RQ4: We have calculated the probability of improvement (per Agarwal et al., 2021) on the same data that was used for Tables 3 & 26.
> > > >
> > > > ### Results: BipedalWalker Probability of Improvement
> > > >
> > > > Table: Probability of improvement for CURATE in the BipedalWalker domain. Probability of improvement is shown in terms of IQM with 95\% CI. CI-L is lower CI bound, and CI-U is upper CI bound. Probabilities where CURATE achieves at least 50\% probability of improvement are **bolded.**
> > > >
> > > > | Method | | BW | BWHardcore | BWMax | BWStairs | BWPitGap | BWStump | BWRoughness | **Aggregate**  |
> > > > | --- | --- | --- | --- | --- | --- | --- | --- | --- | --- |
> > > > | CURATE (ours) | IQM | **98.292%** | **81.145%** | 15.826% | **61.280%** | **60.074%** | **85.664%** | **82.515%** | **69.257%** |
> > > > | vs. | CI-L | 97.811% | 79.151% | 14.099% | 58.775% | 57.340% | 83.827% | 80.541% | 68.500% |
> > > > | Domain Rand. | CI-U | 98.716% | 83.112% | 17.597% | 63.736% | 62.833% | 87.437% | 84.417% | 70.011% |
> > > > | | | | | | | | |
> > > > | CURATE (ours) | IQM | **85.979%** | **68.238%** | 12.737% | **54.273%** | **52.243%** | **82.282%** | **68.151%** | **60.557%** |
> > > > | vs. | CI-L | 84.248% | 65.888% | 11.177% | 51.738% | 49.545% | 80.320% | 65.777% | 59.729% |
> > > > | Robust PLR | CI-U | 87.644% | 70.563% | 14.355% | 56.801% | 54.947% | 84.185% | 70.517% | 61.394% |
> > > > | | | | | | | | |
> > > > | CURATE (ours) | IQM | **80.504%** | **60.427%** | 29.927% | **54.648%** | **63.287%** | **71.190%** | **54.950%** | **59.276%** |
> > > > | vs. | CI-L | 78.490% | 57.937% | 27.645% | 52.146% | 60.803% | 68.934% | 52.383% | 58.368% |
> > > > | ACCEL | CI-U | 82.511% | 62.888% | 32.246% | 57.161% | 65.771% | 73.435% | 57.509% | 60.164% |
> > > > | | | | | | | | |
> > > > | CURATE (ours) | IQM | 34.717% | **78.550%** | 19.143% | **62.228%** | 46.135% | **86.470%** | **83.752%** | **58.714%** |
> > > > | vs. | CI-L | 32.266% | 76.498% | 17.284% | 59.766% | 43.551% | 84.796% | 81.907% | 57.896% |
> > > > | ALP-GMM | CI-U | 37.206% | 80.592% | 21.048% | 64.683% | 48.718% | 88.100% | 85.532% | 59.532% |
> > > > | | | | | | | | |
> > > > | CURATE (ours) | IQM | **97.234%** | **90.799%** | 6.894% | **68.209%** | **77.960%** | **93.139%** | **90.611%** | **74.978%** |
> > > > | vs. | CI-L | 96.615% | 89.424% | 5.825% | 65.770% | 75.863% | 91.858% | 89.172% | 74.388%
> > > > | Target | CI-U | 97.798% | 92.111% | 8.022% | 70.597% | 80.001% | 94.374% | 91.987% | 75.561% |
> > > >
> > > > ### Discussion
> > > >
> > > > We find that although CURATE holds a strong lead in terms of aggregate return, the probability of improvement is narrower, yet still at least 50% over all baselines in aggregate across environments. The lowest probability of improvement in aggregate is over ALP-GMM (58.714%), wherein CURATE has less than 50% probability of improvement in three environments, with ACCEL (59.276%) and Robust PLR (60.557%) following in the ranking. In also considering Table 26, we can paint a broader picture of the results, which we attribute to variance both within and across environments. CURATE yields significant performance leads in two environments, BWHardcore and BWStump, with minimum probability of improvement of 60.427% and 71.190%, respectively (both over ACCEL). In four environments (BW, BWStairs, BWPitGap, BWRoughness), CURATE offers competitive performance, ranking first or second in return, but probability of improvement may be near or under 50%. BWMax demonstrates a failure mode for CURATE, as it ranks last in this environment (although no method does well in terms of return). These holistic results show CURATE's overall performance and strong results in some domains while also offering avenues for future improvement.
> > > >
> > > > If this chart addresses your concerns, please let us know, and we would be happy to include it in the paper.

---

> > > > > ### Comment · Reviewer_3zTD · 2026-06-07
> > > > > **Probability of Improvement Results**
> > > > >
> > > > > Dear authors,
> > > > >
> > > > > Thank you for providing these probability of improvement results. I think these provide a fuller picture to the reader of where CURATE works and doesn't, so recommend the inclusion of these results in the paper.
> > > > >
> > > > > Thanks for your efforts in response to my reviews. I will now submit my recommendation.

---

### Review · Reviewer_bS79 · 2026-05-04

**Summary Of Contributions:**

Summary:

This paper introduces CURATE, an automatic curriculum learning algorithm for reinforcement learning agents in structured task spaces where difficulty increases monotonically along each parameter dimension. The core idea is to maintain a Gaussian distribution over task parameters, i.e., the "curriculum policy", and use Relative Entropy Policy Search to update this distribution so that it targets the easiest unsolved tasks near the agent's current competence. As the agent masters easier tasks, the distribution shifts toward harder ones, producing an approximately easy-to-hard curriculum without requiring hand-designed schedules or initial task distributions. The paper also introduces the Procgen Curriculum Suite, a set of 16 Procgen games augmented with structured task spaces to enable systematic benchmarking for curriculum learning. Experiments span MiniGrid MultiRoom, the Procgen Curriculum Suite, and BipedalWalker, covering discrete and continuous action spaces, state-based and image-based observations, and task spaces ranging from 1D to 8D.

Key Strengths
The method is intuitive and well-motivated. The connection between competence-based task selection and the natural ordering of task difficulty is clearly explained, and the paper provides a mechanism, namely, the competence reward, that operationalizes this insight. The experimental coverage is broad, spanning three domains with different observation modalities, action types, and task-space dimensionalities, which provides reasonable confidence that the approach is not narrowly tuned to a single setting. The introduction of the Procgen Curriculum Suite is a useful contribution to the community, as curriculum learning research in Procgen has lacked standardized definitions of task spaces. The paper is thorough in its ablations, examining sensitivity to the solved threshold, regularization, and distance losses, and informed initialization, which helps the reader understand which components matter and when.

Key Weaknesses
The assumption of a difficulty-ordered task space is quite restrictive and is not always easy to verify or construct in practice. The paper acknowledges this but does not explore how violations of monotonicity would degrade performance. The curriculum policy is parameterized as a single Gaussian, which may struggle to represent multimodal frontiers or task spaces with irregular geometry. The comparison with prior curriculum learning methods is somewhat uneven: ALP-GMM is evaluated only on BipedalWalker, and methods like CURROT and GRADIENT are discussed in the related work but never compared experimentally. The teacher sample overhead is nontrivial and, while the paper accounts for it in the reported metrics, the practical cost of curriculum updates could be a concern in more expensive domains.

**Audience:**

Yes

**Audience Explanation:**

Curriculum learning for RL is an active area with practical relevance, and the paper addresses a clear problem of automatically sequencing tasks of increasing difficulty without hand-designed schedules. Researchers working on curriculum learning, environment design, and sample-efficient RL would find useful takeaways here. The competence-based formulation is straightforward to understand and implement, and the experiments surface concrete lessons about threshold selection, the role of regularization and distance losses, and when off-curriculum sampling helps generalization. The introduction of the Procgen Curriculum Suite also has standalone value as a benchmark contribution, providing the community with standardized task spaces for comparing curriculum methods in a domain already widely used. Even readers who find the DOUPOMDP assumption too restrictive for their own settings can learn from the analysis of how structured difficulty progression interacts with exploration and transfer across tasks.

**Broader Impact Concerns:**

Provided broader impact statement is reasonable given the scope of the work.

**Claims And Evidence:**

Yes

**Claims Explanation:**

The central claim that CURATE learns effective, easy-to-hard curricula and matches or outperforms prior methods is well supported. The paper evaluates across three domains with meaningful diversity, reports interquartile means with bootstrapped 95% confidence intervals following Agarwal et al. (2021), and runs a reasonable number of trials per domain. The ablation studies demonstrate the contributions of individual components, such as the regularization loss, distance loss, and informed initialization. The curriculum visualizations in Figure 4 provide concrete qualitative evidence for the claimed progression from easiest to hardest.
However, a few gaps exist between claims and evidence. The comparison set is incomplete as ALP-GMM appears only in BipedalWalker, and CURROT and GRADIENT are discussed as related work, but never evaluated against. This weakens the claim that prior curriculum methods are broadly matched or outperformed. The DOUPOMDP assumption is central to the method, yet no experiment tests robustness when monotonicity is only approximately satisfied. Wall-clock times are also absent, making the practical cost of teacher overhead hard to assess. The paper is honest about domains where CURATE does not perform well, namely StarPilot and CaveFlyer. Nevertheless, the explanations for those cases remain speculative. These are addressable gaps, either through additional experiments or by narrowing certain claims, and they do not undermine the overall soundness of the contribution.

**Requested Changes:**

Critical:
1) Include comparisons with CURROT and GRADIENT: Both methods are discussed in related work as operating in similar settings, yet neither appears in any experiment.
2) Test robustness to violations of the monotonicity assumption: The method depends entirely on the DOUPOMDP framework, but no experiment examines what happens when monotonicity holds only approximately.

Strengthening:
1) Provide more rigorous analysis of failure cases: The explanations for why CURATE underperforms in StarPilot and CaveFlyer are speculative.
2) Report computational overhead beyond sample counts: Teacher samples are accounted for in the metrics, but without wall-clock times or similar measures, the practical cost of curriculum updates remains unclear, particularly for the Procgen experiments with multiple rounds of evaluation per update.

---

> ### Author Response · Authors · 2026-05-17
> **Thank you for your feedback: Our initial response**
>
> Thank you for your time for reviewing our submission and for the helpful feedback. We appreciate the kind words regarding how our work is intuitive, well-motivated, and has strong experimental coverage. We also appreciate the positive feedback on the Procgen Curriculum Suite and the ablations.
>
> We believe that CURATE offers strong performance with respect to difficult tasks and environments, in part due to the approximately easy-to-hard curriculum that emerges from CURATE’s learning objectives, particularly the competence learning objective. Overall, we are happy to hear that the central claim is well-supported. We also believe the Procgen Curriculum Suite would help facilitate greater research within curriculum learning.
>
> Regarding the requested changes and our follow-up responses:
> 1. Regarding GRADIENT [1] and CURROT [2], we are currently investigating CURROT, as the CURROT paper suggests it matches or exceeds the performance of GRADIENT. Unlike GRADIENT and CURROT, CURATE does not require an informed initialization distribution or a target distribution (just the target values). For the case with GRADIENT, CURATE does not require a predetermined schedule (as GRADIENT requires in setting the number of stages). Furthermore, the task spaces we investigate in this work are generally more difficult and complex than the ones explored in the GRADIENT and CURROT works, and optimal transport itself is a complex process. Thus, we feel that CURATE falls into a separate class of curriculum reinforcement learning algorithms than these methods, which distinguishes it from prior work and would be a useful contribution to the curriculum learning literature.
> 2. We chose to only evaluate ALP-GMM [3] on BipedalWalker as the ALP-GMM paper mentions it is designed for continuously parameterized environments (for which BipedalWalker is our representative domain). Thus, we chose to not evaluate it on the discrete environment parameter domains (MultiRoom, Procgen Curriculum Suite).
> 3. We are currently investigating an experiment with an environment that does not adhere to the DOUPOMDP assumptions.
> 4. Regarding domains where CURATE does not perform as well, for certain domains, due to training dynamics, it may be more sample efficient to train in the target task directly as compared to progressing the policy from easier to harder tasks via transfer learning through curricula. This would primarily be driven by how difficult the target task is. For CaveFlyer, Jumper, Maze, and StarPilot, we find that Target is most performant, with CURATE coming in 3rd/2nd/2nd/2nd, as the target task is relatively easy to train in directly. In other domains, this is not the case, and the RL agent makes limited to no progress training directly in the target task because of its difficulty. Thus, we ascribe these results primarily to how difficult the target task is. We note that although CURATE is not the best algorithm in these domains, its performance is not greatly dissimilar for CaveFlyer, Jumper, and Maze.
> 5. Regarding the calculation of wall-clock time: we note that our implementation of CURATE is not optimized for wall-clock time. The separate curriculum update phase that collects teacher samples through policy evaluation can in principle be more parallelized as compared to our current implementation, as we did not focus on optimizing wall-clock time. Thus, reporting wall-clock time would only describe our current implementation, not what a fully optimized algorithm would be able to achieve. We also note that, to avoid the separate curriculum update phase, an extension to CURATE could be explored that draws teacher samples from the training distribution, rather than the separate evaluation phase as is proposed. We can describe this direction in the future work section.
>
> [1] P. Huang et al., “Curriculum Reinforcement Learning using Optimal Transport via Gradual Domain Adaptation,” NeurIPS 2022.
>
> [2] P. Klink et al., “On the Benefit of Optimal Transport for Curriculum Reinforcement Learning,” IEEE Transactions on Pattern Analysis and Machine Intelligence, 2024.
>
> [3] R. Portelas et al., “Teacher Algorithms for Curriculum Learning of Deep RL in Continuously Parameterized Environments,” CoRL 2019.

---

> > ### Author Response · Authors · 2026-05-26
> > **New version of paper is available**
> >
> > Dear Reviewer bS79,
> >
> > We wanted to inform you that we have uploaded a new version of the paper. We have listed the full changes in the paper information at the top of this page, but we wanted to bring to your attention two specific changes:
> > 1. We expanded our analysis of Procgen in Sec. 5.2.2, which includes a discussion of failure cases.
> > 2. We have expanded the future work (Sec. 6.2) to include a discussion of the runtime of CURATE and ways to improve it (e.g., simultaneously training the student and teacher).
> >
> > We are currently continuing to investigate further experiments requested in your original review.
> >
> > We hope these changes and our initial response address some of your concerns. Could you please review the changes and inform us if you have further concerns? Thank you again for your time.

---

> > > ### Author Response · Authors · 2026-05-31
> > > **Update on experiments**
> > >
> > > Dear Reviewer bS79,
> > >
> > > We wanted to provide you with updates on the experiments that were requested.
> > >
> > > ### Non-monotonic task spaces
> > >
> > > **Introduction:**
> > > For experiments with non-monotonic task spaces, we have conducted experiments with FruitBotInv-T Easy (FruitBotInv-Easy-3D-T) and StarPilotInv-T Easier (StarPilotInv-Easier-4D-T), where these games are variations on the base games of the same name. Specifically, in FruitBotInv-T Easy, the second axis controlling wall spacing is inverted, and in StarPilotInv-T Easier, the second (time between enemies) and fourth (time between shots) axes are inverted. Thus, these games are not DOUPOMDPs as the axes are not ordered in difficulty order. For the target tasks, we assume the target tasks are the tasks with the largest environment parameter values to be consistent with the rest of the games and the assumptions of CURATE, although it is not the most difficult task under these experimental conditions.
> > >
> > > **Results:**
> > >
> > > *Note: To keep this post within the character limits, we are shortening the table captions. The captions are as described in the per-game tables in the paper, except the underline has been replaced with italics.*
> > >
> > > **Table 1:** Target task performance results for FruitBotInv-Easy-3D-T.
> > >
> > > | Method | Success Rate | TTUN | TTUN CI | F. Train | F. Train CI | F. Target | F. Target CI |
> > > | --- | --- | --- | --- | --- | --- | --- | --- |
> > > | CURATE (ours) | **100.000%** (6) | 6.432% | (3.319%, 15.126%) | 9.259 | (9.054, 9.414) | *9.434*$^*$ | (9.297, 9.590) |
> > > | Domain Rand. | **100.000%** (6) | *3.065%*$^*$ | (2.491%, 4.081%) | **9.463** | (9.332, 9.710) | 9.180 | (8.750, 9.473) |
> > > | Robust PLR | **100.000%** (6) | 3.106%$^*$ | (2.311%, 3.950%) | 9.362$^*$ | (9.212, 9.610) | *9.434*$^*$ | (9.102, 9.551) |
> > > | ACCEL | **100.000\%** (6) | 36.791% | (19.543%, 48.287%) | 8.426 | (7.256, 9.528) | 7.109 | (4.902, 7.773) |
> > > | Target | **100.000%** (6) | **2.303%** | (1.622%, 3.638%) | *9.433*$^*$ | (9.218, 9.598) | **9.512** | (9.355, 9.688) |
> > >
> > > **Table 2:** Target task performance results for StarPilotInv-Easier-4D-T.
> > >
> > > | Method | Success Rate | TTUN | TTUN CI | F. Train | F. Train CI | F. Target | F. Target CI |
> > > | --- | --- | --- | --- | --- | --- | --- | --- |
> > > | CURATE (ours) | **100.000%** (6) | 5.605% | (4.179%, 7.112%) | 9.424 | (9.361, 9.529) | *9.355*$^*$ | (9.141, 9.531) |
> > > | Domain Rand. | **100.000%** (6) | *4.163%* | (3.704%, 4.376%) | *9.704* | (9.634, 9.735) | 8.281 | (8.027, 8.535) |
> > > | Robust PLR | **100.000%** (6) | 6.572% | (3.720%, 7.801%) | 7.329 | (4.197, 8.698) | 8.730 | (8.398, 9.023) |
> > > | ACCEL | 0.000\% (0) | 100.000% | (100.000%, 100.000%) | **10.000** | (10.000, 10.000) | 1.055 | (0.703, 1.367) |
> > > | Target | **100.000%** (6) | **1.967%** | (1.672%, 2.155%) | 9.529 | (9.489, 9.583) | **9.453** | (9.219, 9.648) |
> > >
> > > **Discussion:**
> > > We find that CURATE's performance in these non-monotonic task spaces is reduced and high-variance as compared to monotonic task spaces. However, all CURATE trials were successful, and CURATE outperforms some baselines in TTUN (ACCEL and Robust PLR for the case of StarPilotInv-T Easier). In FruitBotInv-T Easy, CURATE ranks fourth but has highly variable performance. Two trials (2.557%, 2.655%) fall within the best method’s 95% CI for normalized time to threshold, but two trials have large outliers (11.865%, 21.567%). For StarPilotInv-T Easier, CURATE ranks third but outperforms Robust PLR and ACCEL. Although DR and NC are best in these domains, we find CURATE’s reduced performance reasonable for these off-design conditions given that all trials were still successful and it still outperforms some baselines. Additionally, CURATE still continues to offer competitive final target task return.
> > >
> > > **Paper:**
> > > We will update the paper with these experiments if they are satisfactory to address your concerns.
> > >
> > > ### Baselines
> > >
> > > We recognize that your original review requested additional baselines, e.g., GRADIENT and CURROT, so we will continue to work on this task. Should the rebuttal end before we have these new baseline results, we are open to narrowing our claims for matching or outperforming prior methods to only the subclass of curriculum learning algorithms for which an informed initialization and predefined schedule are not required. Our rationale is because CURATE has differing assumptions than GRADIENT or CURROT, e.g., CURATE does not require an informed initialization like GRADIENT or CURROT, and doesn’t require predefined schedules like GRADIENT. Thus, we would suggest that CURATE represents a different class of algorithms (e.g., not based on optimal transport) and thus would be a valuable contribution to the literature.

---

> > > > ### Comment · Reviewer_bS79 · 2026-05-31
> > > > **My concerns have been partially addressed.**
> > > >
> > > > Thank you for the updated paper and responses. The expanded failure case analysis in Sec. 5.2.2 offers a more grounded discussion of when curricula hurt rather than help, and the non-monotonic task space experiments are a genuine and meaningful addition that directly address one of the original concerns. The future work section now also gives a clearer picture of the runtime tradeoffs involved in the separate curriculum update phase. These are substantive improvements that strengthen the paper.
> > > >
> > > > **On the absence of CURROT and GRADIENT comparisons**
> > > >
> > > > The authors argue that CURATE belongs to a distinct class of algorithms, making comparisons with CURROT and GRADIENT unnecessary. This reasoning is circular. The claimed advantages of CURATE over these methods are exactly what an empirical evaluation should test. CURROT operates in the same parameterized task space setting as CURATE and also requires no predefined schedule, which undermines a key distinguishing claim. The authors also suggest skipping GRADIENT (NeurIPS 2022) because CURROT (ICML 2022) supersedes it, but the two papers appeared nearly simultaneously, with CURROT being published prior to GRADIENT, which makes this historically inaccurate. Currently, the paper compares against only ALP-GMM as a structured curriculum baseline, and only on BipedalWalker. If the authors choose not to add CURROT comparisons, they must explicitly narrow their claims in the abstract, contributions, and conclusion, and discuss CURROT and GRADIENT as uncompared related methods in the limitations section.
> > > >
> > > > **On runtime**
> > > >
> > > > The argument that reporting wall-clock time would only reflect an unoptimized implementation applies to every paper in this area and does not justify omitting it entirely. The authors must report wall-clock time for the current implementation, with a caveat that optimization could reduce overhead. Future work directions do not substitute for empirical accounting of present costs.

---

> > > > > ### Author Response · Authors · 2026-06-02
> > > > > **Response and update on runtime**
> > > > >
> > > > > Dear Reviewer bS79,
> > > > >
> > > > > We are happy to hear that the updated paper and non-monotonic task space experiments addressed some of your concerns. Thank you for your response and for your time to review the new changes to the paper.
> > > > >
> > > > > Regarding the additional baselines, it is acceptable to us to focus our claims in the abstract, contributions, and conclusion to only the subclass of curriculum learning algorithms for which informed initialization and predefined schedules are not required. We can also mention CURROT and GRADIENT in the limitations. Our proposed focus on CURROT was not due to when these approaches first entered the literature, but rather, that the IEEE Transactions on Pattern Analysis and Machine Intelligence paper (P. Klink et al., “On the Benefit of Optimal Transport for Curriculum Reinforcement Learning”, IEEE Transactions on Pattern Analysis and Machine Intelligence, 2024) by nearly the same authorship team as the ICML 2022 CURROT paper directly compares GRADIENT and CURROT and found that CURROT matches or outperforms GRADIENT for their chosen environments.
> > > > >
> > > > > Regarding wall-clock time, we have finished this analysis as requested. We will present this in the following posts. Please let us know if this addresses your concerns regarding runtime. Should this address concerns over runtime, the next revision of the paper will include the non-monotonic task space experiments and these runtime analyses.

---

> > > > > > ### Author Response · Authors · 2026-06-02
> > > > > > **First runtime analysis**
> > > > > >
> > > > > > ### Introduction to analysis
> > > > > >
> > > > > > For this first analysis, we use the terminology of "algorithm update" to refer to a unit of temporal progression for the curriculum learning algorithm. An algorithm update may or may not include a student PPO update (e.g., Robust PLR has a 50% chance of running a student PPO update on an algorithm update). For CURATE, an algorithm update may or may not include running a curriculum update, depending on if the curriculum update was triggered by CURATE’s logic.
> > > > > >
> > > > > > Further, due to limitations in our logging implementation, we do not have the initial timestamp upon starting the algorithm, which means that we do not have the runtime of the first algorithm update. However, we have the runtime for all other algorithm updates. Thus, for all algorithms, we estimate the time of the first algorithm update based on the average statistics. For example, in DR/HC, these algorithms always conduct a student PPO update each algorithm update, so we add the average time for an algorithm update to account for the first update. For CURATE, we use the average duration of algorithm updates with a curriculum update to estimate the first algorithm update, which always includes the initial curriculum update. Hence, we refer to our metrics as estimated runtime (where the estimation is because of the estimated runtime of the first algorithm update). However, the estimate should be reasonably similar to the actual runtime, given that only one update out of many is estimated.
> > > > > >
> > > > > > Lastly, we note that we did not optimize CURATE for runtime, so we present these results to account for the actual time of the experiments. The best-case runtime for a fully optimized version of CURATE would be improved, perhaps significantly so, from what we present. Runtime can be further improved with optimization, e.g., better parallelization of collecting teacher samples. We use parallelization for Procgen and BipedalWalker, but not for MultiRoom.
> > > > > >
> > > > > > For this first analysis, we focus on MultiRoom as all baselines were successful in solving the target tasks (except for Target, which we exclude from this analysis). We specifically analyze the runtime until the target tasks are solved, which is CURATE’s primary design use case.
> > > > > >
> > > > > > ### Results
> > > > > >
> > > > > > Runtime performance for MultiRoom to solve the target tasks for 10 trials. Summary statistics are shown in terms of total estimated runtime (ER-T) and estimated runtime per algorithm update (ER-AU). ER-T is $\times 10^3$ seconds, and ER-AU is seconds per algorithm update. ER-T and ER-AU are shown in terms of IQM with 95% CI. The best approach for each metric is **bolded**, the second best approach is *italicized*, and an asterisk ($^*$) denotes that the approach falls within the best approach's CI.
> > > > > >
> > > > > > | Method | ER-T $\downarrow$ ($\times 10^3$) | ER-T CI ($\times 10^3$) | ER-AU $\downarrow$ | ER-AU CI |
> > > > > > | --- | --- | --- | --- | --- |
> > > > > > | CURATE-0.3 (ours) | *13.290*$^*$ | (11.751, 39.950) | 11.932 | (11.229, 33.717) |
> > > > > > | Hand Curr. | **13.145** | (10.841, 15.700) | 12.566 | (10.337, 14.847) |
> > > > > > | Domain Rand. | 21.527 | (19.450, 23.584) | *11.582* | (10.658, 12.545) |
> > > > > > | Robust PLR | 46.141 | (42.186, 50.327) | **10.309** | (10.103, 10.555) |
> > > > > > | ACCEL | 76.229 | (52.230, 101.165) | 25.802 | (16.774, 35.352) |
> > > > > >
> > > > > > ### Discussion
> > > > > >
> > > > > > We find that the best performing algorithm in terms of total runtime (ER-T) is HC, and CURATE’s ER-T is competitive in terms of IQM, ranking second and within HC’s CI. However, we observe that CURATE’s ER-T is highly variable. Both algorithms benefit by solving the target tasks relatively quickly compared to the other baselines. However, HC uses a predefined schedule, and thus avoids the need for teacher updates that may lead to the higher variance of CURATE's ER-T. All algorithms except ACCEL have similar per algorithm update runtime (ER-AU) in terms of IQM. CURATE’s ER-AU is also highly variable as the runtime of an individual algorithm update will vary depending on if a curriculum update has or has not occurred. We hypothesize that ACCEL’s increased ER-AU is due to the implementation for mutating levels.

---

> > > > > > > ### Author Response · Authors · 2026-06-02
> > > > > > > **Second analysis on runtime (1/2)**
> > > > > > >
> > > > > > > In our second runtime analysis, we first present the results, then in a second post, we will provide the discussion.
> > > > > > >
> > > > > > > ### Results
> > > > > > >
> > > > > > > Table: Runtime performance for CURATE to either solve the target tasks (MultiRoom, Procgen Curriculum Suite) or reach 20,000 combined (student and teacher) updates (BipedalWalker). MultiRoom and BipedalWalker use 10 trials; Procgen Curriculum Suite uses 6 trials. Summary statistics are shown in terms of estimated runtime (ER), runtime for algorithm updates without a curriculum update (RNC), and estimated runtime for algorithm updates with a curriculum update (ERC). Metrics are presented in terms of totals (-T) or per applicable algorithm update (-AU). ERC-F is the fraction of ERC-T relative to ER-T, expressed as a percentage. ER-T, RNC-T, and ERC-T are $\times 10^3$ seconds. ER-AU, RNC-AU, and ERC-AU are seconds per algorithm update. For all metrics, a lower number is better. All metrics are shown in terms of IQM with 95% CI (Agarwal et al., 2021).
> > > > > > >
> > > > > > > | Method | ER-T | ER-T CI | ER-AU | ER-AU CI | RNC-T | RNC-T CI | RNC-AU | RNC-AU CI | ERC-T | ERC-T CI | ERC-AU | ERC-AU CI | ERC-F | ERC-F CI |
> > > > > > > | --- | --- | --- | --- | --- | --- | --- | --- | --- | --- | --- | --- | --- | --- | --- |
> > > > > > > | MultiRoom | 13.290 | (11.751, 39.950) | 11.932 | (11.229, 33.717) | 5.233 | (4.618, 14.147) | 5.028 | (4.615, 12.304) | 8.071 | (6.957, 25.878) | 147.074 | (141.862, 460.707) | 61.436% | (58.662%, 64.251%) |
> > > > > > > | BigFish | 24.086 | (18.356, 30.682) | 78.434 | (76.332, 81.176) | 21.951 | (16.714, 27.865) | 75.948 | (73.933, 78.680) | 2.134 | (1.624, 2.747) | 117.121 | (114.174, 123.655) | 8.893% | (8.478%, 9.201%) |
> > > > > > > | BossFight | 27.663 | (20.248, 35.527) | 154.753 | (128.408, 187.699) | 25.263 | (16.939, 32.531) | 143.729 | (122.500, 170.763) | 2.915 | (2.029, 3.911) | 227.425 | (219.915, 688.567) | 9.348% | (7.690%, 18.955%) |
> > > > > > > | CaveFlyer | 6.164 | (5.535, 6.705) | 43.491 | (42.231, 44.226) | 5.059 | (4.562, 5.583) | 38.419 | (37.046, 38.828) | 1.070 | (0.959, 1.205) | 122.131 | (116.777, 126.806) | 17.615% | (16.211%, 18.683%) |
> > > > > > > | Chaser | 21.307 | (16.106, 40.527) | 40.042 | (39.053, 41.213) | 19.451 | (14.499, 38.350) | 38.126 | (35.626, 39.347) | 1.856 | (1.428, 2.357) | 113.997 | (109.102, 174.491) | 7.709% | (5.933%, 11.585%) |
> > > > > > > | Climber | 15.496 | (13.450, 18.007) | 53.096 | (48.317, 58.479) | 13.951 | (11.855, 15.583) | 48.369 | (44.530, 54.654) | 1.622 | (1.346, 2.651) | 137.784 | (124.208, 166.166) | 11.383% | (8.582%, 15.220%) |
> > > > > > > | CoinRun | 11.246 | (9.430, 13.705) | 54.070 | (48.122, 60.535) | 9.483 | (7.903, 11.425) | 48.624 | (42.516, 55.440) | 1.800 | (1.497, 1.942) | 140.706 | (133.598, 151.572) | 15.102% | (13.816%, 17.181%) |
> > > > > > > | Dodgeball | 27.792 | (23.407, 33.022) | 37.530 | (37.117, 37.861) | 24.835 | (20.925, 29.337) | 35.049 | (34.694, 35.343) | 2.957 | (2.409, 3.684) | 91.290 | (89.097, 92.994) | 10.628% | (10.323%, 11.150%) |
> > > > > > > | FruitBot | 23.261 | (20.922, 26.166) | 49.628 | (46.059, 56.499) | 15.868 | (13.966, 19.505) | 35.801 | (33.960, 43.389) | 6.663 | (6.233, 8.042) | 232.280 | (226.079, 296.418) | 29.745% | (25.298%, 35.671%) |
> > > > > > > | Heist | 25.267 | (23.413, 28.035) | 59.786 | (57.952, 61.349) | 19.373 | (18.032, 21.423) | 47.810 | (47.347, 48.448) | 6.016 | (5.181, 6.644) | 325.814 | (306.334, 346.208) | 23.714% | (20.990%, 24.901%) |
> > > > > > > | Jumper | 7.203 | (5.998, 8.188) | 74.427 | (69.773, 78.023) | 6.679 | (5.496, 7.491) | 71.659 | (67.284, 75.848) | 0.550 | (0.463, 0.673) | 124.800 | (116.821, 139.192) | 7.826% | (6.653%, 9.287%) |
> > > > > > > | Leaper | 14.946 | (10.667, 17.743) | 41.112 | (33.303, 48.679) | 12.481 | (9.025, 14.179) | 35.230 | (29.442, 43.624) | 1.787 | (1.468, 4.237) | 97.606 | (93.594, 204.917) | 14.227% | (11.149%, 25.027%) |
> > > > > > > | Maze | 12.330 | (11.178, 13.766) | 36.539 | (35.510, 37.088) | 8.695 | (7.928, 9.493) | 27.281 | (26.924, 27.752) | 3.635 | (3.148, 4.244) | 172.501 | (157.157, 183.360) | 29.693% | (27.908%, 31.301%) |
> > > > > > > | Miner | 6.278 | (5.651, 8.597) | 46.126 | (44.649, 48.091) | 5.132 | (4.581, 7.059) | 40.327 | (38.880, 42.266) | 1.147 | (1.057, 1.548) | 127.398 | (124.273, 133.457) | 18.308% | (17.715%, 19.063%) |
> > > > > > > | Ninja | 14.764 | (12.195, 15.896) | 39.632 | (38.529, 40.672) | 12.610 | (10.564, 13.665) | 35.926 | (35.338, 36.596) | 2.078 | (1.574, 2.350) | 106.731 | (100.557, 112.521) | 13.849% | (12.637%, 15.248%) |
> > > > > > > | Plunder | 14.136 | (9.841, 19.637) | 69.657 | (65.887, 70.846) | 12.951 | (8.727, 18.418) | 65.564 | (62.356, 66.840) | 1.138 | (0.875, 1.287) | 202.199 | (186.978, 217.524) | 7.748% | (6.046%, 10.635%) |
> > > > > > > | StarPilot | 35.679 | (28.832, 42.040) | 41.284 | (34.803, 45.380) | 29.035 | (23.672, 33.995) | 35.630 | (29.991, 39.769) | 6.644 | (5.220, 8.045) | 129.066 | (108.510, 132.928) | 18.341% | (17.700%, 19.427%) |
> > > > > > > | Bipedal | 369.182 | (346.708, 385.967) | 18.635 | (17.492, 19.485) | 239.102 | (222.397, 252.845) | 12.125 | (11.266, 12.817) | 127.454 | (122.196, 133.313) | 1387.705 | (1339.928, 1474.129) | 34.744% | (33.596%, 36.866%) |

---

> > > > > > > > ### Author Response · Authors · 2026-06-02
> > > > > > > > **Second analysis on runtime (2/2)**
> > > > > > > >
> > > > > > > > ### Discussion
> > > > > > > >
> > > > > > > > This second analysis investigates the estimated runtime performance for CURATE (ER), separated into either the runtime for algorithm updates in which a curriculum update did not occur (RNC), or the estimated runtime for algorithm updates in which a curriculum update did occur (ERC). As in the preceding analysis, the first algorithm update that contains the initial curriculum update is estimated based on average statistics.
> > > > > > > >
> > > > > > > > Generally, we find a large variance in terms of how expensive the curriculum updates are, relative to the overall runtime. For MultiRoom, we find that this fraction (ERC-F) is the highest at 61.436%. However, this is primarily driven by teacher samples being collected in series in this domain (1 x 8 per teacher update). Parallelization would likely reduce this, potentially significantly so. Although curriculum updates are a large portion of CURATE’s runtime, we find the overall runtime competitive to the best performing algorithm (per our first analysis) in terms of IQM (although highly variable).
> > > > > > > >
> > > > > > > > ERC-F is lower for the other domains, which use parallelized samples (Procgen: 4 x 4 per teacher update, BipedalWalker: 8 x 8 per teacher update). For Procgen, aggregate ERC-F over all 16 games has an IQM of 13.808% (CI: 10.266%, 18.419%), which is in part due to the optimized implementation of the backend of Procgen (e.g., it is written in C++ and has built-in support for parallelization). BipedalWalker is the longest runtime experiment by far due to the large number of policy updates required in this domain. The relatively larger ERC-F (34.744%) is due to requiring 64 teacher samples per teacher update because of the larger task space dimensionality. However, the curriculum update is not triggered as frequently, preventing ERC-F from being larger.
> > > > > > > >
> > > > > > > > In all cases, although CURATE’s runtime can be further improved with additional optimizations, we found that runtime was satisfactory for experimentation purposes.